# REPLAY CONCURRENTLY OR SEQUENTIALLY? A THEORETICAL PERSPECTIVE ON REPLAY IN CONTINUAL LEARNING

## ABSTRACT

Replay-based methods have shown superior performance to address catastrophic forgetting in continual learning (CL), where a subset of past data is stored and generally replayed together with new data in current task learning. While seemingly natural, it is questionable, though rarely questioned, if such a concurrent replay strategy is always the right way for replay in CL. Inspired by the fact in human learning that revisiting very different courses sequentially before final exams is more effective for students, an interesting open question to ask is whether a sequential replay can benefit CL more compared to a standard concurrent replay. However, answering this question is highly nontrivial considering a major lack of theoretical understanding in replay-based CL methods. To this end, we investigate CL in overparameterized linear models and provide a comprehensive theoretical analysis to compare two replay schemes: 1) *Concurrent Replay*, where the model is trained on replay data and new data concurrently; 2) *Sequential Replay*, where the model is trained first on new data and then sequentially on replay data for each old task. By characterizing the explicit form of forgetting and generalization error, we show in theory that sequential replay tends to outperform concurrent replay when tasks are less similar, which is corroborated by our simulations in linear models. More importantly, our results inspire a novel design of a hybrid replay method, where only replay data of similar tasks are used concurrently with the current data and dissimilar tasks are sequentially revisited using their replay data. As depicted in our experiments on real datasets using deep neural networks, such a hybrid replay method improves the performance of standard concurrent replay by leveraging sequential replay for dissimilar tasks. By providing the first comprehensive theoretical analysis on replay, our work has great potentials to open up more principled designs for replay-based CL.

## 1 INTRODUCTION

Continual learning (CL) (Parisi et al., 2019) seeks to build an agent that can learn a sequence of tasks continuously without access to old task data, resembling human's capability of lifelong learning. One of the major challenges therein is the so-called *catastrophic forgetting* (Kirkpatrick et al., 2017), i.e., the agent can easily forget the knowledge of old tasks when learning new tasks. A large amount of studies have been proposed to address this issue, among which *replay-based* approaches (Rolnick et al., 2019) have demonstrated the state-of-the-art performance. The main idea behind is to store a subset of old task data in the memory and replay them when learning new tasks, where a widely adopted strategy for training is **concurrent replay** (Evron et al., 2024), i.e., train the model *concurrently* on new task data and the replay data.

While the concurrent replay strategy seems very natural and has shown successful performance to address catastrophic forgetting, it is indeed questionable whether this strategy is always the right way for replay in CL as we consider the following aspects. 1) *From the perspective of human learning.* In daily life, a common strategy to prevent forgetting is to review old knowledge. For example, suppose a student needs to learn a series of topics over a semester before taking an exam, and each topic corresponds to one task in CL. Intuitively, if these topics are highly related to each other, incorporating the knowledge of old topics into learning a new topic can be an effective strategy

to strengthen the new learning and simultaneously reduce the forgetting of old knowledge, which is analogous to *concurrent replay*. However, if the topics are very different from each other, a common practice that a student often takes is to learn new topics first and then go over old topics to mitigate the forgetting. Here, such a *sequential* replay may lead to better outcome in the exam. 2) *From the perspective of multi-task learning.* Learning multiple tasks all at once may lead to poor learning performance due to the potential interference among gradients of different tasks Yu et al. (2020), whereas standard CL without regularization and replay may even achieve less forgetting for more dissimilar tasks Lin et al. (2023). Thus motivated, an interesting and open question to ask is:

*Question: Whether sequential replay will serve as an appealing replay strategy to complement the standard concurrent replay, and when will it be advantageous over concurrent replay for CL?*

To answer this question from a theoretical perspective, we study replay-based CL through the lens of overparameterized linear models to gain useful insights, by following a recent series of theoretical studies in CL (Lin et al., 2023; Evron et al., 2022; Ding et al., 2024; Li et al., 2024). However, none of those previous studies analyzed the replay-based methods. The only theoretical work that studied the replay-based methods is the recent concurrent work (Banayeeanzade et al., 2024). But this work considered only the standard *concurrent* replay method, not from the new perspective of *sequential* replay.

In this work, to capture the idea and advantage of sequential replay, we propose a novel replay strategy, in which the agent *sequentially* revisits each old task and trains the model with the corresponding replay data after the current task is well learned.

**Main Contributions.** We summarize our main contributions as follows.

- First of all, we provide the *first* explicit closed-form expressions for the expected value of forgetting and generalization error for both concurrent replay strategy and sequential replay strategy under an overparameterized linear regression setting. Note that the blending of samples from old tasks in concurrent replay introduces significant intricacies related to task correlation in theoretical analysis. To address this challenge, we partition training data into blocks based on different tasks, which enables us to further calculate the task interference using the properties of block matrix. In particular, our theoretical results demonstrate how the performance of replay-based CL is affected by various factors, including task similarity and memory size.

- Secondly, we propose a novel replay strategy, i.e., *sequential replay*, to sequentially revisit old tasks after the current task is fully learned. By characterizing the explicit closed-form expressions for the expected forgetting and generalization error for sequential replay and comparing with the concurrent replay, we give an affirmative answer to the open question above. More importantly, we rigorously characterize the conditions when sequential replay can benefit CL more than concurrent replay, in terms of both forgetting and generalization error, which is also consistent with our motivations above: Sequential replay outperforms concurrent replay if tasks in CL are dissimilar, and the performance improvement is larger when the tasks are more dissimilar. Numerical simulations on linear models further corroborate our theoretical results.

- Last but not least, our theoretical insights can indeed go beyond the linear models and guide the practical algorithm design for replay-based CL with deep neural networks (DNNs). More specifically, we merge the idea of sequential replay into standard replay-based CL with concurrent replay, leading to a hybrid replay approach where 1) old tasks dissimilar to the current task will be revisited by using sequential replay (guided by our theory that suggests more benefit if dissimilar tasks are revisited sequentially) and 2) the replay data for the remaining old tasks (that are sufficiently similar to the current task) will still be used concurrently with current task data. Our experiments on real datasets with DNNs verify that our hybrid approach can perform better than concurrent replay and the advantage is more apparent when tasks are less similar.

## 2 RELATED WORK

**Empirical studies in CL.** CL has drawn significant attention in recent years, with numerous empirical approaches developed to mitigate the issue of catastrophic forgetting. Architecture-based approaches combat catastrophic forgetting by dynamically adjusting network parameters (Rusu et al., 2016) or introducing architectural adaptations such as an ensemble of experts (Rypeść et al., 2024).

Regularization-based methods constrain model parameter updates to preserve the knowledge of previous tasks (Kirkpatrick et al., 2017; Magistri et al., 2024). Memory-based methods address forgetting by storing some information of old tasks in the memory and leveraging the information during current task learning, which can be further divided into orthogonal projection based methods and replay-based methods. The former stores gradient information of old tasks and uses this to modify the optimization space for the current task (Saha et al., 2021; Lin et al., 2022), while the latter stores and reuses a tiny subset of representative data, known as exemplars. Critical design considerations in empirical replay-based methods mainly include varying exemplar sampling and utilization schemes. Exemplar sampling methods involve reservoir sampling (Chrysakis & Moens, 2020) and an information-theoretic evaluation of exemplar candidates (Sun et al., 2022). Some other work such as Shin et al. (2017) retains past knowledge by replaying "pseudo-data" constructed from input data instead of storing raw input. Replay methods mostly assume a concurrent training scheme that trains the model using a mix of input data and sampled exemplars (Dokania et al., 2019; Rebuffi et al., 2017; Garg et al., 2024). Other exemplar utilization methods include Lopez-Paz & Ranzato (2017) and Chaudhry et al. (2018), which use exemplar to impose constraints in the gradient space.

**Theoretical studies in CL.** Compared to the vast amount of empirical studies in CL, the theoretical understanding of CL is very limited but has started to attract much attention very recently. Bennani et al. (2020); Doan et al. (2021) investigated CL performance for the orthogonal gradient descent approach in NTK models theoretically. Yin et al. (2020) focused on regularization-based methods and proposed a framework, which requires second-order information to approximate loss function. Cao et al. (2022); Li et al. (2022) characterized the benefits of continual representation learning from a theoretical perspective. Evron et al. (2023) connected regularization-based methods with Projection Onto Convex Sets. Recently, a series of theoretical studies proposed to leverage the tools of overparameterized linear models to facilitate better understanding of CL. Evron et al. (2022) studied the performance of forgetting under such a setup. After that, Lin et al. (2023) characterized the performance of CL in a more comprehensive way, where they discuss the impact of task similarities and the task order. Goldfarb & Hand (2023) illustrated the joint effect of task similarity and overparameterization. Zhao et al. (2024) provided a statistical analysis of regularization-based methods. More recently, Li et al. (2024) further theoretically investigated the impact of mixture-of-experts on the performance of CL in linear models.

Different from all the previous studies, we seek to fill up the theoretical understanding for replay-based CL. Note that one concurrent study Banayeeanzade et al. (2024) also investigates replay-based CL in overparameterized linear models with concurrent replay. However, one key difference here is that we propose a novel replay strategy, i.e., the sequential replay, and theoretically show its benefit over concurrent replay for dissimilar tasks. Our theoretical results further motivate a new algorithm design for CL in practice, which demonstrates promising performance on DNNs.

## 3 PROBLEM SETTING

We consider a common CL setup consisting of $T$ tasks where each task arrives sequentially in time $t \in [T]$ and is learned sequentially by one model. Here $[T] := \{1, 2, ..., T\}$ for any positive integer $T$. Let $\boldsymbol{I}_p$ denote the $p \times p$ identity matrix and let $\|\cdot\|$ denote the $\ell_2$-norm.

**Data Model.** We adopt the setting of linear ground truth which is commonly used in the theoretical analysis of various machine learning methods including CL (e.g., Lin et al. (2023)). Specifically, For each task $t \in [T]$, a sample $(\hat{\boldsymbol{x}}_t, y_t)$ is generated by a linear ground truth model:

$$y_t = \hat{\boldsymbol{x}}_t^\top \hat{\boldsymbol{w}}_t^* + z_t, \tag{1}$$

where $\hat{\boldsymbol{x}}_t \in \mathbb{R}^{s_t}$ denotes $s_t$ true features, $y_t \in \mathbb{R}$ denotes the output, $\hat{\boldsymbol{w}}^* \in \mathbb{R}^{s_t}$ denotes the ground truth parameters, and $z_t \in \mathbb{R}$ denotes the noise. Notice that in practice, true features are unknown, and typically more features are selected to ensure that all relevant features are included. Mathematically, letting $\mathcal{S}_t$ denote the set of true features of task $t$ and letting $\mathcal{W}$ denote the set of chosen features in our model. We assume $\bigcup_{t \in [T]} \mathcal{S}_t \subseteq \mathcal{W}$. We use $p$ to denote the number of chosen features, i.e., $|\mathcal{W}| = p$. (Of course, $\bigcup_{t \in [T]} \mathcal{S}_t \subseteq \mathcal{W}$ implies that $p \geq \max_{t \in [T]} s_t$.) With this assumption, we expand $\hat{\boldsymbol{w}}_t^* \in \mathbb{R}^{s_t}$ to a sparse $p$-dimensional vector $\boldsymbol{w}_t^* \in \mathbb{R}^p$ by filling zeros in the positions corresponding to $\mathcal{W}\backslash\mathcal{S}_t$. Thus, eq. (1) can be written as:

$$y_t = \boldsymbol{x}_t^\top \boldsymbol{w}_t^* + z_t, \tag{2}$$

where $\boldsymbol{x}_t, \boldsymbol{w}_t^* \in \mathbb{R}^p$. In other words, $(\boldsymbol{x}_t, y_t)$ is the sample used in the training process.

**Dataset.** For each task $t \in [T]$, there are $n_t$ training samples $(\boldsymbol{x}_{t,i}, y_{t,i})_{i \in [n_t]}$. We stack those samples into matrices/vectors to obtain the dataset $\mathcal{D}_t = \{(\boldsymbol{X}_t, \boldsymbol{Y}_t) \in \mathbb{R}^{p \times n_t} \times \mathbb{R}^{n_t}\}$, By eq. (2), we have

$$\boldsymbol{Y}_t = \boldsymbol{X}_t^\top \boldsymbol{w}_t^* + \boldsymbol{z}_t, \tag{3}$$

where $\boldsymbol{X}_t \coloneqq [\boldsymbol{x}_{t,1}\ \boldsymbol{x}_{t,2}\ \cdots\ \boldsymbol{x}_{t,n_t}]$, $\boldsymbol{Y}_t \coloneqq [y_{t,1}\ y_{t,2}\ \cdots\ y_{t,n_t}]^\top$, and $\boldsymbol{z}_t \coloneqq [z_{t,1}\ z_{t,2}\ \cdots\ z_{t,n_t}]^\top$. To simplify our theoretical analysis, we consider *i.i.d.* Gaussian features and noise, i.e., each element of $\boldsymbol{X}_t$ follows *i.i.d.* standard Gaussian distribution, and $\boldsymbol{z}_t \sim \mathcal{N}(0, \sigma_t^2 \boldsymbol{I}_{n_t})$ where $\sigma_t \geq 0$ denotes the noise level. To make our result easier to interpret, we let $\sigma_t = \sigma$ and $n_t = n$ for all $t \in [T]$.

**Memory.** For any task $t \geq 2$, besides $\mathcal{D}_t$, the agent has an overall memory dataset $\mathcal{M}_t$ that contains separate memory datasets $\mathcal{M}_{t,i}$ for each of the previous tasks $i \in [t-1]$, i.e., $\mathcal{M}_t = \bigcup_{i=1}^{t-1} \mathcal{M}_{t,i}$ where $\mathcal{M}_{t,i} = (\widetilde{\boldsymbol{X}}_{t,i}, \widetilde{\boldsymbol{Y}}_{t,i}) \in \mathbb{R}^{p \times M_{t,i}} \times \mathbb{R}^{M_{t,i}}$ denotes the samples from previous task $i$ and we define $M_{t,i}$ as the number of samples in $\mathcal{M}_{t,i}$. In most CL applications, the memory space is fully utilized and the memory size does not change over time. We denote this memory size by $M$ that does not change with $t$. In this case, we have $\sum_{i=1}^{t-1} M_{t,i} = M$ for any $t \geq 2$. In this work, we focus on the situation in which the memory data are all fresh and have not been used in previous training. We equally allocate the memory to all previous tasks at each time $t$, i.e., $M_{t,i} = \frac{M}{t-1}$ for $i \in [t-1]$. For simplicity, we assume $\frac{M}{t-1}$ is an integer[1] for any $t \in \{2, 3, \cdots, T\}$.

**Performance metrics.** We first introduce the model error of parameter $\boldsymbol{w}$ over task $i$'s ground truth as:

$$\mathcal{L}_i(\boldsymbol{w}) = \|\boldsymbol{w} - \boldsymbol{w}_i^*\|^2. \tag{4}$$

The performance of CL is measured by two key metrics, which are forgetting and generalization error. To define these metrics, we let $\boldsymbol{w}_t$ be the parameters of the training result at task $t$.

1. *Forgetting:* It measures the average forgetting of old tasks after learning the new task. In our setup, forgetting at task $T$ w.r.t. previous tasks $[T-1]$ is defined as follows.

$$F_T = \frac{1}{T-1} \sum_{i=1}^{T-1} (\mathcal{L}_i(\boldsymbol{w}_t) - \mathcal{L}_i(\boldsymbol{w}_i)). \tag{5}$$

2. *Generalization error:* It measures the overall model generalization after the final task is learned. In our setup, generalization error is defined as follow.

$$G_T = \frac{1}{T} \sum_{i=1}^{T} \mathcal{L}_i(\boldsymbol{w}_T). \tag{6}$$

The definitions are consistent with the standard CL performance measures in experimental studies, e.g., (Saha et al., 2021).

## 4   A NOVEL SEQUENTIAL REPLAY VS. POPULAR CONCURRENT REPLAY

In this section, we first introduce the popular concurrent replay strategy that is widely used in current CL applications to mitigate catastrophic forgetting. We will then propose a novel sequential replay strategy, which may have appealing advantage compared to concurrent replay.

To describe these replay strategies, recall we denote $\boldsymbol{w}_t$ as the parameters of the training result at task $t$, which will be used as the initial point for the next task $t+1$ at each time $t+1$. The initial model parameter of task 1 is set to be $\boldsymbol{0}$, i.e., $\boldsymbol{w}_0 = \boldsymbol{0}$. The training loss for task $t$ is defined by mean-squared-error (MSE). We focus on the over-parameterized case, i.e., $p > n_t + M_t$. It is known that the convergence point of stochastic gradient descent (SGD) for MSE is the feasible point closest to the initial point with respect to the $\ell_2$-norm, i.e., the minimum-norm solution.

**Concurrent replay.** We first introduce the popular concurrent replay strategy as follows. At each task $t \geq 2$, we apply SGD on the current data set and the memory dataset jointly to update the

---

[1] We note that without the assumption of $\frac{M}{t-1} \in \mathbb{Z}$, memory can still be allocated as equally as possible, resulting in only a minor error. Our theoretical results remain of referential significance.

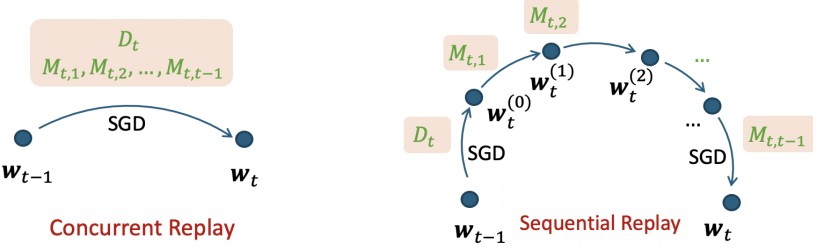

Figure 1: An illustration of concurrent replay and sequential replay.

model parameter. Specifically, as illustrated in Figure 1, at time $t$, we minimize the MSE loss via SGD on the combined dataset $\mathcal{D}_t \bigcup \mathcal{M}_t$ with the initial point $\boldsymbol{w}_{t-1}$ and obtain the convergent point $\boldsymbol{w}_t$, which can be written as

$$\boldsymbol{w}_t = \arg\min_{\boldsymbol{w}} \|\boldsymbol{w} - \boldsymbol{w}_{t-1}\|^2 \quad s.t. \ \ \boldsymbol{X}_t^\top \boldsymbol{w} = \boldsymbol{Y}_t, \ \ \widetilde{\boldsymbol{X}}_{t,i}^\top \boldsymbol{w} = \widetilde{\boldsymbol{Y}}_{t,i}, \ \ \text{for all} \ \ i \in [t-1]. \quad (7)$$

**Novel sequential replay.** In scenarios where previous tasks are very different from the current task, concurrent replay may result in contradicting gradient update directions, and can hurt the knowledge transfer among tasks. Consequently, concurrent replay may not always perform well. This motivate us to propose a replay strategy that sequentially replay history tasks one by one after training the current task, analogously to the way how a student reviews previously learned topics to avoid forgetting before exams.

To formally describe the training (see Figure 1 for an illustration), at each task $t \geq 2$, we first train on the current dataset $\mathcal{D}_t$ to learn the new task and converge to the initial stopping point $\boldsymbol{w}_t^{(0)}$. Then, for $i = 1, 2, ..., t-1$, we start from the previous stopping point $\boldsymbol{w}_t^{(i-1)}$ and train on the memory dataset $\mathcal{M}_{t,i}$ to converge to the next stopping point. Eventually, $w_t$ is obtained after revisiting all memory sets, i.e., $w_t = w_t^{(t-1)}$. We define $\widetilde{\boldsymbol{X}}_{t,0} \coloneqq \boldsymbol{X}_t, \widetilde{\boldsymbol{Y}}_{t,0} \coloneqq \boldsymbol{Y}_t$ and $\boldsymbol{w}_t^{(-1)} \coloneqq \boldsymbol{w}_{t-1}$. Then, the training process is equivalent to solve the following optimization problems recursively for $k = 0, 1, ..., t-1$:

$$\boldsymbol{w}_t^{(k)} = \arg\min_{\boldsymbol{w}} \left\|\boldsymbol{w} - \boldsymbol{w}_t^{(k-1)}\right\|^2 \quad s.t. \ \ \widetilde{\boldsymbol{X}}_{t,i}^\top \boldsymbol{w} = \widetilde{\boldsymbol{Y}}_{t,i}. \quad (8)$$

## 5 MAIN RESULTS

The main theoretical results in this work consist of two parts. First, we derive closed forms of the expected value of forgetting and generalization error for both concurrent and sequential replay methods. Second, based on those closed forms, we compare the performance of these two replay-based schemes, concluding that sequential replay outperforms concurrent replay when tasks are more dissimilar.

### 5.1 CHARACTERIZATION OF FORGETTING AND GENERALIZATION ERROR

In replay-based CL methods, the interference among tasks throughout the entire training process is highly intricate, primarily due to the presence of the memory dataset. This introduces an unavoidable challenge in understanding the impact of memory on the performance of replay-based methods. In the following theorem, we first present a common performance structure shared by both concurrent replay and sequential replay methods. The specific forms of the coefficients in the performance expressions will be provided later.

**Theorem 1.** *Under the problem setups considered in this work, the expected value of the forgetting and the generalization error at time $T \geq 2$ in both replay-based methods take the following forms.*

$$F_T = \frac{1}{T-1} \left[ \sum_{i=1}^{T-1} c_i \|\boldsymbol{w}_i^*\|^2 + \sum_{i=1}^{T-1} \sum_{j,k \leq T-1} c_{ijk} \|\boldsymbol{w}_j^* - \boldsymbol{w}_k^*\|^2 + \sum_{i=1}^{T-1} (noise_T(\sigma) - noise_i(\sigma)) \right],$$

$$G_T = \frac{1}{T}\left[d_{0T}\sum_{i=1}^{T}\|\boldsymbol{w}_i^*\|^2 + \sum_{i=1}^{T}\sum_{j,k\leq T}d_{ijkT}\|\boldsymbol{w}_j^* - \boldsymbol{w}_k^*\|^2\right] + noise_T(\sigma), \tag{9}$$

*where the coefficients are provided in Propositions 1 and 2, respectively, for concurrent and sequential replay methods.*

Theorem 1 indicates that since both concurrent and sequential methods are replay-based, they share the same high-level performance dependence on the system parameters. It can be seen that both of their forgetting and generalization error consist of the following three components. The first component exhibits the form of $C\|\boldsymbol{w}_i^*\|^2$ for some constant $C$. This component arises from the error associated with linear regression and is independent of the influence of other tasks. The second component captures the impact of task dissimilarities, representing the interference among different tasks during the training process. Extracting central information from this component is particularly useful for understanding how task dissimilarity affects the comparison between the two replay-based methods, which is the focus of Section 5.2. The third part captures the impact of the noise level.

In order to facilitate the comparison between the two replay-based methods, in the following two propositions, we provide the exact expressions for the coefficients in Theorem 1. We first provide the coefficients determining the generalization error as follows. To clarify, we note that the following proposition holds for all $t \in [T]$.

**Proposition 1.** *Under the problem setups considered in this work, the coefficients that express the expected value of generalization error $G_t$ take the following forms.*

$$d_{0t}^{(concurrent)} = r_0 r_M^{t-1}, \qquad\qquad d_{0t}^{(sequential)} = r_0 \Delta(t-1)$$

$$d_{ijkt}^{(concurrent)} = \begin{cases} (1-r_0)r_M^{t-j-1} + \sum_{l=0}^{t-j-1} r_M^l B_l \\ \quad + \sum_{l=0}^{t-2} \frac{pr_M^l B_l^2}{p-n-M-1} + \frac{r_M^{t-k}nB_l}{p-n-M-1} & if \quad j \in [t-1], k=i \\ (1-r_0) + \frac{r_M^{t-k}nB_l}{p-n-M-1} & if \quad j=t, k=i \\ \sum_{l=0}^{t-2} \frac{pr_M^l B_l^2}{p-n-M-1} & if \quad j<k \ and \ j,k \neq i,t \\ \frac{r_M^{t-k}nB_l}{p-n-M-1} & if \quad j<k \ and \ j,k \neq i \end{cases}$$

$$d_{ijkt}^{(sequential)} = \begin{cases} (1-r_0)\Delta(t-1) + \sum_{l=0}^{t-2}\Delta(l)(1-B_l)^{t-l-2}B_l & if \quad j=1, k=i \\ (1-r_0)(1-B_{t-j})^{j-1}\Delta(t-j) & if \quad j=2,3,...,t-1, \\ \quad + \sum_{l=0}^{t-j-1}\Delta(l)(1-B_l)^{t-l-2}B_l & and \ k=i \\ (1-r_0)(1-B_0)^{t-1} & if \quad j=t, k=i \end{cases}$$

$$noise_t^{(concurrent)}(\sigma) = r_0 r_M^{t-1}\Lambda(n,\sigma) + \sum_{l=0}^{t-2} r_M^l \Lambda(n+M,\sigma),$$

$$noise_t^{(sequential)}(\sigma) = \sum_{l=0}^{t-2}\Delta(l)\left[\sum_{l=1}^{t-1}(1-B_0)^{t-l-1}\Lambda(\frac{M}{t-1},\sigma) + (1-B_0)^{t-1}\Lambda(n,\sigma)\right].$$

*where* $r_a := \left(1 - \frac{n+a}{p}\right)$, $B_l := \frac{M}{(t-l-1)p}$, $\Delta(a) = \prod_{l=0}^{a-1}\left[(1-B_l)^{t-l-1}r_0\right]$, $\Lambda(a,\sigma) = \frac{a\sigma^2}{p-a-1}$.

By substituting $t = T$, we obtain the expressions of coefficients in Theorem 1. We provide the coefficients determining the forgetting in the following proposition.

**Proposition 2.** *Under the problem setups considered in this work, the coefficients that express the expected value of forgetting in Theorem 1 take the following forms:*

$$c_i = d_{0T} - d_{0i} \quad and \quad c_{ijk} = d_{ijkT} - d_{ijki},$$

*where $d_{0t}$ and $d_{ijkt}$ are defined in Proposition 1.*

The above two propositions will be useful in Section 5.2 to compare between concurrent and sequential replay methods. Here, we first draw some basic insights from these expressions. (i) It is straightforward to verify that by letting $M = 0$, both training methods yield the same result, which is consistent with the memoryless case shown by Lin et al. (2023). (ii) We can also observe that low

task similarity negatively impacts model generalization, as $d_{ijkT}$ are non-negative. (iii) We observe that the expected value of both forgetting and generalization error approach to 0 when $p \to \infty$. This implies that a model with substantial capacity (i.e., when $p$ is sufficiently large) will facilitate effective learning for each task, which can also alleviate the negative impact of task dissimilarity.

## 5.2 COMPARISON BETWEEN CONCURRENT REPLAY AND SEQUENTIAL REPLAY

The main challenge to compare the performance between the two replay-based methods lies in the complexity of the second term, which captures how the task similarity as well as memory data affect the performance. Here the task similarity is characterized by the distance between the true parameters for two tasks. In this section, we will first study a simple case with two tasks, i.e., when $T = 2$, to build our intuition, and then extend to the case with general $T$ based on the central insight obtained in the simple case.

**Two-task Case** ($T = 2$): Following Theorem 1, the performance of both replay methods shares the following common form:

$$F_2 = \hat{c}_1 \|\boldsymbol{w}_1^*\|^2 + \hat{c}_2 \|\boldsymbol{w}_1^* - \boldsymbol{w}_2^*\|^2 + \text{noise}_2(\sigma) - \text{noise}_1(\sigma),$$

$$G_2 = \frac{1}{2}\hat{d}_1(\|\boldsymbol{w}_1^*\|^2 + \|\boldsymbol{w}_2^*\|^2) + \frac{1}{2}\hat{d}_2 \|\boldsymbol{w}_1^* - \boldsymbol{w}_2^*\|^2 + \text{noise}_2(\sigma),$$

where $\hat{c}_1, \hat{c}_2, \hat{d}_1, \hat{d}_2$ are some constants. The specific forms of the coefficients in the above equation are provided in Appendix C. We take the forgetting as an example to analyze the comparison between the two methods. Based on the expressions, it can be observed that $\hat{c}_1^{(\text{concurrent})} < \hat{c}_1^{(\text{sequential})}$ and $\hat{c}_2^{(\text{concurrent})} > \hat{c}_2^{(\text{sequential})}$. Thus, at the high level, the task dissimilarity is sufficiently large (i.e., tasks are very different), then $c_2$ will dominant the forgetting performance, and hence sequential replay will have less forgetting than concurrent replay (because $\hat{c}_2^{(\text{concurrent})} > \hat{c}_2^{(\text{sequential})}$). Alternatively, if the tasks are very similar and the noise is small, then $c_1$ will dominate the performance, and concurrent replay will yield less forgetting. Similar observations can be made for the generalization error by noting that $\hat{d}_1^{(\text{concurrent})} < \hat{d}_1^{(\text{sequential})}$ and $\hat{d}_2^{(\text{concurrent})} > \hat{d}_2^{(\text{sequential})}$. The following theorem formally establishes our high-level observations.

**Theorem 2.** *Under the problems setups considered in the work, under the positive constants $\xi_1, \xi_2, \mu_1, \mu_2$ with detailed forms given in Appendix C, we have*

$$F_2^{(concurrent)} > F_2^{(sequential)} \quad \text{if and only if} \quad \xi_1 \|\boldsymbol{w}_1^* - \boldsymbol{w}_2^*\|^2 + \xi_2\sigma^2 > \|\boldsymbol{w}_1^*\|^2,$$

$$G_2^{(concurrent)} > G_2^{(sequential)} \quad \text{if and only if} \quad \mu_1 \|\boldsymbol{w}_1^* - \boldsymbol{w}_2^*\|^2 + \mu_2\sigma^2 > \|\boldsymbol{w}_1^*\|^2.$$

Theorem 2 provably establishes an intriguing fact that the widely used concurrent replay may not always perform better, and sequential replay can perform better when tasks are more different from each other. We further elaborate our comparison between the two methods for the case with $T = 2$ in Appendix C (where the impact of noise is also considered) and with $T = 3$ in Appendix D). The insights obtained from Theorem 2 can also be extended to the general case as follows.

**General Case** ($T \geq 2$): Comparing the performance in two replay methods provided in Theorem 1 under general $T$ is significantly more challenging, because the mathematical expression of the coefficients become highly complex. However, our insights obtained from the two-task case can still be useful, i.e., sequential replay tends to performance better when tasks are very different. To see this, we consider the expected value of the forgetting and the generalization error on an individual prior task $i$, which is $\mathbb{E}[\mathcal{L}_i(\boldsymbol{w}_t)] - \mathbb{E}[\mathcal{L}_i(\boldsymbol{w}_i)]$ and $\mathbb{E}[\mathcal{L}_i(\boldsymbol{w}_t)]$ respectively. We observe the facts similar to the case with $T = 2$. Specifically, it can be shown that the coefficients presented in Theorem 1 satisfy $c_{ijk}^{(\text{concurrent})} > c_{ijk}^{(\text{sequential})}$ and $d_{ijkT}^{(\text{concurrent})} > d_{ijkT}^{(\text{sequential})}$, whereas $c_i^{(\text{concurrent})} < c_i^{(\text{sequential})}$ and $d_{0T}^{(\text{concurrent})} < d_{0T}^{(\text{sequential})}$ for general $T$ under certain conditions. These observations suggest that if the tasks are all very different from each other, then sequential replay will have smaller forgetting and generalization error than concurrent replay because $c_{ijk}^{(\text{concurrent})} > c_{ijk}^{(\text{sequential})}$ and $d_{ijkT}^{(\text{concurrent})} > d_{ijkT}^{(\text{sequential})}$ will dominate the comparison. While it is challenging to provide an exact closed-form characterization of the conditions under which sequential replay outperforms concurrent replay, the following theorem presents an example setting where sequential replay outperforms concurrent replay, based on the understanding outlined above.

**Theorem 3.** *Under the problem setups in this work, suppose the ground truth $\boldsymbol{w}_i^*$ is orthonormal to each other for $i \in [T]$, $M \geq 2$, and $p = \mathcal{O}(T^4 n^2 M^2)$. Then we have:*

$$F_T^{(concurrent)} > F_T^{(sequential)} \quad and \quad G_T^{(concurrent)} > G_T^{(sequential)}.$$

In Theorem 3, orthonormal $w_i^*$ is an extreme case to have very different tasks. Typically, since the forgetting and generalization error are continuous functions of the task dissimilarity, we expect that in the regime that the tasks are highly different, sequential replay will still be advantageous to enjoy less forgetting and smaller generalization error, and such an advantage should be more apparent as tasks become more dissimilar. To explain this, we consider the generalization error as an example. Assuming that the norm of ground truth is fixed, a higher level of task dissimilarities exacerbates the generalization error since each coefficient $d_{ijkT}$ is positive for both training methods. However, a weaker dependence on task similarities indicates that the generalization error of sequential replay grows slower than concurrent replay as tasks become more dissimilar, resulting advantage for sequential replay to enjoy smaller generalization error. A similar reason is applicable to the forgetting performance, although it is important to note that $c_{ijk}$ is not always positive. These facts are further verified by our numerical simulation in Section 6.1.

**Remark.** It is clear that the order in which old tasks are replayed after current task learning is very important under the framework of sequential replay, which affects both forgetting and generalization errors. Needless to say, the sequential order considered in this work, where tasks are reviewed from the oldest to the newest, is not necessarily the optimal strategy for sequential replay, where however has already demonstrated exciting advantages over concurrent replay. How to design an effective replay order to achieve better performance is a very interesting yet challenging future direction.

# 6 EXPERIMENTAL STUDIES AND IMPLICATIONS ON PRACTICAL CL

In this section, we first conduct experiments on linear models to verify our theoretical results. Next, and also more interestingly, we show that our theoretical results can guide the algorithm design of CL in practice, where a novel replay-based CL algorithm is proposed and evaluated with DNNs.

## 6.1 SIMULATION ON LINEAR REGRESSION MODELS

Following our theoretical investigation, we consider the CL setup where each task is a linear regression problem, and set $T = 5$, $p = 500$, $n = 24$, $\sigma = 0$, $M = 24$. We construct several sets of ground truth on the unit sphere defined by $||\boldsymbol{w}_j^*||^2 = 1$, with consistent task similarity, i.e., $||\boldsymbol{w}_j^* - \boldsymbol{w}_i^*||^2$ is constant and same for any two tasks with $j \neq i$. The comparisons between theoretical results and simulation results are shown in Figure 2 in terms of both forgetting and generalization error. Here the theoretical results are calculated using eqs. (33) to (36). For the simulation results, we evaluate the forgetting and generalization error based on the solutions after solving each task, and calculate the empirical expectation over $10^3$ iterations.

Several important insights can be immediately obtained from Figure 2: 1) Our theoretical results exactly match with our simulation results, which can clearly corroborate the correctness of our theory. 2) When tasks are similar, i.e., the task gap $||\boldsymbol{w}_j^* - \boldsymbol{w}_i^*||^2$ is small than some threshold, concurrent replay is better than sequential replay. However, when tasks become dissimilar, sequential replay starts to outperform concurrent replay in terms of both forgetting and generalization error. And the advantage of sequential replay becomes more significant as the task gap increases, which also aligns with our theoretical results.

## 6.2 A NEW ALGORITHM DESIGN FOR CL IN PRACTICE

Our theoretical results not only rigorously characterize replay-based CL in overparameterized linear models, but also shed light on the algorithm design for practical CL with real datasets and DNNs. As our theory suggests that sequential replay can benefit CL more than concurrent replay when tasks are dissimilar, an interesting idea and a potential way to improve the performance is to merge sequential replay into replayed-based CL with concurrent replay. Thus inspired, we propose a novel hybrid replay framework, which adapts between concurrent replay and sequential replay for each task based on its similarity with old tasks in the memory. More specifically, before learning a new

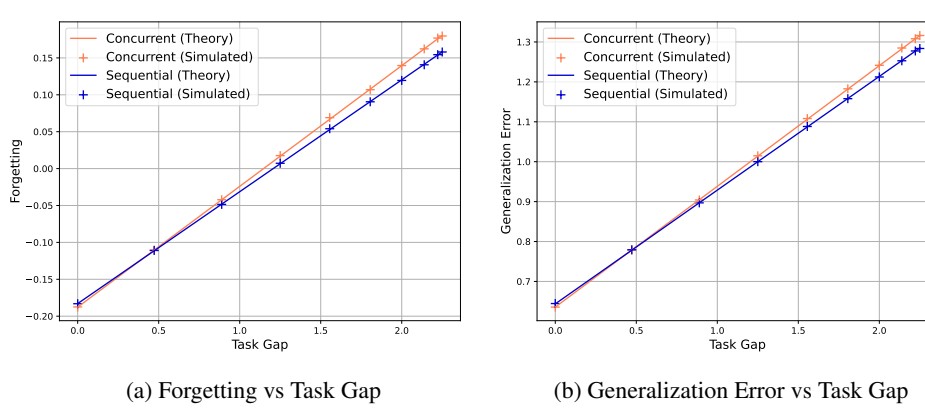

(a) Forgetting vs Task Gap  (b) Generalization Error vs Task Gap

Figure 2: Forgetting and Generalization Error vs Task Gap

task $\mathcal{T}_t$, we first characterize its similarity with old tasks in the memory, and divide the old tasks into two sets, i.e., $\mathcal{M}_{sim}^t$ that includes old tasks similar to $\mathcal{T}_t$, and $\mathcal{M}_{dis}^t$ containing the remaining old tasks which are deemed as dissimilar tasks to $\mathcal{T}_t$. To learn $\mathcal{T}_t$, we first apply concurrent replay to train the model jointly with the data of $\mathcal{T}_t$ and the replay data of old tasks in $\mathcal{M}_{sim}^t$, and then use sequential replay to sequentially finetune the learned model using the replay data for each old task in $\mathcal{M}_{dis}^t$. The general procedure is described in Algorithm 1.

---

**Algorithm 1** *Hybrid Replay* Training Framework

---

**Require:** Training data set $\mathcal{D}$
1: **procedure** TRAIN($\mathcal{D}$)
2:     $\mathcal{M} \leftarrow \{\}$                                                             ▷ Initialize empty replay buffer
3:     $\theta \leftarrow$ Initialize DNN model parameters
4:     **for** task $\mathcal{T}_t = \mathcal{T}_0, \ldots$ **do**
5:         **if** $\mathcal{M} \neq \emptyset$ **then**                                    ▷ If replay buffer is non-empty
6:             $\mathcal{M}_{sim}, \mathcal{M}_{dis} \leftarrow$ DIVIDEBUFFER($M, \mathcal{D}_t$)
7:         **end if**
8:         $\theta \leftarrow$ CONCURRENTTRAIN($\mathcal{D}_t \cup \mathcal{M}_{sim}$)       ▷ Train the new task data and similar exemplars concurrently
9:         **for** $\mathcal{M}_i \in \mathcal{M}_{dis}$ **do**
10:           $\theta \leftarrow$ SEQUENTIALTRAIN($\mathcal{M}_i$)        ▷ Train exemplars from dissimilar tasks sequentially
11:         **end for**
12:         $\mathcal{M} \leftarrow \mathcal{M} \cup \mathcal{M}_t$         ▷ Update replay buffer with new exemplars $\mathcal{M}_t \sim \mathcal{D}_t$
13:     **end for**
14: **end procedure**

---

To verify the performance of the proposed hybrid replay framework, we consider a task-incremental CL setup using the real-world dataset CIFAR-100 (Krizhevsky et al., 2009). where each task a multi-class classification problem. Following recent work (Van de Ven et al., 2022), we randomly split the CIFAR-100 dataset into ten tasks $\{\mathcal{T}_0, \ldots, \mathcal{T}_9\}$, each containing ten distinct classes, later referred as Split-CIFAR-100. The objective for each task $\mathcal{T}_t$ is to classify between its ten classes $\{\mathcal{Y}_{t,0}, \ldots, \mathcal{Y}_{t,9}\}$ with the task label $t$ explicitly provided during training and testing. We use ResNet18 as our base model to learn each task sequentially, where each task has a unique classification layer. It is clear that how to determine the task similarity is critical for implementing the hybrid replay. Since the similarity pattern is not clear and complex among the real-life images in Split-CIFAR-100, we manually control the task similarity in a heuristic manner by introducing image corruption into the tasks. In particular, to understand the benefit of the hybrid replay in a clean manner, we consider the following specific training comparison between two schemes: 1) Concurrent replay is applied on all ten tasks; 2) Hybrid replay is applied on task $\mathcal{T}_5$, while concurrent replay is applied on the remaining tasks. In this way, concurrent replay on tasks $\mathcal{T}_t, t \in \{0, 1, 2, 3, 4\}$ can

Table 1: Accuracy ($ACC$, the larger the better) and Backward Transfer ($BWT$, the larger the better) of different training methods (concurrent replay vs. hybrid replay) on CIFAR-100 with varying number of corrupted tasks. "1 Corruption", for example, indicates that data corruption was applied to 1 out of 10 tasks, making it more dissimilar than others. "Improvement" shows the $ACC$ overhead that Hybrid Replay achieves over Concurrent Replay under the same setup. All results are averaged over 10 independent runs.

| Setting | Original Dataset | | 1 Corruption | | 2 Corruption | | 3 Corruption | |
|---|---|---|---|---|---|---|---|---|
| Metric | $ACC$ | $BWT$ | $ACC$ | $BWT$ | $ACC$ | $BWT$ | $ACC$ | $BWT$ |
| *Concurrent Replay* | 69.206 | -6.738 | 64.244 | -7.760 | 60.667 | -9.275 | 58.933 | -8.572 |
| *Hybrid Replay* | **69.568** | **-6.574** | **64.807** | **-7.233** | **61.304** | **-8.752** | **59.720** | **-8.352** |
| Improvement | +0.362 | +0.164 | +0.563 | +0.527 | +0.637 | +0.523 | +0.787 | +0.220 |

be thought as a warm-up training strategy for both schemes. For tasks $\mathcal{T}_t, t \in \{6, 7, 8, 9\}$, concurrent replay is applied to isolate the effect of hybrid replay on Task $\mathcal{T}_5$ and for simplicity. More training details and specifications for image corruption are listed in Appendix G.

To evaluate the performance, following the standard in practical CL and also being consistent with our theoretical investigation, we consider both average accuracy and forgetting. More specifically, the model's average *Accuracy* across all seen tasks is denoted $ACC$, which captures the generalization error. The forgetting, or backward transfer, is defined as $BWT = \frac{2}{T(T-1)} \sum_{k=2}^{T} \sum_{t=1}^{k-1} (a_{k,t} - a_{t,t})$ (Lesort et al., 2020) where $a_{k,t}$ represents the testing accuracy on task $t$ after training task $k$.

As shown in Table 1, hybrid replay outperforms concurrent replay on Split-CIFAR-100 (i.e., Original Dataset), in terms of both average accuracy and forgetting. Moreover, we control the similarity by using the number of corrupted tasks (i.e., task with corrupted images) in the task sequence. In particular, we consider three different scenarios, '1 Corruption' with 1 corrupted task, '2 Corruption' with 2 corrupted tasks, and '3 Corruption' with 3 corrupted tasks. Intuitively, the tasks are more dissimilar when more tasks are corrupted. It can be seen from Table 1 that hybrid replay consistently outperforms concurrent replay, and more importantly, the performance improvement becomes more significant as tasks are more dissimilar. These results further justify the correctness and usefulness of our theoretical results. It is worth to note that the performance of hybrid replay has not been optimized in terms of the replay order and selection of similar tasks, which may further improve the effectiveness of sequential replay. This encouraging result highlights the great potentials of exploiting sequential replay in improving the performance of replay-based CL.

## 7 CONCLUSION

In this work, we took a closer look at the replay strategy in replay-based CL and questioned the effectiveness of the widely used training technique, i.e., concurrent replay, as inspired by human learning. In particular, we proposed a novel replay strategy, namely sequential replay, which replays old tasks in the memory sequentially after current task learning. By leveraging overparameterized linear models with equal memory allocation, we provided the first explicit expressions of the expected value of both forgetting and generalization errors under two replay methods, concurrent replay and sequential replay. Comparisons between their theoretical performance led to the insight that sequential replay outperforms concurrent replay in terms of forgetting and generalization error when the tasks are less similar, which is consistent with our motivations from human learning and multitask learning. Our simulation results on linear models further corroborated the correctness of our theoretical results. More importantly, based on our theory, we proposed a novel hybrid replay framework for practical CL and experiments on CIFAR100 with DNNs verified the superior performance of this framework over concurrent replay. To the best of our knowledge, our work provides the first comprehensive theoretical study on replay for replay-based CL, which will hopefully motivate more principled designs for better replay-based CL.

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

# Supplementary Materials

## A    SUPPORTING LEMMAS

Recall that $P_{\boldsymbol{X}} = \boldsymbol{X}(\boldsymbol{X}^\top \boldsymbol{X})^{-1} \boldsymbol{X}^\top$ and $\boldsymbol{X}^\dagger = \boldsymbol{X}(\boldsymbol{X}^\top \boldsymbol{X})^{-1}$. We first provide some useful lemmas for the derivation of forgetting and generalization error. In the following lemma, we provide the expression of the SGD convergence point when training on a single task.

**Lemma 1.** *Suppose $\boldsymbol{X} \in \mathbb{R}^{p \times n}$ and $\boldsymbol{Y} \in \mathbb{R}^n$, where $\boldsymbol{Y} = \boldsymbol{X}^\top \boldsymbol{w}^* + \boldsymbol{z}$. Consider the optimization problem:*

$$\boldsymbol{w}_{out} = \arg\min_{\boldsymbol{w}} \|\boldsymbol{w} - \boldsymbol{w}_{in}\|_2^2$$
$$s.t. \ \boldsymbol{X}^\top \boldsymbol{w} = \boldsymbol{Y}.$$

*The solution of the above problem can be written as:*

$$\boldsymbol{w}_{out} = \boldsymbol{w}_{in} + \boldsymbol{X}^\dagger (\boldsymbol{Y} - \boldsymbol{X}^\top \boldsymbol{w}_{in}),$$

*or equivalently,*

$$\boldsymbol{w}_{out} = (\boldsymbol{I} - P_{\boldsymbol{X}})\boldsymbol{w}_{in} + P_{\boldsymbol{X}}\boldsymbol{w}^* + \boldsymbol{X}^\dagger \boldsymbol{z}.$$

*Proof.* The proof follows from Lemma B.1 in Lin et al. (2023). $\qquad\square$

**Lemma 2.** *Suppose each element of the random matrix $\boldsymbol{X} \in \mathbb{R}^{p \times n}$ follows from the standard distribution $\mathcal{N}(0,1)$ independently and $v \in \mathbb{R}^p$ is a vector, then we have:*

$$\mathbb{E} \|P_{\boldsymbol{X}} v\|^2 = \frac{n}{p} \|v\|^2.$$

*Proof.* The detailed proof refers to Proposition 3 in Ju et al. (2023). $\qquad\square$

**Lemma 3.** *Suppose each element of the random matrix $\boldsymbol{X} \in \mathbb{R}^{p \times n}$ follows from the standard distribution $\mathcal{N}(0,1)$ independently. Also, $z \in \mathbb{R}^n$ is a vector and it follows from $\mathcal{N}(0,\sigma^2 \boldsymbol{I}_n)$ independently. Then, we have:*

$$\mathbb{E} \|\boldsymbol{X}^\dagger \boldsymbol{z}\|^2 = \frac{n\sigma^2}{p-n-1}.$$

*Proof.* The proof follows Lemma B.2 in Lin et al. (2023). We apply the "trace trick" to have:

$$\begin{aligned}
\mathbb{E} \|\boldsymbol{X}^\dagger \boldsymbol{z}\|^2 &= \mathbb{E} \left[ \boldsymbol{z}^\top \left( \boldsymbol{X}^\top \boldsymbol{X} \right)^{-1} \boldsymbol{z} \right] \\
&= \mathbb{E} \left[ \mathrm{tr} \left[ \left( \boldsymbol{X}^\top \boldsymbol{X} \right)^{-1} \boldsymbol{z}\boldsymbol{z}^\top \right] \right] \\
&\overset{(i)}{=} \mathrm{tr} \left[ \mathbb{E} \left[ \left( \boldsymbol{X}^\top \boldsymbol{X} \right)^{-1} \right] \mathbb{E} \left[ \boldsymbol{z}\boldsymbol{z}^\top \right] \right] \\
&\overset{(ii)}{=} \sigma^2 \mathrm{tr} \left[ \mathbb{E} \left[ \left( \boldsymbol{X}^\top \boldsymbol{X} \right)^{-1} \right] \right] \\
&\overset{(iii)}{=} \frac{n\sigma^2}{p-n-1},
\end{aligned}$$

where $(i)$ follows from the independence between $\boldsymbol{X}$ and $\boldsymbol{z}$, $(ii)$ follows from the fact that $\mathbb{E} \left[ \boldsymbol{z}\boldsymbol{z}^\top \right] = \sigma^2 \boldsymbol{I}_n$ and $(iii)$ follows from the fact that $\left( \boldsymbol{X}^\top \boldsymbol{X} \right)^{-1} \sim \mathcal{W}^{-1}(\boldsymbol{I}_n, p)$. $\qquad\square$

**Lemma 4.** *For any vector $\boldsymbol{v}_1, \boldsymbol{v}_2 \in \mathbb{R}^p$, we have:*

$$\langle (\boldsymbol{I} - P_{\boldsymbol{X}})\boldsymbol{v}_1, \boldsymbol{X}^\dagger \boldsymbol{v}_2 \rangle = 0,$$
$$\langle (\boldsymbol{I} - P_{\boldsymbol{X}})\boldsymbol{v}_1, P_{\boldsymbol{X}} \boldsymbol{v}_2 \rangle = 0.$$

*Proof.* The proof follows from the definition of $P_{\boldsymbol{X}}$ and $\boldsymbol{X}^\dagger$ straightforward. $\qquad\square$

Now, we provide useful lemmas in proving the expected model value of model errors in the concurrent replay method.

**Lemma 5.** *Suppose $P \in \mathbb{R}^{p \times p}$ is a projection matrix and $\boldsymbol{v} \in \mathbb{R}^p$ is a random vector with i.i.d. standard Gaussian elements, then $P\boldsymbol{v}$ and $(\boldsymbol{I} - P)\boldsymbol{v}$ are independent. Moreover, if $\boldsymbol{V} \in \mathbb{R}^{p \times m}$ is a random matrix with i.i.d. standard Gaussian elements, then we have $P\boldsymbol{V}$ and $(\boldsymbol{I} - P)\boldsymbol{V}$ are independent*

*Proof.* We prove the vector case in two steps. First, we prove that $P\boldsymbol{v}$ and $(\boldsymbol{I} - P)\boldsymbol{v}$ are jointly Gaussian. Next, we prove that they are uncorrelated. By combining these two facts, we can conclude that $P\boldsymbol{v}$ and $(\boldsymbol{I} - P)\boldsymbol{v}$ are independent. To prove $P\boldsymbol{v}$ and $(\boldsymbol{I} - P)\boldsymbol{v}$ are jointly Gaussian, we concatenate them to form a random vector $\boldsymbol{z} = \begin{bmatrix} P\boldsymbol{v} \\ (\boldsymbol{I} - P)\boldsymbol{v} \end{bmatrix}$. For any $\boldsymbol{w} = \begin{bmatrix} \boldsymbol{w}_1 \\ \boldsymbol{w}_2 \end{bmatrix}$, where $\boldsymbol{w}_1, \boldsymbol{w}_2 \in \mathbb{R}^p$, we can see that the linear combination of its elements $\boldsymbol{w}^\top \boldsymbol{z} = (\boldsymbol{w}_1^\top P + \boldsymbol{w}_2^\top (\boldsymbol{I} - P))\boldsymbol{v}$ is still Gaussian. To prove they are uncorrelated, we have:

$$
\begin{aligned}
\mathrm{Cov}(P\boldsymbol{v}, (\boldsymbol{I} - P)\boldsymbol{v}) &= \mathbb{E}\left[ P\boldsymbol{v}((\boldsymbol{I} - P)\boldsymbol{v})^\top \right] \\
&= P\mathbb{E}(\boldsymbol{v}\boldsymbol{v}^\top)(\boldsymbol{I} - P) \\
&\overset{(i)}{=} P(\boldsymbol{I} - P) \\
&= 0,
\end{aligned}
$$

where $(i)$ follows from the fact that $\boldsymbol{v}$ has i.i.d. standard Gaussian elements. Now, for the matrix case, we can equivalently consider the vector $\hat{\boldsymbol{v}} \in \mathbb{R}^{pm}$ which is formed by concatenating all the columns of $\boldsymbol{V}$ and the projection matrix $\hat{P} = \mathrm{diag}([P, P, .., P]) \in \mathbb{R}^{pm \times pm}$. $\qquad \square$

**Lemma 6.** *Suppose $\boldsymbol{X} \in \mathbb{R}^{p \times n}$ is a random matrix with i.i.d. standard Gaussian elements and $v \in \mathbb{R}^p$ is a fixed vector, then we have:*

$$
\mathbb{E}\left[ \boldsymbol{X}^\top v v^\top \boldsymbol{X} \right] = \|v\|^2 \cdot \boldsymbol{I}.
$$

*Proof.* To clarify, we denote $\boldsymbol{X} = [\boldsymbol{x}_1, ..., \boldsymbol{x}_n]$, where $\boldsymbol{x}_i$ is the $i^{th}$ column of $\boldsymbol{X}$. We also denote $[\cdot]_{i,j}$ as the element of $i^{th}$ row and $j^{th}$ column of a matrix. Then we have:

$$
\left[ \mathbb{E}\left[ \boldsymbol{X}^\top \boldsymbol{v}\boldsymbol{v}^\top \boldsymbol{X} \right] \right]_{i,j} = \mathrm{cov}(\boldsymbol{v}^\top \boldsymbol{x}_i, \boldsymbol{v}^\top \boldsymbol{x}_j) = \begin{cases} 0 & \text{if } i \neq j, \\ \|v\|^2 & \text{if } i = j. \end{cases}
$$

$\qquad \square$

**Lemma 7.** *Suppose $\boldsymbol{X} \in \mathbb{R}^{p \times n}$ is a random matrix with i.i.d. standard Gaussian elements and $P \in \mathbb{R}^{p \times p}$ is any projection matrix from $p$-dimension to $d$-dimension, then we have:*

$$
\mathrm{tr}\left( \mathbb{E}\left[ \left( \boldsymbol{X}^\top (\boldsymbol{I} - P)\boldsymbol{X} \right)^{-1} \right] \right) = \frac{n}{p - d - n - 1}.
$$

*Proof.* We first note that $(\boldsymbol{I} - P)$ is a projection matrix with $p - d$ many eigenvalues 1 and $d$ many eigenvalues 0. With loss of generalization, we write $(\boldsymbol{I} - P) = U^\top \Sigma U$ where $\Sigma = \mathrm{diag}([1, 1, ..., 1, 0, ..., 0])$ is a diagonal matrix, whose first $p - d$ elements are 1 while others are 0, and $U$ is an orthogonal matrix. Also, we denote $\hat{\boldsymbol{X}} \in \mathbb{R}^{(p-d) \times n}$ as the first $p - d$ rows of $\boldsymbol{X}$.

$$
\begin{aligned}
\mathrm{tr}\left( \mathbb{E}\left[ \left( \boldsymbol{X}^\top (\boldsymbol{I} - P)\boldsymbol{X} \right)^{-1} \right] \right) &= \mathrm{tr}\left( \mathbb{E}\left[ \left( \boldsymbol{X}^\top U^\top \Sigma U \boldsymbol{X} \right)^{-1} \right] \right) \\
&\overset{(i)}{=} \mathrm{tr}\left( \mathbb{E}\left[ \left( \boldsymbol{X}^\top \Sigma \boldsymbol{X} \right)^{-1} \right] \right) \\
&= \mathrm{tr}\left( \mathbb{E}\left[ \left( \hat{\boldsymbol{X}}^\top \hat{\boldsymbol{X}} \right)^{-1} \right] \right) \\
&\overset{(ii)}{=} \frac{n}{p - d - n - 1}
\end{aligned}
$$

where $(i)$ follows from the rotational symmetry of standard Gaussian distribution, $(ii)$ follows from the fact that $\left( \hat{\boldsymbol{X}}^\top \hat{\boldsymbol{X}} \right)^{-1} \sim \mathcal{W}^{-1}(\boldsymbol{I}_n, p - d)$. $\qquad \square$

**Lemma 8.** *Suppose $\boldsymbol{V} = [\boldsymbol{X}_1, \boldsymbol{X}_2]$ where $\boldsymbol{X}_1 \in \mathbb{R}^{p \times n_1}$, $\boldsymbol{X}_2 \in \mathbb{R}^{p \times n_2}$ are two random matrices with i.i.d. standard Gaussian elements and $\boldsymbol{v} \in \mathbb{R}^p$ is a fixed vector. Then we have:*

$$
\mathbb{E} \left\| \boldsymbol{V}^\dagger \begin{bmatrix} \boldsymbol{X}_1^\top \\ \boldsymbol{0} \end{bmatrix} \boldsymbol{v} \right\|^2 = \frac{n_1}{p} \cdot \left( 1 + \frac{n_2}{p - n_1 - n_2 - 1} \right) \|\boldsymbol{v}\|^2
$$

*Proof.* we consider the block expression of matrix $(\boldsymbol{V}_2^\top \boldsymbol{V}_2)^{-1}$. First, we have:

$$
\boldsymbol{V}^\top \boldsymbol{V} = \begin{bmatrix} \boldsymbol{X}_1^\top \\ \boldsymbol{X}_2^\top \end{bmatrix} [\boldsymbol{X}_1 \quad \boldsymbol{X}_2] = \begin{bmatrix} \boldsymbol{X}_1^\top \boldsymbol{X}_1 & \boldsymbol{X}_1^\top \boldsymbol{X}_2 \\ \boldsymbol{X}_2^\top \boldsymbol{X}_1 & \boldsymbol{X}_2^\top \boldsymbol{X}_2 \end{bmatrix}.
$$

Now, we partition the matrix $(\boldsymbol{V}^\top \boldsymbol{V})^{-1}$ into four blocks:

$$
(\boldsymbol{V}_2^\top \boldsymbol{V}_2)^{-1} = \begin{bmatrix} A_{1,1} & A_{1,2} \\ A_{2,1} & A_{2,2} \end{bmatrix},
$$

where

$$
A_{1,1} = (\boldsymbol{X}_1^\top \boldsymbol{X}_1)^{-1} - (\boldsymbol{X}_1^\top \boldsymbol{X}_1)^{-1} \boldsymbol{X}_1^\top \boldsymbol{X}_2 \left( \boldsymbol{X}_2^\top \boldsymbol{X}_2 - \boldsymbol{X}_2^\top \boldsymbol{X}_1 (\boldsymbol{X}_1^\top \boldsymbol{X}_1)^{-1} \boldsymbol{X}_1^\top \boldsymbol{X}_2 \right)^{-1} \boldsymbol{X}_2^\top \boldsymbol{X}_1 (\boldsymbol{X}_1^\top \boldsymbol{X}_1)^{-1}
$$

$$
= P_{\boldsymbol{X}_1} + P_{\boldsymbol{X}_1} \boldsymbol{X}_2 \left( \boldsymbol{X}_2^\top (\boldsymbol{I} - P_{\boldsymbol{X}_1}) \boldsymbol{X}_2 \right)^{-1} \boldsymbol{X}_2^\top P_{\boldsymbol{X}_1}.
$$

Therefore, we have

$$
\mathbb{E} \left\| \boldsymbol{V}^\dagger \begin{bmatrix} \boldsymbol{X}_1^\top \\ \boldsymbol{0} \end{bmatrix} \boldsymbol{v} \right\|^2 = \mathbb{E} \left[ \boldsymbol{v}^\top \left[ P_{\boldsymbol{X}_1} + P_{\boldsymbol{X}_1} \boldsymbol{X}_2 \left( \boldsymbol{X}_2^\top (\boldsymbol{I} - P_{\boldsymbol{X}_1}) \boldsymbol{X}_2 \right)^{-1} \boldsymbol{X}_2^\top P_{\boldsymbol{X}_1} \right] \boldsymbol{v} \right]
$$

$$
\overset{(i)}{=} \frac{n_1}{p} \|\boldsymbol{v}\|^2 + \mathbb{E} \left[ \boldsymbol{v}^\top \left[ P_{\boldsymbol{X}_1} \boldsymbol{X}_2 \left( \boldsymbol{X}_2^\top (\boldsymbol{I} - P_{\boldsymbol{X}_1}) \boldsymbol{X}_2 \right)^{-1} \boldsymbol{X}_2^\top P_{\boldsymbol{X}_1} \right] \boldsymbol{v} \right], \quad (10)
$$

where $(i)$ follows from Lemma 2. Now, we consider

$$
\mathbb{E} \left[ \boldsymbol{v}^\top \left[ P_{\boldsymbol{X}_1} \boldsymbol{X}_2 \left( \boldsymbol{X}_2^\top (\boldsymbol{I} - P_{\boldsymbol{X}_1}) \boldsymbol{X}_2 \right)^{-1} \boldsymbol{X}_2^\top P_{\boldsymbol{X}_1} \right] \boldsymbol{v} \right]
$$

$$
= \mathbb{E} \left[ \text{tr} \left( \boldsymbol{X}_2^\top P_{\boldsymbol{X}_1} \boldsymbol{v} \boldsymbol{v}^\top P_{\boldsymbol{X}_1} \boldsymbol{X}_2 \left( \boldsymbol{X}_2^\top (\boldsymbol{I} - P_{\boldsymbol{X}_1}) \boldsymbol{X}_2 \right)^{-1} \right) \right]
$$

$$
\overset{(i)}{=} \mathbb{E}_{X_1} \left[ \text{tr} \left( \mathbb{E}_{X_2} \left[ \boldsymbol{X}_2^\top P_{\boldsymbol{X}_1} \boldsymbol{v} \boldsymbol{v}^\top P_{\boldsymbol{X}_1} \boldsymbol{X}_2 \right] \cdot \mathbb{E}_{X_2} \left[ \left( \boldsymbol{X}_2^\top (\boldsymbol{I} - P_{\boldsymbol{X}_1}) \boldsymbol{X}_2 \right)^{-1} \right] \right) \right]
$$

$$
\overset{(ii)}{=} \mathbb{E}_{X_1} \left[ \text{tr} \left( \|P_{\boldsymbol{X}_1} \boldsymbol{v}\|^2 \cdot \boldsymbol{I} \cdot \mathbb{E}_{X_2} \left[ \left( \widetilde{\boldsymbol{X}}_2^\top (\boldsymbol{I} - P_{\boldsymbol{X}_1}) \widetilde{\boldsymbol{X}}_2 \right)^{-1} \right] \right) \right]
$$

$$
= \mathbb{E}_{X_1} \left[ \|P_{\boldsymbol{X}_1} \boldsymbol{v}\|^2 \cdot \text{tr} \left( \mathbb{E}_{X_2} \left[ \left( \boldsymbol{X}_2^\top (\boldsymbol{I} - P_{\boldsymbol{X}_1}) \boldsymbol{X}_2 \right)^{-1} \right] \right) \right]
$$

$$
\overset{(iii)}{=} \mathbb{E}_{X_1} \left[ \|P_{\boldsymbol{X}_1} \boldsymbol{v}\|^2 \cdot \frac{n_2}{p - n_1 - n_2 - 1} \right]
$$

$$
\overset{(iv)}{=} \frac{n_2}{p - n_1 - n_2 - 1} \cdot \frac{n_1}{p} \|\boldsymbol{v}\|^2, \quad (11)
$$

where $(i)$ follows from Lemma 5, $(ii)$ follows from Lemma 6, $(iii)$ follows from the fact that Lemma 7 actually holds for any $\boldsymbol{X}_2$ and $(iv)$ follows from Lemma 2. By combining eqs. (10) and (11), we complete the proof. $\square$

**Lemma 9.** *Suppose $\boldsymbol{V} = [\boldsymbol{X}_1, \boldsymbol{X}_2, \boldsymbol{X}_3]$ where $\boldsymbol{X}_1 \in \mathbb{R}^{p \times n_1}$, $\boldsymbol{X}_2 \in \mathbb{R}^{p \times n_2}$, $\boldsymbol{X}_3 \in \mathbb{R}^{p \times n_3}$ are random matrices with i.i.d. standard Gaussian elements and $\boldsymbol{v} \in \mathbb{R}^p$ is a fixed vector. Then we have:*

$$
\mathbb{E} \left[ \boldsymbol{v}^\top [\boldsymbol{X}_1 \quad \boldsymbol{0} \quad \boldsymbol{0}] (\boldsymbol{V}^\top \boldsymbol{V})^{-1} \begin{bmatrix} \boldsymbol{0} \\ \boldsymbol{X}_2^\top \\ \boldsymbol{0} \end{bmatrix} \boldsymbol{v} \right] = -\frac{n_1 n_2}{p(p - n_1 - n_2 - n_3 - 1)} \|\boldsymbol{v}\|^2
$$

*Proof.* First of all, we observe that:

$$
2 \boldsymbol{v}^\top [\boldsymbol{X}_1 \quad \boldsymbol{0} \quad \boldsymbol{0}] (\boldsymbol{V}^\top \boldsymbol{V})^{-1} \begin{bmatrix} \boldsymbol{0} \\ \boldsymbol{X}_2^\top \\ \boldsymbol{0} \end{bmatrix} \boldsymbol{v} = \left\| \boldsymbol{V}^\dagger \begin{bmatrix} \boldsymbol{X}_1^\top \\ \boldsymbol{X}_2^\top \\ \boldsymbol{0} \end{bmatrix} \boldsymbol{v} \right\|^2 - \left\| \boldsymbol{V}^\dagger \begin{bmatrix} \boldsymbol{X}_1^\top \\ \boldsymbol{0} \\ \boldsymbol{0} \end{bmatrix} \boldsymbol{v}_1 \right\|^2 - \left\| \boldsymbol{V}^\dagger \begin{bmatrix} \boldsymbol{0}^\top \\ \boldsymbol{X}_2^\top \\ \boldsymbol{0} \end{bmatrix} \boldsymbol{v} \right\|^2.
$$

By taking expectation over both sides of the equation, we have:

$$2\mathbb{E}\left[\boldsymbol{v}^\top\begin{bmatrix}\boldsymbol{X}_1 & \boldsymbol{0} & \boldsymbol{0}\end{bmatrix}(\boldsymbol{V}^\top\boldsymbol{V})^{-1}\begin{bmatrix}\boldsymbol{0}\\ \boldsymbol{X}_2^\top\\ \boldsymbol{0}\end{bmatrix}\boldsymbol{v}\right]$$

$$=\mathbb{E}\left\|\boldsymbol{V}^\dagger\begin{bmatrix}\boldsymbol{X}_1^\top\\ \boldsymbol{X}_2^\top\\ \boldsymbol{0}\end{bmatrix}\boldsymbol{v}\right\|^2-\mathbb{E}\left\|\boldsymbol{V}^\dagger\begin{bmatrix}\boldsymbol{X}_1^\top\\ \boldsymbol{0}\\ \boldsymbol{0}\end{bmatrix}\boldsymbol{v}\right\|^2-\mathbb{E}\left\|\boldsymbol{V}^\dagger\begin{bmatrix}\boldsymbol{0}^\top\\ \boldsymbol{X}_2\\ \boldsymbol{0}\end{bmatrix}\boldsymbol{v}\right\|^2$$

$$\stackrel{(i)}{=}\frac{n_1+n_2}{p}\cdot\left(1+\frac{n_3}{p-n_1-n_2-n_3-1}\right)\|\boldsymbol{v}\|^2-\frac{n_1}{p}\cdot\left(1+\frac{n_2+n_3}{p-n_1-n_2-n_3-1}\right)\|\boldsymbol{v}\|^2$$

$$-\frac{n_2}{p}\cdot\left(1+\frac{n_1+n_3}{p-n_1-n_2-n_3-1}\right)\|\boldsymbol{v}\|^2$$

$$=-\frac{2n_1n_2}{p(p-n_1-n_2-n_3-1)}\|\boldsymbol{v}\|^2,$$

where $(i)$ follows from Lemma 8. By dividing both sides by 2, we complete the proof. $\square$

**Corollary 1.** *Suppose* $\boldsymbol{V}=[\boldsymbol{X}_1,\boldsymbol{X}_2,\boldsymbol{X}_3]$ *where* $\boldsymbol{X}_1\in\mathbb{R}^{p\times n_1}$, $\boldsymbol{X}_2\in\mathbb{R}^{p\times n_2}$, $\boldsymbol{X}_3\in\mathbb{R}^{p\times n_3}$ *are random matrices with i.i.d. standard Gaussian elements and* $\boldsymbol{v}_1,\boldsymbol{v}_2\in\mathbb{R}^p$ *are fixed vectors. Then we have:*

$$\mathbb{E}\left[\boldsymbol{v}_1^\top\begin{bmatrix}\boldsymbol{X}_1 & \boldsymbol{0} & \boldsymbol{0}\end{bmatrix}(\boldsymbol{V}^\top\boldsymbol{V})^{-1}\begin{bmatrix}\boldsymbol{0}\\ \boldsymbol{X}_2^\top\\ \boldsymbol{0}\end{bmatrix}\boldsymbol{v}_2\right]=\frac{n_1n_2\left(\|\boldsymbol{v}_1-\boldsymbol{v}_2\|^2-\|\boldsymbol{v}_1\|^2-\|\boldsymbol{v}_2\|^2\right)}{2p(p-n_1-n_2-n_3-1)}$$

*Proof.* To simplify the notation, we denote $\boldsymbol{V}_1=\begin{bmatrix}\boldsymbol{X}_1 & \boldsymbol{0} & \boldsymbol{0}\end{bmatrix}$ and $\boldsymbol{V}_2=\begin{bmatrix}\boldsymbol{0} & \boldsymbol{X}_2 & \boldsymbol{0}\end{bmatrix}$. Then according to Lemma 9, we first have:

$$\mathbb{E}\left[(\boldsymbol{v}_1-\boldsymbol{v}_2)^\top\boldsymbol{V}_1(\boldsymbol{V}^\top\boldsymbol{V})^{-1}\boldsymbol{V}_2^\top(\boldsymbol{v}_1-\boldsymbol{v}_2)\right]=-\frac{n_1n_2}{p(p-n_1-n_2-n_3-1)}\|\boldsymbol{v}_1-\boldsymbol{v}_2\|^2.$$

On the other hand, we have:

$$\mathbb{E}\left[(\boldsymbol{v}_1-\boldsymbol{v}_2)^\top\boldsymbol{V}_1(\boldsymbol{V}^\top\boldsymbol{V})^{-1}\boldsymbol{V}_2^\top(\boldsymbol{v}_1-\boldsymbol{v}_2)\right]$$

$$=\mathbb{E}\left[\boldsymbol{v}_1^\top\boldsymbol{V}_1(\boldsymbol{V}^\top\boldsymbol{V})^{-1}\boldsymbol{V}_2^\top\boldsymbol{v}_1\right]+\mathbb{E}\left[\boldsymbol{v}_2^\top\boldsymbol{V}_1(\boldsymbol{V}^\top\boldsymbol{V})^{-1}\boldsymbol{V}_2^\top\boldsymbol{v}_2\right]-2\mathbb{E}\left[\boldsymbol{v}_1^\top\boldsymbol{V}_1(\boldsymbol{V}^\top\boldsymbol{V})^{-1}\boldsymbol{V}_2^\top\boldsymbol{v}_2\right]$$

$$\stackrel{(i)}{=}-\frac{n_1n_2\|\boldsymbol{v}_1\|^2}{p(p-n_1-n_2-n_3-1)}-\frac{n_1n_2\|\boldsymbol{v}_2\|^2}{p(p-n_1-n_2-n_3-1)}-2\mathbb{E}\left[\boldsymbol{v}_1^\top\boldsymbol{V}_1(\boldsymbol{V}^\top\boldsymbol{V})^{-1}\boldsymbol{V}_2^\top\boldsymbol{v}_2\right],$$

where $(i)$ follows from Lemma 9. By combining the above two equations, we complete the proof. $\square$

Next, we provide our supporting lemmas that help to prove the advantage of sequential replay as follows.

**Lemma 10.** *Given* $n,p,t,M,T$ *are fixed positive integers where* $t\leq T$ *and* $n+M<p$, *then we have:*

$$\left(1-\frac{M}{(t-l-1)p}\right)^{t-l-1}\left(1-\frac{n}{p}\right)>1-\frac{n+M}{p},$$

*for any non-negative integer* $l<t$

*Proof.* We first notice the fact that for $k=0,1,2,...,t-l-2$, we have

$$\left(1-\frac{M}{(t-l-1)p}\right)\left(1-\frac{n+\frac{kM}{t-l-1}}{p}\right)>1-\frac{n+\frac{(k+1)M}{t-l-1}}{p}.$$

By applying this argument recursively, we will have

$$\left(1-\frac{M}{(t-l-1)p}\right)^{t-l-1}\left(1-\frac{n}{p}\right)>1-\frac{n+(t-l-1)\frac{M}{t-l-1}}{p}=1-\frac{n+M}{p}.$$

$\square$

**Lemma 11.** *Given $n, p, t, M, T$ are fixed positive integers where $t \leq T$ and $n + M < p$, then for any non-negative integer $l < t - 1$, we have:*

$$\left(1 - \frac{M}{(t-l-1)p}\right)^{t-l-1} \left(1 - \frac{n}{p}\right) < 1 - \frac{n+M}{p} + \frac{(n+M)M}{p^2},$$

*if $p > TM$.*

*Proof.* According to the binomial theorem, we have:

$$\left(1 - \frac{M}{(t-l-1)p}\right)^{t-l-1} \left(1 - \frac{n}{p}\right)$$
$$= \left(1 - \frac{M}{p} + \sum_{k=2}^{t-l-1} \binom{t-l-1}{k} \left(-\frac{M}{(t-l-1)p}\right)^k\right) \left(1 - \frac{n}{p}\right) \quad (12)$$

If $t - l - 1 = 1$ or $t - l - 1 = 2$, the proof is trivial. If $t - l - 1 \geq 3$, we have

$$\sum_{k=2}^{t-l-1} \binom{t-l-1}{k} \left(-\frac{M}{(t-l-1)p}\right)^k = \binom{t-l-1}{2} \left(\frac{M}{(t-l-1)p}\right)^2$$
$$+ \sum_{k=3}^{t-l-1} \binom{t-l-1}{k} \left(-\frac{M}{(t-l-1)p}\right)^k. \quad (13)$$

To simplify the notation, we denote $m = \frac{M}{t-l-1}$. We first discuss if $t - l - 1$ is even. Then, we have:

$$\sum_{k=3}^{t-l-1} \binom{t-l-1}{k} \left(-\frac{M}{(t-l-1)p}\right)^k$$
$$= \sum_{k=3}^{(t-l+1)/2} \left[\binom{t-l-1}{2k-3} \left(-\frac{m}{p}\right)^{2k-3} + \binom{t-l-1}{2k-2} \left(-\frac{m}{p}\right)^{2k-2}\right]$$
$$= \sum_{k=3}^{(t-l+1)/2} \left[\frac{(t-l-1)!}{(2k-3)!(t-l-2k+2)!} \left(-\frac{m}{p}\right)^{2k-3} + \frac{(t-l-1)!}{(2k-2)!(t-l-2k+1)!} \left(-\frac{m}{p}\right)^{2k-2}\right]$$
$$= -\sum_{k=3}^{(t-l+1)/2} \frac{(t-l-1)!}{(2k-3)!(t-l-2k+1)!} \left(\frac{m}{p}\right)^{2k-3} \left[\frac{1}{t-l-2k+2} - \frac{1}{2k-2} \cdot \frac{m}{p}\right]$$
$$\overset{(i)}{<} 0 \quad (14)$$

where $(i)$ follows from the fact that $p > TM$. We then discuss if $t - l - 1$ is odd, we have:

$$\sum_{k=3}^{t-l-1} \binom{t-l-1}{k} \left(-\frac{M}{(t-l-1)p}\right)^k$$
$$= \sum_{k=3}^{(t-l)/2} \left[\binom{t-l-1}{2k-3} \left(-\frac{m}{p}\right)^{2k-3} + \binom{t-l-1}{2k-2} \left(-\frac{m}{p}\right)^{2k-2}\right] + \left(-\frac{m}{p}\right)^{t-l-1}$$
$$\overset{(i)}{<} \sum_{k=3}^{(t-l)/2} \left[\frac{(t-l-1)!}{(2k-3)!(t-l-2k+2)!} \left(-\frac{m}{p}\right)^{2k-3} + \frac{(t-l-1)!}{(2k-2)!(t-l-2k+1)!} \left(-\frac{m}{p}\right)^{2k-2}\right]$$
$$= -\sum_{k=3}^{(t-l)/2} \frac{(t-l-1)!}{(2k-3)!(t-l-2k+1)!} \left(\frac{m}{p}\right)^{2k-3} \left[\frac{1}{t-l-2k+2} - \frac{1}{2k-2} \cdot \frac{m}{p}\right]$$
$$\overset{(ii)}{<} 0 \quad (15)$$

where $(i)$ follows from the fact that $t - l - 1$ is odd and $(ii)$ follows from the fact that $p > TM$. By combing eqs. (12) to (15), we conclude:

$$
\left(1 - \frac{M}{(t-l-1)p}\right)^{t-l-1}\left(1 - \frac{n}{p}\right)
$$

$$
< \left(1 - \frac{M}{p} + \binom{t-l-1}{2}\frac{M^2}{(t-l-1)^2 p^2}\right)\left(1 - \frac{n}{p}\right)
$$

$$
= 1 - \frac{n+M}{p} + \frac{nM + \frac{(t-l-1)(t-l-2)}{2}\frac{M^2}{(t-l-1)^2}}{p^2} - \binom{t-l-1}{2}\frac{nM^2}{(t-l-1)^2 p^3}
$$

$$
< 1 - \frac{n+M}{p} + \frac{(n+M)M}{p^2}.
$$

which completes the proof. $\qquad\square$

**Lemma 12.** *Given $n, p, t, M, T$ are fixed positive integers where $t \leq T$ and $n + M < p$, then we have:*

$$
\left(1 - \frac{n+M}{p} + \frac{(n+M)M}{p^2}\right)^t < \left(1 - \frac{n+M}{p}\right)^t + \frac{T^2(n+M)M}{p^2}.
$$

*Proof.* We first have:

$$
\left(1 - \frac{n+M}{p} + \frac{(n+M)M}{p^2}\right)^t = \left(1 - \frac{n+M}{p}\right)^t + \sum_{k=0}^{t-1}\underbrace{\binom{t}{k}\left(1 - \frac{n+M}{p}\right)^k\left(\frac{(n+M)M}{p^2}\right)^{t-k}}_{\alpha_k}
$$

$$
\tag{16}
$$

We further notice that for $k = 0, 1, .., t-2$:

$$
\binom{t}{k}\left(1 - \frac{n+M}{p}\right)^k\left(\frac{(n+M)M}{p^2}\right)^{t-k} - \binom{t}{k+1}\left(1 - \frac{n+M}{p}\right)^{k+1}\left(\frac{(n+M)M}{p^2}\right)^{t-k-1}
$$

$$
= \frac{t!}{k!(t-k-1)!}\left(1 - \frac{n+M}{p}\right)^k\left(\frac{(n+M)M}{p^2}\right)^{t-k-1}\left[\frac{(n+M)M}{(t-k)p^2} - \frac{1}{k+1}\left(1 - \frac{n+M}{p}\right)\right]
$$

$$
< \frac{t!}{k!(t-k-1)!}\left(1 - \frac{n+M}{p}\right)^k\left(\frac{(n+M)M}{p^2}\right)^{t-k-1}\left[\frac{(n+M)M}{p^2} - \frac{1}{T}\left(1 - \frac{n+M}{p}\right)\right]
$$

$$
\overset{(i)}{<} 0,
$$

$$
\tag{17}
$$

where $(i)$ follows from the fact that $p > (n+M)(T+1)$. We note that eq. (17) shows that the term $\alpha_k$ achieves the maximum at $k = t-1$. Therefore, we can upper bound eq. (16) by

$$
\left(1 - \frac{n+M}{p} + \frac{(n+M)M}{p^2}\right)^t < \left(1 - \frac{n+M}{p}\right)^t + t\binom{t}{t-1}\left(1 - \frac{n+M}{p}\right)^{t-1}\left(\frac{(n+M)M}{p^2}\right)
$$

$$
< \left(1 - \frac{n+M}{p}\right)^t + \frac{T^2(n+M)M}{p^2}
$$

which completes the proof. $\qquad\square$

Here, we present a tighter version of Lemma 12, which helps us to prove Theorem 3 in Section 5.2.

**Lemma 13.** *Given $n, p, t, M, T$ are fixed positive integers where $t \leq T$ and $n + M < p$, then we have:*

$$
\left(1 - \frac{n+M}{p} + \frac{(n+M)M}{p^2}\right)^t < \left(1 - \frac{n+M}{p}\right)^t + \frac{t(n+M)M}{p^2} + \frac{T^3(n+M)^2 M^2}{2p^4}.
$$

*Proof.* We first have:

$$
\left(1 - \frac{n+M}{p} + \frac{(n+M)M}{p^2}\right)^t = \left(1 - \frac{n+M}{p}\right)^t + \binom{t}{t-1}\left(1 - \frac{n+M}{p}\right)^{t-1}\left(\frac{(n+M)M}{p^2}\right)
$$

$$
+ \sum_{k=0}^{t-2}\binom{t}{k}\left(1 - \frac{n+M}{p}\right)^k\left(\frac{(n+M)M}{p^2}\right)^{t-k}
$$

$$
< \left(1 - \frac{n+M}{p}\right)^t + \frac{T(n+M)M}{p^2}
$$

$$
+ \sum_{k=0}^{t-2}\underbrace{\binom{t}{k}\left(1 - \frac{n+M}{p}\right)^k\left(\frac{(n+M)M}{p^2}\right)^{t-k}}_{\alpha_k} \tag{18}
$$

By the same argument as eq. (17), we know that the term $\alpha_k$ achieves the maximum at $k = t - 2$. Therefore, we can upper bound eq. (18) by

$$
\left(1 - \frac{n+M}{p} + \frac{(n+M)M}{p^2}\right)^t
$$

$$
< \left(1 - \frac{n+M}{p}\right)^t + \frac{t(n+M)M}{p^2} + (t-1)\binom{t}{t-2}\left(1 - \frac{n+M}{p}\right)^{t-2}\left(\frac{(n+M)M}{p^2}\right)^2
$$

$$
< \left(1 - \frac{n+M}{p}\right)^t + \frac{t(n+M)M}{p^2} + \frac{T^3(n+M)^2M^2}{2p^4}.
$$

$\square$

**Lemma 14.** *Given $n, p, t, M, T$ are fixed positive integers where $M \geq 2$, $t \leq T$ and $n + M < p$, then for any non-negative integer $l < t - 1$, we have:*

$$
\left(1 - \frac{n+M}{p} + \frac{(n+M)M}{p^2}\right)^l\left(1 - \frac{M}{(t-l-1)p}\right)^{t-l-1} < \left(1 - \frac{1}{Tp}\right)\left(1 - \frac{n+M}{p}\right)^l,
$$

*if $p > \frac{T(n+M)M}{M-1} + n + M$.*

*Proof.* By dividing $\left(1 - \frac{n+M}{p}\right)^l$ on both sides, it is equivalent to prove

$$
\left(1 + \frac{(n+M)M}{p^2 - p(n+M)}\right)^l\left(1 - \frac{M}{(t-l-1)p}\right)^{t-l-1} < 1 - \frac{1}{Tp}.
$$

According to AM-GM inequality, we have:

$$
\left(1 + \frac{(n+M)M}{p^2 - p(n+M)}\right)^l\left(1 - \frac{M}{(t-l-1)p}\right)^{t-l-1}
$$

$$
\leq \left[\frac{l\left(1 + \frac{(n+M)M}{p^2-p(n+M)}\right) + (t-l-1)\left(1 - \frac{M}{(t-l-1)p}\right)}{t-1}\right]^{t-1}
$$

$$
= \left[1 + \frac{\frac{l(n+M)M}{p^2-p(n+M)} - \frac{M}{p}}{t-1}\right]^{t-1}. \tag{19}
$$

When $p > \frac{T(n+M)M}{M-1} + n + M$, we have:

$$
\frac{l(n+M)M}{p^2 - p(n+M)} - \frac{M}{p} < \frac{T(n+M)M}{p^2 - p(n+M)} - \frac{M}{p} < -\frac{1}{p}. \tag{20}
$$

Therefore, by combining eqs. (19) and (20), we have:

$$\left(1 + \frac{(n+M)M}{p^2 - p(n+M)}\right)^l \left(1 - \frac{M}{(t-l-1)p}\right)^{t-l} < \left(1 - \frac{1}{(t-1)p}\right)^{t-1}$$

$$< 1 - \frac{1}{(t-1)p} < 1 - \frac{1}{Tp},$$

which completes the proof. □

**Lemma 15.** *Given $n, p, t, M, T$ are fixed positive integers where $M \geq 2$, $t \leq T$ and $n + M < p$, then we have:*

$$\prod_{l=0}^{t-2}\left[\left(1 - \frac{M}{(t-l-1)p}\right)^{t-l-1}\left(1 - \frac{n}{p}\right)\right] - \prod_{l=0}^{i-2}\left[\left(1 - \frac{M}{(i-l-1)p}\right)^{i-l-1}\left(1 - \frac{n}{p}\right)\right]$$

$$> \left[\left(1 - \frac{n+M}{p}\right)^{t-1} - \left(1 - \frac{n+M}{p}\right)^{i-1}\right],$$

*if $p > 2T^3(n+M)^2$.*

*Proof.* We first have:

$$\prod_{l=0}^{t-2}\left[\left(1 - \frac{M}{(t-l-1)p}\right)^{t-l-1}\left(1 - \frac{n}{p}\right)\right] - \prod_{l=0}^{i-2}\left[\left(1 - \frac{M}{(i-l-1)p}\right)^{i-l-1}\left(1 - \frac{n}{p}\right)\right]$$

$$= \prod_{l=0}^{t-i-1}\left[\left(1 - \frac{M}{(t-l-1)p}\right)^{t-l-1}\left(1 - \frac{n}{p}\right)\right]\prod_{l=t-i}^{t-2}\left[\left(1 - \frac{M}{(t-l-1)p}\right)^{t-l-1}\left(1 - \frac{n}{p}\right)\right]$$

$$- \prod_{l=0}^{i-2}\left[\left(1 - \frac{M}{(i-l-1)p}\right)^{i-l-1}\left(1 - \frac{n}{p}\right)\right]$$

$$= \prod_{l=0}^{t-i-1}\left[\left(1 - \frac{M}{(t-l-1)p}\right)^{t-l-1}\left(1 - \frac{n}{p}\right)\right]\prod_{l=0}^{i-2}\left[\left(1 - \frac{M}{(i-l-1)p}\right)^{i-l-1}\left(1 - \frac{n}{p}\right)\right]$$

$$- \prod_{l=0}^{i-2}\left[\left(1 - \frac{M}{(i-l-1)p}\right)^{i-l-1}\left(1 - \frac{n}{p}\right)\right]$$

$$= \prod_{l=0}^{i-2}\left[\left(1 - \frac{M}{(i-l-1)p}\right)^{i-l-1}\left(1 - \frac{n}{p}\right)\right]\underbrace{\left\{\prod_{l=0}^{t-i-1}\left[\left(1 - \frac{M}{(t-l-1)p}\right)^{t-l-1}\left(1 - \frac{n}{p}\right)\right] - 1\right\}}_{\gamma_1}$$

$$\overset{(i)}{>} \left(1 - \frac{n+M}{p} + \frac{(n+M)M}{p^2}\right)^{i-1}\left\{\left[\left(1 - \frac{M}{p}\right)\left(1 - \frac{n}{p}\right)\right]^{t-i} - 1\right\}$$

$$= \left(1 - \frac{n+M}{p} + \frac{(n+M)M}{p^2}\right)^{i-1}\left[\left(1 - \frac{n+M}{p} + \frac{nM}{p^2}\right)^{t-i} - 1\right]$$

$$= \left[\left(1 - \frac{n+M}{p}\right)^{i-1} + \sum_{k=1}^{i-1}\binom{i-1}{k}\left(\frac{(n+M)M}{p^2}\right)^k\left(1 - \frac{n+M}{p}\right)^{i-k-1}\right]$$

$$\left[\left(1 - \frac{n+M}{p}\right)^{t-i} - 1 + \sum_{k=1}^{t-i}\binom{t-i}{k}\left(\frac{nM}{p^2}\right)^k\left(1 - \frac{n+M}{p}\right)^{t-i-k}\right]$$

$$> \left(1 - \frac{n+M}{p}\right)^{i-1}\left[\left(1 - \frac{n+M}{p}\right)^{t-i} - 1\right]$$

$$+ \left(1 - \frac{n+M}{p}\right)^{i-1} \underbrace{\sum_{k=1}^{t-i} \binom{t-i}{k} \left(\frac{nM}{p^2}\right)^k \left(1 - \frac{n+M}{p}\right)^{t-i-k}}_{\gamma_2}$$

$$+ \underbrace{\left[\left(1 - \frac{n+M}{p}\right)^{t-i} - 1\right] \sum_{k=1}^{i-1} \underbrace{\binom{i-1}{k} \left(\frac{(n+M)M}{p^2}\right)^k \left(1 - \frac{n+M}{p}\right)^{i-k-1}}_{\gamma_{3,k}}}_{\gamma_3} \tag{21}$$

where $(i)$ follows from Lemma 11 together with the fact that term $\gamma_1 < 0$ and from the fact that $\left(1 - \frac{M}{(t-l-1)p}\right)^{t-l-1} > 1 - \frac{M}{p}$ for $l = 0, 1, .., t-i-1$. Now. we prove $\gamma_2 + \gamma_3 < 0$. We first focus on $\gamma_2$. We have:

$$\gamma_2 > \left(1 - \frac{n+M}{p}\right)^{i-1} \binom{t-i}{1} \left(\frac{nM}{p^2}\right) \left(1 - \frac{n+M}{p}\right)^{t-i-1}$$

$$> \left(1 - \frac{n+M}{p}\right)^T \frac{nM}{p^2}$$

$$> \left(1 - \frac{T(n+M)}{p}\right) \frac{nM}{p^2} \tag{22}$$

We then focus on term $\gamma_3$. Consider:

$$\binom{i-1}{k} \left(\frac{(n+M)M}{p^2}\right)^k \left(1 - \frac{n+M}{p}\right)^{i-k-1} - \binom{i-1}{k+1} \left(\frac{(n+M)M}{p^2}\right)^{k+1} \left(1 - \frac{n+M}{p}\right)^{i-k-2}$$

$$= \frac{(i-1)!}{k!(i-k-2)!} \left(\frac{(n+M)M}{p^2}\right)^k \left(1 - \frac{n+M}{p}\right)^{i-k-2}$$

$$\cdot \left[\frac{1}{i-k-1} \left(1 - \frac{n+M}{p}\right) - \frac{1}{k+1} \frac{(n+M)M}{p^2}\right]$$

$$\overset{(i)}{>} \frac{(i-1)!}{k!(i-k-2)!} \left(\frac{(n+M)M}{p^2}\right)^k \left(1 - \frac{n+M}{p}\right)^{i-k-2} \left[\frac{1}{T} \left(1 - \frac{n+M}{p}\right) - \frac{(n+M)M}{2p^2}\right]$$

$$\overset{(ii)}{>} 0, \tag{23}$$

where $(i)$ follows from $k \in [i-1]$ and $(ii)$ follows from the fact that $p > 2(n+M)$. This indicates that $\gamma_{3,k}$ achieves maximum at $k = 1$. We recall that $\left[\left(1 - \frac{n+M}{p}\right)^{t-i} - 1\right] < 0$. Therefore, we have:

$$\gamma_3 > \left[\left(1 - \frac{n+M}{p}\right)^{t-i} - 1\right] (i-1) \binom{i-1}{1} \left(\frac{(n+M)M}{p^2}\right) \left(1 - \frac{n+M}{p}\right)^{i-2}$$

$$> \left[\left(1 - \frac{n+M}{p}\right)^{t-i} - 1\right] \frac{T^2(n+M)M}{p^2}$$

$$= \left[\sum_{k=1}^{t-i} \binom{t-i}{k} \left(-\frac{n+M}{p}\right)^k\right] \frac{T^2(n+M)M}{p^2}. \tag{24}$$

For $k$ is even and less than or equal to $t - i$(i.e., $k = 2, 4, 6, ...,$ and $k \geq t - i$), we have:

$$\binom{t-i}{k} \left(-\frac{n+M}{p}\right)^k + \binom{t-i}{k+1} \left(-\frac{n+M}{p}\right)^{k+1}$$

$$= \frac{(t-i)!}{k!(t-i-k-1)!} \left(\frac{n+M}{p}\right)^k \left[\frac{1}{t-i-k} - \frac{n+M}{(k+1)p}\right]$$

$$> \frac{(t-i)!}{k!(t-i-k-1)!} \left(\frac{n+M}{p}\right)^k \left[\frac{1}{T} - \frac{n+M}{3p}\right]$$

$$\overset{(i)}{>} 0, \tag{25}$$

where $(i)$ follows from $p > \frac{(n+M)T}{3}$. By combining eqs. (24) and (25) and simply discussing when $t-i$ is odd or even, we can conclude

$$\sum_{k=1}^{t-i} \binom{t-i}{k} \left(-\frac{n+M}{p}\right)^k > \binom{t-i}{1}\left(-\frac{n+M}{p}\right) > -\frac{T(n+M)}{p},$$

which implies:

$$\gamma_3 > -\frac{T^3(n+M)^2 M}{p^3}. \tag{26}$$

Now, by combining eqs. (21), (22) and (26), we have:

$$\prod_{l=0}^{t-2}\left[\left(1-\frac{M}{(t-l-1)p}\right)^{t-l-1}\left(1-\frac{n}{p}\right)\right] - \prod_{l=0}^{i-2}\left[\left(1-\frac{M}{(i-l-1)p}\right)^{i-l-1}\left(1-\frac{n}{p}\right)\right]$$

$$> \left(1-\frac{n+M}{p}\right)^{i-1}\left[\left(1-\frac{n+M}{p}\right)^{t-i}-1\right] + \left(1-\frac{T(n+M)}{p}\right)\frac{nM}{p^2} - \frac{T^3(n+M)^2 M}{p^3}$$

$$\overset{(i)}{>} \left(1-\frac{n+M}{p}\right)^{i-1}\left[\left(1-\frac{n+M}{p}\right)^{t-i}-1\right], \tag{27}$$

where $(i)$ follows from the fact that $p > 2T^3(n+M)^2$. $\qquad\square$

**Lemma 16.** *Given $n, p, t, M, T$ are fixed positive integers where $M \geq 2$, $t \leq T$ and $n + M < p$, then for any non-negative integer $i < t$, we have:*

$$\prod_{l=0}^{t-2}\left[\left(1-\frac{M}{(t-l-1)p}\right)^{t-l-1}\left(1-\frac{n}{p}\right)\right] - \prod_{l=0}^{i-2}\left[\left(1-\frac{M}{(i-l-1)p}\right)^{i-l-1}\left(1-\frac{n}{p}\right)\right]$$

$$< \left(1-\frac{n+M}{p}\right)^{i-1}\left[\left(1-\frac{n+M}{p}\right)^{t-i}-1\right] + \frac{T^2(n+M)M}{p^2}.$$

*if $p > (n+M)T$.*

*Proof.* We first consider:

$$\prod_{l=0}^{t-2}\left[\left(1-\frac{M}{(t-l-1)p}\right)^{t-l-1}\left(1-\frac{n}{p}\right)\right] - \prod_{l=0}^{i-2}\left[\left(1-\frac{M}{(i-l-1)p}\right)^{i-l-1}\left(1-\frac{n}{p}\right)\right]$$

$$= \prod_{l=0}^{t-i-1}\left[\left(1-\frac{M}{(t-l-1)p}\right)^{t-l-1}\left(1-\frac{n}{p}\right)\right]\prod_{l=t-i}^{t-2}\left[\left(1-\frac{M}{(t-l-1)p}\right)^{t-l-1}\left(1-\frac{n}{p}\right)\right]$$

$$\quad - \prod_{l=0}^{i-2}\left[\left(1-\frac{M}{(i-l-1)p}\right)^{i-l-1}\left(1-\frac{n}{p}\right)\right]$$

$$= \prod_{l=0}^{t-i-1}\left[\left(1-\frac{M}{(t-l-1)p}\right)^{t-l-1}\left(1-\frac{n}{p}\right)\right]\prod_{l=0}^{i-2}\left[\left(1-\frac{M}{(i-l-1)p}\right)^{i-l-1}\left(1-\frac{n}{p}\right)\right]$$

$$\quad - \prod_{l=0}^{i-2}\left[\left(1-\frac{M}{(i-l-1)p}\right)^{i-l-1}\left(1-\frac{n}{p}\right)\right]$$

$$= \prod_{l=0}^{i-2}\left[\left(1-\frac{M}{(i-l-1)p}\right)^{i-l-1}\left(1-\frac{n}{p}\right)\right]\underbrace{\left\{\prod_{l=0}^{t-i-1}\left[\left(1-\frac{M}{(t-l-1)p}\right)^{t-l-1}\left(1-\frac{n}{p}\right)\right]-1\right\}}_{\gamma_1}$$

$$\overset{(i)}{<} \left(1 - \frac{n+M}{p}\right)^{i-1} \left[\left(1 - \frac{n+M}{p} + \frac{(n+M)M}{p^2}\right)^{t-i} - 1\right],$$

$$\overset{(ii)}{<} \left(1 - \frac{n+M}{p}\right)^{i-1} \left[\left(1 - \frac{n+M}{p}\right)^{t-i} - 1 + \frac{T^2(n+M)M}{p^2}\right]$$

$$< \left(1 - \frac{n+M}{p}\right)^{i-1} \left[\left(1 - \frac{n+M}{p}\right)^{t-i} - 1\right] + \frac{T^2(n+M)M}{p^2} \tag{28}$$

where $(i)$ follows from Lemmas 10 and 11 and the fact that $\gamma_1 < 0$; $(ii)$ follows from Lemma 12. $\qquad\square$

Here, we present a tighter version of Lemma 16, which helps to prove Theorem 3 in Section 5.2.

**Lemma 17.** *Given $n, p, t, M, T$ are fixed positive integers where $M \geq 2$, $t \leq T$ and $n + M < p$, then for any non-negative integer $i < t$, we have:*

$$\prod_{l=0}^{t-2}\left[\left(1 - \frac{M}{(t-l-1)p}\right)^{t-l-1}\left(1 - \frac{n}{p}\right)\right] - \prod_{l=0}^{i-2}\left[\left(1 - \frac{M}{(i-l-1)p}\right)^{i-l-1}\left(1 - \frac{n}{p}\right)\right]$$

$$< \left(1 - \frac{n+M}{p}\right)^{i-1}\left[\left(1 - \frac{n+M}{p}\right)^{t-i} - 1\right] + \frac{(t-i)(n+M)M}{p^2} + \frac{T^3(n+M)^2 M^2}{p^4}.$$

*if $p > (n+M)T$.*

*Proof.* The proof follows from the same as Lemma 16 but we use Lemma 13 instead of Lemma 12. $\qquad\square$

## B  PROOF OF PROPOSITIONS 1 AND 2 AND THEOREM 1

In this section, we will prove Propositions 1 and 2 by deriving the expected value of model error $\mathbb{E}[\mathcal{L}_i(\boldsymbol{w}_t)]$ for a generic pair $t, i$ with $t \geq i$. We omit the tilde notation of the memory data to simplify notations: $\boldsymbol{X}_{t,i} \coloneqq \widetilde{\boldsymbol{X}}_{t,i}$, $\boldsymbol{Y}_{t,i} \coloneqq \widetilde{\boldsymbol{Y}}_{t,i}$ and $\boldsymbol{z}_{t,i} \coloneqq \widetilde{\boldsymbol{z}}_{t,i}$ for $i \in [t-1]$. Similar to eq. (3), for the memory data, we have

$$\boldsymbol{Y}_{t,i} = \boldsymbol{X}_{t,i}^\top \boldsymbol{w}_i^* + \boldsymbol{z}_{t,i}. \tag{29}$$

where $\boldsymbol{z}_{t,i} \sim \mathcal{N}(0, \sigma_i^2 \boldsymbol{I}_p)$ is i.i.d. noise. Since there is no memory data involved in both training methods when $t = 1$, by combining Lemma 1 and the fact that $\boldsymbol{w}_0 = \boldsymbol{0}$, we can easily derive the first parameter as

$$\boldsymbol{w}_1 = P_{\boldsymbol{X}_1}\boldsymbol{w}_1^* + \boldsymbol{X}_1^\dagger \boldsymbol{z}_1,$$

Then, we calculate the expected value of the model error $\mathcal{L}_i(\boldsymbol{w}_1)$ as follows.

$$\mathbb{E}\left\|\boldsymbol{w}_1 - \boldsymbol{w}_i^*\right\|^2 \overset{(i)}{=} \mathbb{E}\left\|P_{\boldsymbol{X}_1}(\boldsymbol{w}_1^* - \boldsymbol{w}_i^*)\right\|^2 + \mathbb{E}\left\|(\boldsymbol{I} - P_{\boldsymbol{X}_1})\boldsymbol{w}_i^*\right\|^2 + \mathbb{E}\left\|\boldsymbol{X}_1^\dagger \boldsymbol{z}_1\right\|^2$$

$$\overset{(ii)}{=} \frac{n}{p}\mathbb{E}\left\|\boldsymbol{w}_1^* - \boldsymbol{w}_i^*\right\|^2 + \left(1 - \frac{n}{p}\right)\left\|\boldsymbol{w}_i^*\right\|^2 + \frac{n\sigma^2}{p-n-1}, \tag{30}$$

where $(i)$ follows from Lemma 4 and the fact that $\boldsymbol{z}_1$ are independent Gaussian with zero mean and $(ii)$ follows from Lemma 2 and Lemma 3. For $t \geq 2$, the two training methods use memory in different ways. We present them in the following two subsections.

### B.1  PROOF OF CONCURRENT REPLAY IN PROPOSITIONS 1 AND 2

To simplify, we apply the following notations to denote the current data in this subsection: $\boldsymbol{X}_t \coloneqq \boldsymbol{X}_{t,t}$, $\boldsymbol{Y}_t \coloneqq \boldsymbol{Y}_{t,t}$ and $\boldsymbol{z}_t \coloneqq \boldsymbol{z}_{t,t}$. Then, for each task $t$, the SGD convergent point $\boldsymbol{w}_t$ of training loss $\mathcal{L}_t^{\mathrm{tr}}(\boldsymbol{w}, \mathcal{D}_t \bigcup \mathcal{M}_t)$ is equivalent to the optimization problem:

$$\boldsymbol{w}_t = \min_{\boldsymbol{w}} \left\|\boldsymbol{w} - \boldsymbol{w}_{t-1}\right\|^2 \quad s.t. \ \boldsymbol{X}_{t,i}^\top \boldsymbol{w} = \boldsymbol{Y}_{t,i}, \ i \in [t].$$

Define $\boldsymbol{V}_t = [\boldsymbol{X}_{t,1}, \boldsymbol{X}_{t,2}, ..., \boldsymbol{X}_{t,t}]$ and $\vec{z}_t = [\boldsymbol{z}_{t,1}, \boldsymbol{z}_{t,2}, ..., \boldsymbol{z}_{t,t}]^{\top}$. According to Lemma 1, we have

$$
\boldsymbol{w}_t = \boldsymbol{w}_{t-1} + \boldsymbol{V}_t^{\dagger} \left( \begin{bmatrix} \boldsymbol{Y}_{t,1} \\ \boldsymbol{Y}_{t,2} \\ ... \\ \boldsymbol{Y}_{t,t} \end{bmatrix} - \boldsymbol{V}_t^{\top} \boldsymbol{w}_{t-1} \right)
$$

$$
= (\boldsymbol{I} - P_{\boldsymbol{V}_t}) \boldsymbol{w}_{t-1} + \boldsymbol{V}_t^{\dagger} \begin{bmatrix} \boldsymbol{X}_{t,1}^{\top} \boldsymbol{w}_1^* \\ \boldsymbol{X}_{t,2}^{\top} \boldsymbol{w}_2^* \\ ... \\ \boldsymbol{X}_{t,t}^{\top} \boldsymbol{w}_t^* \end{bmatrix} + \boldsymbol{V}_t^{\dagger} \vec{z}_t.
$$

Now, we fix $i$. The Coefficients $d_{0T}^{(\text{concurrent})}$ and $d_{ijkT}^{(\text{concurrent})}$ are extracted from expected value of model error $\mathbb{E}[\mathcal{L}_i(\boldsymbol{w}_t)]$ as follows.

$$
\mathbb{E}\|\boldsymbol{w}_t - \boldsymbol{w}_i^*\|^2 = \mathbb{E}\left\| (\boldsymbol{I} - P_{\boldsymbol{V}_t})(\boldsymbol{w}_{t-1} - \boldsymbol{w}_i^*) + \boldsymbol{V}_t^{\dagger} \begin{bmatrix} \boldsymbol{X}_{t,1}^{\top}(\boldsymbol{w}_1^* - \boldsymbol{w}_i^*) \\ \boldsymbol{X}_{t,2}^{\top}(\boldsymbol{w}_2^* - \boldsymbol{w}_i^*) \\ ... \\ \boldsymbol{X}_{t,t}^{\top}(\boldsymbol{w}_t^* - \boldsymbol{w}_i^*) \end{bmatrix} + \boldsymbol{V}_t^{\dagger} \vec{z}_t \right\|^2
$$

$$
\stackrel{(i)}{=} \mathbb{E}\|(\boldsymbol{I} - P_{\boldsymbol{V}_t})(\boldsymbol{w}_{t-1} - \boldsymbol{w}_i^*)\|^2 + \mathbb{E}\left\| \boldsymbol{V}_t^{\dagger} \begin{bmatrix} \boldsymbol{X}_{t,1}^{\top}(\boldsymbol{w}_1^* - \boldsymbol{w}_i^*) \\ \boldsymbol{X}_{t,2}^{\top}(\boldsymbol{w}_2^* - \boldsymbol{w}_i^*) \\ ... \\ \boldsymbol{X}_{t,t}^{\top}(\boldsymbol{w}_t^* - \boldsymbol{w}_i^*) \end{bmatrix} \right\|^2 + \mathbb{E}\left\| \boldsymbol{V}_t^{\dagger} \vec{z}_t \right\|^2
$$

$$
\stackrel{(ii)}{=} \left( 1 - \frac{n_t + M_t}{p} \right) \mathbb{E}\|\boldsymbol{w}_{t-1} - \boldsymbol{w}_i^*\|^2 + \mathbb{E}\left\| \boldsymbol{V}_t^{\dagger} \begin{bmatrix} \boldsymbol{X}_{t,1}^{\top}(\boldsymbol{w}_1^* - \boldsymbol{w}_i^*) \\ \boldsymbol{X}_{t,2}^{\top}(\boldsymbol{w}_2^* - \boldsymbol{w}_i^*) \\ ... \\ \boldsymbol{X}_{t,t}^{\top}(\boldsymbol{w}_t^* - \boldsymbol{w}_i^*) \end{bmatrix} \right\|^2
$$

$$
+ \frac{(n + M)\sigma^2}{p - n - M - 1}, \quad (31)
$$

where $(i)$ follows from Lemma 4 and the fact that $\vec{z}_t$ are independent Gaussian with zero mean and $(ii)$ follows from Lemma 2 and Lemma 3. Before we calculate the second term in eq. (31), we make the following notation simplification. We denote $\boldsymbol{V}_{t,j}$ as $\boldsymbol{V}_t$ with all zero elements except $\boldsymbol{X}_{t,j}$, i.e.,

$$
\boldsymbol{V}_{t,j} = [\boldsymbol{0}, ..., \boldsymbol{X}_{t,j}, ..., \boldsymbol{0}] .
$$

Then we have:

$$
\mathbb{E}\left\| \boldsymbol{V}_t^{\dagger} \begin{bmatrix} \boldsymbol{X}_{t,1}^{\top}(\boldsymbol{w}_1^* - \boldsymbol{w}_i^*) \\ \boldsymbol{X}_{t,2}^{\top}(\boldsymbol{w}_2^* - \boldsymbol{w}_i^*) \\ ... \\ \boldsymbol{X}_{t,t}^{\top}(\boldsymbol{w}_t^* - \boldsymbol{w}_i^*) \end{bmatrix} \right\|^2
$$

$$
= \mathbb{E}\left\| \sum_{j=1}^{t} \boldsymbol{V}_t^{\dagger} \boldsymbol{V}_{t,j}^{\top}(\boldsymbol{w}_j^* - \boldsymbol{w}_i^*) \right\|^2
$$

$$
= \sum_{j=1}^{t-1} \mathbb{E}\left\| \boldsymbol{V}_t^{\dagger} \boldsymbol{V}_{t,j}^{\top}(\boldsymbol{w}_j^* - \boldsymbol{w}_i^*) \right\|^2 + \sum_{j=1}^{t} \sum_{k=1, k \neq j}^{t} (\boldsymbol{w}_j^* - \boldsymbol{w}_i^*)^{\top} \boldsymbol{V}_{t,j}(\boldsymbol{V}_t^{\top} \boldsymbol{V}_t)^{-1} \boldsymbol{V}_{t,k}^{\top}(\boldsymbol{w}_k^* - \boldsymbol{w}_i^*)
$$

$$
\stackrel{(i)}{=} \sum_{j=1}^{t-1} \frac{M_{t,j}}{p} \left( 1 + \frac{n_t + M_t - M_{t,j}}{p - n_t - M_t - 1} \right) \|\boldsymbol{w}_j^* - \boldsymbol{w}_i^*\|^2 + \frac{n_t}{p} \left( 1 + \frac{M_t}{p - n_t - M_t - 1} \right) \|\boldsymbol{w}_t^* - \boldsymbol{w}_i^*\|^2
$$

$$
+ \sum_{j=1}^{t-2} \sum_{k=j+1}^{t-1} \frac{M_{t,j} M_{t,k}}{p(p - n_t - M_t - 1)} \left( \|\boldsymbol{w}_j^* - \boldsymbol{w}_k^*\|^2 - \|\boldsymbol{w}_j^* - \boldsymbol{w}_i^*\|^2 - \|\boldsymbol{w}_k^* - \boldsymbol{w}_i^*\|^2 \right)
$$

$$
+ \sum_{j=1}^{t-1} \frac{n_t M_{t,j}}{p(p - n_t - M_t - 1)} \left( \|\boldsymbol{w}_j^* - \boldsymbol{w}_t^*\|^2 - \|\boldsymbol{w}_j^* - \boldsymbol{w}_i^*\|^2 - \|\boldsymbol{w}_t^* - \boldsymbol{w}_i^*\|^2 \right) \quad (32)
$$

where $(i)$ follows from Lemma 8 and corollary 1. Recall that $n_t = n$, $M_{t,j} = \frac{M}{t-1}$ and the fact that $M_t = M$. By combining eqs. (31) and (32), we have:

$$
\begin{aligned}
\mathbb{E}\left\|\boldsymbol{w}_t - \boldsymbol{w}_i^*\right\|^2 = {} & \left(1 - \frac{n+M}{p}\right) \mathbb{E}\left\|\boldsymbol{w}_{t-1} - \boldsymbol{w}_i^*\right\|^2 \\
& + \sum_{j=1}^{t-1} \frac{M}{(t-1)p}\left(1 + \frac{n+M-\frac{M}{t-1}}{p-n-M-1}\right)\left\|\boldsymbol{w}_j^* - \boldsymbol{w}_i^*\right\|^2 \\
& + \frac{n}{p}\left(1 + \frac{M}{p-n-M-1}\right)\left\|\boldsymbol{w}_t^* - \boldsymbol{w}_i^*\right\|^2 \\
& + \sum_{j=1}^{t-2}\sum_{k=j+1}^{t-1} \frac{(\frac{M}{t-1})^2}{p(p-n-M-1)}\left(\left\|\boldsymbol{w}_j^* - \boldsymbol{w}_k^*\right\|^2 - \left\|\boldsymbol{w}_j^* - \boldsymbol{w}_i^*\right\|^2 - \left\|\boldsymbol{w}_k^* - \boldsymbol{w}_i^*\right\|^2\right) \\
& + \sum_{j=1}^{t-1} \frac{\frac{nM}{t-1}}{p(p-n-M-1)}\left(\left\|\boldsymbol{w}_j^* - \boldsymbol{w}_t^*\right\|^2 - \left\|\boldsymbol{w}_j^* - \boldsymbol{w}_i^*\right\|^2 - \left\|\boldsymbol{w}_t^* - \boldsymbol{w}_i^*\right\|^2\right) \\
& + \frac{(n+M)\sigma^2}{p-n-M-1},
\end{aligned}
$$

for $t \geq 2$. By iterating the above equation and combining it with eq. (30), we can have:

$$
\begin{aligned}
& \mathbb{E}\left\|\boldsymbol{w}_t - \boldsymbol{w}_i^*\right\|^2 \\
={} & \left(1 - \frac{n+M}{p}\right)^{t-1} \mathbb{E}\left\|\boldsymbol{w}_1 - \boldsymbol{w}_i^*\right\|^2 \\
& + \sum_{l=0}^{t-2}\left(1 - \frac{n+M}{p}\right)^l \sum_{j=1}^{t-l-1} \frac{M}{(t-l-1)p}\left(1 + \frac{n+M-\frac{M}{t-l-1}}{p-n-M-1}\right)\left\|\boldsymbol{w}_j^* - \boldsymbol{w}_i^*\right\|^2 \\
& + \sum_{l=0}^{t-2}\left(1 - \frac{n+M}{p}\right)^l \frac{n}{p}\left(1 + \frac{M}{p-n-M-1}\right)\left\|\boldsymbol{w}_{t-l}^* - \boldsymbol{w}_i^*\right\|^2 \\
& + \sum_{l=0}^{t-2}\left(1 - \frac{n+M}{p}\right)^l \sum_{j=1}^{t-l-2}\sum_{k=j+1}^{t-l-1} \frac{(\frac{M}{t-l-1})^2}{p(p-n-M-1)}\left(\left\|\boldsymbol{w}_j^* - \boldsymbol{w}_k^*\right\|^2 - \left\|\boldsymbol{w}_j^* - \boldsymbol{w}_i^*\right\|^2 - \left\|\boldsymbol{w}_k^* - \boldsymbol{w}_i^*\right\|^2\right) \\
& + \sum_{l=0}^{t-2}\left(1 - \frac{n+M}{p}\right)^l \sum_{j=1}^{t-l-1} \frac{\frac{nM}{t-l-1}}{p(p-n-M-1)}\left(\left\|\boldsymbol{w}_j^* - \boldsymbol{w}_{t-l}^*\right\|^2 - \left\|\boldsymbol{w}_j^* - \boldsymbol{w}_i^*\right\|^2 - \left\|\boldsymbol{w}_{t-l}^* - \boldsymbol{w}_i^*\right\|^2\right) \\
& + \sum_{l=0}^{t-2}\left(1 - \frac{n+M}{p}\right)^l \frac{(n+M)\sigma^2}{p-n-M-1} \\
={} & \left(1 - \frac{n}{p}\right)\left(1 - \frac{n+M}{p}\right)^{t-1}\left\|\boldsymbol{w}_i^*\right\|^2 + \left(1 - \frac{n+M}{p}\right)^{t-1}\frac{n}{p}\mathbb{E}\left\|\boldsymbol{w}_1^* - \boldsymbol{w}_i^*\right\|^2 \\
& + \sum_{l=0}^{t-2}\left(1 - \frac{n+M}{p}\right)^l \sum_{j=1}^{t-l-1} \frac{M}{(t-l-1)p}\left(1 + \frac{n+M-\frac{M}{t-l-1}}{p-n-M-1}\right)\left\|\boldsymbol{w}_j^* - \boldsymbol{w}_i^*\right\|^2 \\
& + \sum_{l=0}^{t-2}\left(1 - \frac{n+M}{p}\right)^l \frac{n}{p}\left(1 + \frac{M}{p-n-M-1}\right)\left\|\boldsymbol{w}_{t-l}^* - \boldsymbol{w}_i^*\right\|^2 \\
& + \sum_{l=0}^{t-2}\left(1 - \frac{n+M}{p}\right)^l \sum_{j=1}^{t-l-2}\sum_{k=j+1}^{t-l-1} \frac{(\frac{M}{t-l-1})^2}{p(p-n-M-1)}\left(\left\|\boldsymbol{w}_j^* - \boldsymbol{w}_k^*\right\|^2 - \left\|\boldsymbol{w}_j^* - \boldsymbol{w}_i^*\right\|^2 - \left\|\boldsymbol{w}_k^* - \boldsymbol{w}_i^*\right\|^2\right) \\
& + \sum_{l=0}^{t-2}\left(1 - \frac{n+M}{p}\right)^l \sum_{j=1}^{t-l-1} \frac{\frac{nM}{t-l-1}}{p(p-n-M-1)}\left(\left\|\boldsymbol{w}_j^* - \boldsymbol{w}_{t-l}^*\right\|^2 - \left\|\boldsymbol{w}_j^* - \boldsymbol{w}_i^*\right\|^2 - \left\|\boldsymbol{w}_{t-l}^* - \boldsymbol{w}_i^*\right\|^2\right)
\end{aligned}
$$

$$+ \left(1 - \frac{n}{p}\right)\left(1 - \frac{n+M}{p}\right)^{t-1} \frac{n\sigma^2}{p-n-1} + \sum_{l=0}^{t-2}\left(1 - \frac{n+M}{p}\right)^l \frac{(n+M)\sigma^2}{p-n-M-1}$$

$$= \left(1 - \frac{n}{p}\right)\left(1 - \frac{n+M}{p}\right)^{t-1} \|\boldsymbol{w}_i^*\|^2$$

$$+ \left\{\left(1 - \frac{n+M}{p}\right)^{t-1}\frac{n}{p} + \sum_{l=0}^{t-2}\left(1 - \frac{n+M}{p}\right)^l\left[\frac{M}{(t-l-1)p}\left(1 + \frac{n+M-\frac{M}{t-l-1}}{p-n-M-1}\right)\right.\right.$$

$$\left.\left. - \frac{\frac{nM}{(t-l-1)} + (t-l-2)(\frac{M}{t-l-1})^2}{p(p-n-M-1)}\right]\right\}\|\boldsymbol{w}_1^* - \boldsymbol{w}_i^*\|^2$$

$$+ \sum_{l=0}^{t-2}\left(1 - \frac{n+M}{p}\right)^l \sum_{j=2}^{t-l-1}\left[\frac{M}{(t-l-1)p}\left(1 + \frac{n+M-\frac{M}{t-l-1}}{p-n-M-1}\right)\right.$$

$$\left. - \frac{\frac{nM}{t-l-1} + (t-l-2)(\frac{M}{t-l-1})^2}{p(p-n-M-1)}\right]\left\|\boldsymbol{w}_j^* - \boldsymbol{w}_i^*\right\|^2$$

$$+ \sum_{l=0}^{t-2}\left(1 - \frac{n+M}{p}\right)^l\left[\frac{n}{p}\left(1 + \frac{M}{p-n-M-1}\right) - \frac{(t-l-1)\frac{nM}{t-l-1}}{p(p-n-M-1)}\right]\left\|\boldsymbol{w}_{t-l}^* - \boldsymbol{w}_i^*\right\|^2$$

$$+ \sum_{l=0}^{t-2}\left(1 - \frac{n+M}{p}\right)^l \sum_{j=1}^{t-l-2}\sum_{k=j+1}^{t-l-1}\frac{(\frac{M}{t-l-1})^2}{p(p-n-M-1)}\left\|\boldsymbol{w}_j^* - \boldsymbol{w}_k^*\right\|^2$$

$$+ \sum_{l=0}^{t-2}\left(1 - \frac{n+M}{p}\right)^l \sum_{j=1}^{t-l-1}\frac{\frac{nM}{t-l-1}}{p(p-n-M-1)}\left\|\boldsymbol{w}_j^* - \boldsymbol{w}_{t-l}^*\right\|^2 + \text{noise}_t^{(\text{concurrent})}(\sigma)$$

$$= \left(1 - \frac{n}{p}\right)\left(1 - \frac{n+M}{p}\right)^{t-1}\|\boldsymbol{w}_i^*\|^2$$

$$+ \left\{\left(1 - \frac{n+M}{p}\right)^{t-1}\frac{n}{p} + \sum_{l=0}^{t-2}\left(1 - \frac{n+M}{p}\right)^l\frac{M}{(t-l-1)p}\right\}\|\boldsymbol{w}_1^* - \boldsymbol{w}_i^*\|^2$$

$$+ \sum_{j=2}^{t-1}\left\{\sum_{l=0}^{t-j-1}\left(1 - \frac{n+M}{p}\right)^l\frac{M}{(t-l-1)p} + \left(1 - \frac{n+M}{p}\right)^{t-j}\frac{n}{p}\right\}\left\|\boldsymbol{w}_j^* - \boldsymbol{w}_i^*\right\|^2$$

$$+ \frac{n}{p}\left\|\boldsymbol{w}_t^* - \boldsymbol{w}_i^*\right\|^2$$

$$+ \sum_{l=0}^{t-2}\left(1 - \frac{n+M}{p}\right)^l \sum_{j=1}^{t-l-2}\sum_{k=j+1}^{t-l-1}\frac{(\frac{M}{t-l-1})^2}{p(p-n-M-1)}\left\|\boldsymbol{w}_j^* - \boldsymbol{w}_k^*\right\|^2$$

$$+ \sum_{l=0}^{t-2}\left(1 - \frac{n+M}{p}\right)^l \sum_{j=1}^{t-l-1}\frac{\frac{nM}{t-l-1}}{p(p-n-M-1)}\left\|\boldsymbol{w}_j^* - \boldsymbol{w}_{t-l}^*\right\|^2 + \text{noise}_t^{(\text{concurrent})}(\sigma), \qquad (33)$$

where

$$\text{noise}_t^{(\text{concurrent})}(\sigma) = \left(1 - \frac{n}{p}\right)\left(1 - \frac{n+M}{p}\right)^{t-1}\frac{n\sigma^2}{p-n-1} + \sum_{l=0}^{t-2}\left(1 - \frac{n+M}{p}\right)^l\frac{(n+M)\sigma^2}{p-n-M-1}.$$

By rearranging the terms and substituting $t = T$, we complete the poof for $d_{0T}^{(\text{concurrent})}$ and $d_{ijkT}^{(\text{concurrent})}$. Furthermore, the expressions of $c_i^{(\text{concurrent})}$ and $c_{ijk}^{(\text{concurrent})}$ in Proposition 2 can be extracted from $\mathbb{E}[\mathcal{L}_i(\boldsymbol{w}_t)] - \mathbb{E}[\mathcal{L}_i(\boldsymbol{w})]$ as follows.

$$\left[\mathbb{E}\|\boldsymbol{w}_t - \boldsymbol{w}_i^*\|^2 - \mathbb{E}\|\boldsymbol{w}_i - \boldsymbol{w}_i^*\|^2\right]^{(\text{concurrent})}$$

$$= \left(1 - \frac{n}{p}\right)\left[\left(1 - \frac{n+M}{p}\right)^{t-1} - \left(1 - \frac{n+M}{p}\right)^{i-1}\right]\|\boldsymbol{w}_i^*\|^2$$

$$+ \left\{ \left[ \left(1 - \frac{n+M}{p}\right)^{t-1} - \left(1 - \frac{n+M}{p}\right)^{i-1}\right] \frac{n}{p} + \sum_{l=0}^{t-2} \left(1 - \frac{n+M}{p}\right)^{l} \frac{M}{(t-l-1)p} \right.$$

$$\left. - \sum_{l=0}^{i-2} \left(1 - \frac{n+M}{p}\right)^{l} \frac{M}{(i-l-1)p} \right\} \left\| \boldsymbol{w}_1^* - \boldsymbol{w}_i^* \right\|^2$$

$$+ \sum_{j=i}^{t-1} \left\{ \sum_{l=0}^{t-j-1} \left(1 - \frac{n+M}{p}\right)^{l} \frac{M}{(t-l-1)p} + \left(1 - \frac{n+M}{p}\right)^{t-j} \frac{n}{p} \right\} \left\| \boldsymbol{w}_j^* - \boldsymbol{w}_i^* \right\|^2$$

$$+ \sum_{j=2}^{i-1} \left\{ \sum_{l=0}^{t-j-1} \left(1 - \frac{n+M}{p}\right)^{l} \frac{M}{(t-l-1)p} + \left(1 - \frac{n+M}{p}\right)^{t-j} \frac{n}{p} \right.$$

$$\left. - \sum_{l=0}^{i-j-1} \left(1 - \frac{n+M}{p}\right)^{l} \frac{M}{(i-l-1)p} - \left(1 - \frac{n+M}{p}\right)^{i-j} \frac{n}{p} \right\} \left\| \boldsymbol{w}_j^* - \boldsymbol{w}_i^* \right\|^2$$

$$+ \frac{n}{p} \left\| \boldsymbol{w}_t^* - \boldsymbol{w}_i^* \right\|^2$$

$$\left. \begin{array}{l} + \sum_{l=0}^{t-2} \left(1 - \frac{n+M}{p}\right)^{l} \sum_{j=1}^{t-l-2} \sum_{k=j+1}^{t-l-1} \frac{(\frac{M}{t-l-1})^2}{p(p-n-M-1)} \left\| \boldsymbol{w}_j^* - \boldsymbol{w}_k^* \right\|^2 \\ - \sum_{l=0}^{i-2} \left(1 - \frac{n+M}{p}\right)^{l} \sum_{j=1}^{i-l-2} \sum_{k=j+1}^{i-l-1} \frac{(\frac{M}{i-l-1})^2}{p(p-n-M-1)} \left\| \boldsymbol{w}_j^* - \boldsymbol{w}_k^* \right\|^2 \end{array} \right\} \beta_1$$

$$\left. \begin{array}{l} + \sum_{l=0}^{t-2} \left(1 - \frac{n+M}{p}\right)^{l} \sum_{j=1}^{t-l-1} \frac{\frac{nM}{t-l-1}}{p(p-n-M-1)} \left\| \boldsymbol{w}_j^* - \boldsymbol{w}_{t-l}^* \right\|^2 \\ - \sum_{l=0}^{i-2} \left(1 - \frac{n+M}{p}\right)^{l} \sum_{j=1}^{i-l-1} \frac{\frac{nM}{i-l-1}}{p(p-n-M-1)} \left\| \boldsymbol{w}_j^* - \boldsymbol{w}_{i-l}^* \right\|^2. \end{array} \right\} \beta_2$$

$$+ \text{noise}_t^{(\text{concurrent})}(\sigma) - \text{noise}_i^{(\text{concurrent})}(\sigma) \tag{34}$$

Here, we will show that $\beta_1$ consists of terms $\delta_{j,k} \left\| \boldsymbol{w}_j^* - \boldsymbol{w}_k^* \right\|^2$ with $\delta_{j,k} \geq -\frac{T^2(n+M)M^2}{p^3}$ and $j, k \neq t$ and $\beta_2$ consists of terms $\eta_{j,k} \left\| \boldsymbol{w}_j^* - \boldsymbol{w}_k^* \right\|^2$ with $\eta_{j,k} \geq -\frac{T^2(n+M)nM}{p^3}$ in Appendix E.1.

### B.2 PROOF OF SEQUENTIAL REPLAY IN PROPOSITIONS 1 AND 2

To simplify, we apply the following notations to denote the current data in this subsection: $\boldsymbol{X}_t := \boldsymbol{X}_{t,0}$, $\boldsymbol{Y}_t := \boldsymbol{Y}_{t,0}$ and $\boldsymbol{z}_t := \boldsymbol{z}_{t,0}$.

When $t \geq 2$, the sequence of SGD convergent points $\boldsymbol{w}_t^{(j)}$ is equivalent the sequential optimization problems:

$$\hat{\boldsymbol{w}}_t^{(j)} = \min_{\boldsymbol{w}} \left\| \boldsymbol{w} - \hat{\boldsymbol{w}}_t^{(j-1)} \right\|_2^2 \quad s.t. \quad \boldsymbol{X}_{t,j}^\top \boldsymbol{w} = \boldsymbol{Y}_{t,j}, \;\; j = 0, 1, ..., t-1,$$

where $\hat{\boldsymbol{w}}_t^{(-1)} = \boldsymbol{w}_{t-1}$ and $\boldsymbol{w}_t = \hat{\boldsymbol{w}}_t^{(t-1)}$. Therefore, according to Lemmas 1 to 4, we have:

$$\mathbb{E} \left\| \hat{\boldsymbol{w}}_t^{(j)} - \boldsymbol{w}_i^* \right\|^2 = \mathbb{E} \left\| (\boldsymbol{I} - P_{\boldsymbol{X}_{t,j}})(\hat{\boldsymbol{w}}_t^{(j-1)} - \boldsymbol{w}_i^*) + P_{\boldsymbol{X}_{t,j}}(\boldsymbol{w}_j^* - \boldsymbol{w}_i^*) + \boldsymbol{X}_{t,j}^\dagger \boldsymbol{z}_{t,j} \right\|^2$$

$$= \left(1 - \frac{M}{(t-1)p}\right) \mathbb{E} \left\| \hat{\boldsymbol{w}}_t^{(j-1)} - \boldsymbol{w}_i^* \right\|^2 + \frac{M}{(t-1)p} \left\| \boldsymbol{w}_j^* - \boldsymbol{w}_i^* \right\|^2 + \frac{\frac{M}{(t-1)p}\sigma^2}{p - \frac{M}{(t-1)p} - 1},$$

for $j = 1, 2, ..., t-1$. Also, we have:

$$\mathbb{E} \left\| \hat{\boldsymbol{w}}_t^{(0)} - \boldsymbol{w}_i^* \right\|^2 = \mathbb{E} \left\| (\boldsymbol{I} - P_{\boldsymbol{X}_t})(\boldsymbol{w}_{t-1} - \boldsymbol{w}_i^*) + P_{\boldsymbol{X}_t}(\boldsymbol{w}_t^* - \boldsymbol{w}_i^*) \right\|^2$$

$$= \left(1 - \frac{n}{p}\right) \mathbb{E} \left\| \boldsymbol{w}_{t-1} - \boldsymbol{w}_i^* \right\|^2 + \frac{n}{p} \left\| \boldsymbol{w}_t^* - \boldsymbol{w}_i^* \right\|^2 + \frac{n\sigma^2}{p - n - 1}.$$

By combining the above two equations, we can derive:

$$\mathbb{E}\left\|\boldsymbol{w}_t - \boldsymbol{w}_i^*\right\|^2$$

$$= \left(1 - \frac{M}{(t-1)p}\right)^{t-1} \left(1 - \frac{n}{p}\right) \mathbb{E}\left\|\boldsymbol{w}_{t-1} - \boldsymbol{w}_i^*\right\|^2$$

$$+ \sum_{j=1}^{t-1} \left(1 - \frac{M}{(t-1)p}\right)^{t-j-1} \frac{M}{(t-1)p} \mathbb{E}\left\|\boldsymbol{w}_j^* - \boldsymbol{w}_i^*\right\|^2 + \left(1 - \frac{M}{(t-1)p}\right)^{t-1} \frac{n}{p} \left\|\boldsymbol{w}_t^* - \boldsymbol{w}_i^*\right\|^2$$

$$+ \sum_{j=1}^{t-1} \left(1 - \frac{M}{(t-1)p}\right)^{t-j-1} \frac{\frac{M}{t-1}\sigma^2}{p - \frac{M}{t-1} - 1} + \left(1 - \frac{M}{(t-1)p}\right)^{t-1} \frac{n\sigma^2}{p-n-1}.$$

By applying this process recursively, we obtain the expression of the expected value of the model error $\mathbb{E}[\mathcal{L}_i(\boldsymbol{w}_t)]$ as follows, in we can extract the expressions of $d_{0T}^{(\text{sequential})}$ and $d_{ijkT}^{(\text{sequential})}$:

$$\mathbb{E}\left\|\boldsymbol{w}_t - \boldsymbol{w}_i^*\right\|^2$$

$$= \prod_{l=0}^{t-2}\left[\left(1 - \frac{M}{(t-l-1)p}\right)^{t-l-1}\left(1 - \frac{n}{p}\right)\right]\mathbb{E}\left\|\boldsymbol{w}_1 - \boldsymbol{w}_i^*\right\|^2$$

$$+ \sum_{j=1}^{t-1}\left\{\sum_{l=0}^{t-j-1}\prod_{k=0}^{l-1}\left[\left(1 - \frac{M}{(t-k-1)p}\right)^{t-k-1}\left(1 - \frac{n}{p}\right)\right]\right.$$

$$\left.\cdot\left(1 - \frac{M}{(t-l-1)p}\right)^{t-j-l-1}\frac{M}{(t-l-1)p}\right\}\left\|\boldsymbol{w}_j^* - \boldsymbol{w}_i^*\right\|^2$$

$$+ \sum_{l=0}^{t-2}\prod_{k=0}^{l-1}\left[\left(1 - \frac{M}{(t-k-1)p}\right)^{t-k-1}\left(1 - \frac{n}{p}\right)\right]\left(1 - \frac{M}{(t-l-1)p}\right)^{t-l-1}\frac{n}{p}\left\|\boldsymbol{w}_{t-l}^* - \boldsymbol{w}_i^*\right\|^2$$

$$+ \left(1 - \frac{M}{(t-1)p}\right)^{t-1}\frac{n}{p}\left\|\boldsymbol{w}_t^* - \boldsymbol{w}_i^*\right\|^2 + \text{noise}_t^{(\text{sequential})}(\sigma)$$

$$= \left(1 - \frac{n}{p}\right)\prod_{l=0}^{t-2}\left[\left(1 - \frac{M}{(t-l-1)p}\right)^{t-l-1}\left(1 - \frac{n}{p}\right)\right]\left\|\boldsymbol{w}_i^*\right\|^2$$

$$+ \left\{\frac{n}{p}\prod_{l=0}^{t-2}\left[\left(1 - \frac{M}{(t-l-1)p}\right)^{t-l-1}\left(1 - \frac{n}{p}\right)\right] + \sum_{l=0}^{t-2}\prod_{k=0}^{l-1}\left[\left(1 - \frac{M}{(t-k-1)p}\right)^{t-k-1}\left(1 - \frac{n}{p}\right)\right]\right.$$

$$\left.\cdot\left(1 - \frac{M}{(t-l-1)p}\right)^{t-l-2}\frac{M}{(t-l-1)p}\right\}]\left\|\boldsymbol{w}_1^* - \boldsymbol{w}_i^*\right\|^2$$

$$+ \sum_{j=2}^{t-1}\left\{\sum_{l=0}^{t-j-1}\prod_{k=0}^{l-1}\left[\left(1 - \frac{M}{(t-k-1)p}\right)^{t-k-1}\left(1 - \frac{n}{p}\right)\right]\left(1 - \frac{M}{(t-l-1)p}\right)^{t-j-l-1}\frac{M}{(t-l-1)p}\right.$$

$$\left.+ \prod_{k=0}^{t-j-1}\left[\left(1 - \frac{M}{(t-l-1)p}\right)^{t-k-1}\left(1 - \frac{n}{p}\right)\right]\left(1 - \frac{M}{(j-1)p}\right)^{j-1}\frac{n}{p}\right\}\left\|\boldsymbol{w}_j^* - \boldsymbol{w}_i^*\right\|^2$$

$$+ \left(1 - \frac{M}{(t-1)p}\right)^{t-1}\frac{n}{p}\left\|\boldsymbol{w}_t^* - \boldsymbol{w}_i^*\right\|^2 + \text{noise}_t^{(\text{sequential})}(\sigma), \tag{35}$$

where $\text{noise}_t^{(\text{sequential})}(\sigma) = \sum_{l=0}^{t-2}\prod_{k=0}^{l-1}\left[\left(1 - \frac{M}{(t-k-1)p}\right)^{t-k-1}\left(1 - \frac{n}{p}\right)\right]$

$$\cdot\left[\sum_{j=1}^{t-1}\left(1 - \frac{M}{(t-1)p}\right)^{t-j-1}\frac{\frac{M}{t-1}\sigma^2}{p-\frac{M}{t-1}-1} + \left(1 - \frac{M}{(t-1)p}\right)^{t-1}\frac{n\sigma^2}{p-n-1}\right].$$

Furthermore, the expressions of $c_i^{\text{(sequential)}}$ and $c_{ijk}^{\text{(sequential)}}$ in Proposition 2 can be extracted from the derivation of $\mathbb{E}[\mathcal{L}_i(\boldsymbol{w}_t)] - \mathbb{E}[\mathcal{L}_i(\boldsymbol{w})]$ as follows.

$$\left[\mathbb{E}\left\|\boldsymbol{w}_t - \boldsymbol{w}_i^*\right\|_2^2 - \mathbb{E}\left\|\boldsymbol{w}_i - \boldsymbol{w}_i^*\right\|_2^2\right]^{\text{(sequential)}}$$

$$= \left(1 - \tfrac{n}{p}\right)\left\{\prod_{l=0}^{t-2}\left[\left(1 - \tfrac{M}{(t-l-1)p}\right)^{t-l-1}\left(1 - \tfrac{n}{p}\right)\right]\right.$$

$$\left. - \prod_{l=0}^{i-2}\left[\left(1 - \tfrac{M}{(i-l-1)p}\right)^{i-l-1}\left(1 - \tfrac{n}{p}\right)\right]\right\}\|\boldsymbol{w}_i^*\|^2$$

$$+ \left\{\tfrac{n}{p}\left\{\prod_{l=0}^{t-2}\left[\left(1 - \tfrac{M}{(t-l-1)p}\right)^{t-l-1}\left(1 - \tfrac{n}{p}\right)\right] - \prod_{l=0}^{i-2}\left[\left(1 - \tfrac{M}{(i-l-1)p}\right)^{i-l-1}\left(1 - \tfrac{n}{p}\right)\right]\right\}\right.$$

$$+ \sum_{l=0}^{t-2}\prod_{k=0}^{l-1}\left[\left(1 - \tfrac{M}{(t-k-1)p}\right)^{t-k-1}\left(1 - \tfrac{n}{p}\right)\right]\left(1 - \tfrac{M}{(t-l-1)p}\right)^{t-l-2}\tfrac{M}{(t-l-1)p}$$

$$\left. - \sum_{l=0}^{i-2}\prod_{k=0}^{l-1}\left[\left(1 - \tfrac{M}{(i-k-1)p}\right)^{i-k-1}\left(1 - \tfrac{n}{p}\right)\right]\left(1 - \tfrac{M}{(i-l-1)p}\right)^{i-l-2}\tfrac{M}{(i-l-1)p}\right\}\|\boldsymbol{w}_1^* - \boldsymbol{w}_i^*\|^2$$

$$+ \sum_{j=i}^{t-1}\left\{\sum_{l=0}^{t-j-1}\prod_{k=0}^{l-1}\left[\left(1 - \tfrac{M}{(t-k-1)p}\right)^{t-k-1}\left(1 - \tfrac{n}{p}\right)\right]\left(1 - \tfrac{M}{(t-l-1)p}\right)^{t-j-l-1}\tfrac{M}{(t-l-1)p}\right.$$

$$\left. + \prod_{k=0}^{t-j-1}\left[\left(1 - \tfrac{M}{(t-k-1)p}\right)^{t-k-1}\left(1 - \tfrac{n}{p}\right)\right]\left(1 - \tfrac{M}{(j-1)p}\right)^{j-1}\tfrac{n}{p}\right\}\|\boldsymbol{w}_j^* - \boldsymbol{w}_i^*\|^2$$

$$+ \sum_{j=2}^{i-1}\left\{\sum_{l=0}^{t-j-1}\prod_{k=0}^{l-1}\left[\left(1 - \tfrac{M}{(t-k-1)p}\right)^{t-k-1}\left(1 - \tfrac{n}{p}\right)\right]\left(1 - \tfrac{M}{(t-l-1)p}\right)^{t-j-l-1}\tfrac{M}{(t-l-1)p}\right.$$

$$- \sum_{l=0}^{i-j-1}\prod_{k=0}^{l-1}\left[\left(1 - \tfrac{M}{(i-k-1)p}\right)^{i-k-1}\left(1 - \tfrac{n}{p}\right)\right]\left(1 - \tfrac{M}{(i-l-1)p}\right)^{i-j-l-1}\tfrac{M}{(i-l-1)p}$$

$$+ \prod_{k=0}^{t-j-1}\left[\left(1 - \tfrac{M}{(t-k-1)p}\right)^{t-k-1}\left(1 - \tfrac{n}{p}\right)\right]\left(1 - \tfrac{M}{(j-1)p}\right)^{j-1}\tfrac{n}{p}$$

$$\left. - \prod_{k=0}^{i-j-1}\left[\left(1 - \tfrac{M}{(i-k-1)p}\right)^{i-k-1}\left(1 - \tfrac{n}{p}\right)\right]\left(1 - \tfrac{M}{(j-1)p}\right)^{j-1}\tfrac{n}{p}\right\}\|\boldsymbol{w}_j^* - \boldsymbol{w}_i^*\|^2$$

$$+ \left(1 - \tfrac{M}{(t-1)p}\right)^{t-1}\tfrac{n}{p}\|\boldsymbol{w}_t^* - \boldsymbol{w}_i^*\|^2 + \text{noise}_t^{\text{(sequential)}}(\sigma) - \text{noise}_i^{\text{(sequential)}}(\sigma) \tag{36}$$

### B.3 Proof of Theorem 1

Theorem 1 follows directly from Propositions 1 and 2 and the definitions of $F_T$ and $G_T$.

## C Proof of Theorem 2

In this section, we prove Theorem 2 and provide details about constants $\xi_1, \xi_2, \mu_1, \mu_2$. According to eqs. (33) to (36), we can write forgetting and generalization error when $T = 2$ as follows. For concurrent replay method, we have:

$$F_2^{\text{(concurrent)}} = \mathbb{E}\left\|\boldsymbol{w}_2 - \boldsymbol{w}_1^*\right\|^2 - \mathbb{E}\left\|\boldsymbol{w}_1 - \boldsymbol{w}_1^*\right\|^2$$

$$= \left(-\frac{n+M}{p}\right)\left(1 - \frac{n}{p}\right)\|\boldsymbol{w}_1^*\|^2 + \frac{n}{p}\left(1 + \frac{M}{p-n-M-1}\right)\|\boldsymbol{w}_1^* - \boldsymbol{w}_2^*\|^2$$

$$+ \frac{(n+M)\sigma^2}{p-(n+M)-1} - \frac{n+M}{p}\cdot\frac{n\sigma^2}{p-n-1}. \tag{37}$$

And also, we have

$$G_2^{\text{(concurrent)}} = \frac{1}{2}\left(\mathbb{E}\left\|\boldsymbol{w}_2 - \boldsymbol{w}_1^*\right\|^2 + \mathbb{E}\left\|\boldsymbol{w}_2 - \boldsymbol{w}_2^*\right\|^2\right)$$

$$= \frac{1}{2}\left(1 - \frac{n+M}{p}\right)\left(1 - \frac{n}{p}\right)(\|\boldsymbol{w}_1^*\|^2 + \|\boldsymbol{w}_2^*\|^2)$$

$$+ \frac{1}{2} \left( \frac{2n+M}{p} + \frac{2nM}{p(p-n-M-1)} - \frac{n(n+M)}{p^2} \right) \|\boldsymbol{w}_1^* - \boldsymbol{w}_2^*\|^2$$

$$+ \frac{(n+M)\sigma^2}{p-(n+M)-1} + \left( 1 - \frac{n+M}{p} \right) \frac{n\sigma^2}{p-n-1}. \tag{38}$$

On the other hand, the performance of sequential replay method is:

$$F_2^{\text{sequential}} = \mathbb{E} \|\boldsymbol{w}_2 - \boldsymbol{w}_1^*\|^2 - \mathbb{E} \|\boldsymbol{w}_1 - \boldsymbol{w}_1^*\|^2$$

$$= \left( -\frac{n+M}{p} + \frac{nM}{p^2} \right) \left( 1 - \frac{n}{p} \right) \|\boldsymbol{w}_1^*\|^2 + \left( 1 - \frac{M}{p} \right) \frac{n}{p} \|\boldsymbol{w}_1^* - \boldsymbol{w}_2^*\|^2$$

$$+ \left( 1 - \frac{n+2M}{p} + \frac{nM}{p^2} \right) \frac{n\sigma^2}{p-n-1} + \frac{M\sigma^2}{p-M-1}. \tag{39}$$

And also, we have

$$G_2^{\text{sequential}} = \frac{1}{2} (\mathbb{E} \|\boldsymbol{w}_2 - \boldsymbol{w}_1^*\|^2 + \mathbb{E} \|\boldsymbol{w}_2 - \boldsymbol{w}_2^*\|^2)$$

$$= \frac{1}{2} \left( 1 - \frac{M}{p} \right) \left( 1 - \frac{n}{p} \right)^2 (\|\boldsymbol{w}_1^*\|^2 + \|\boldsymbol{w}_2^*\|^2)$$

$$+ \frac{1}{2} \left( \frac{2n+M}{p} - \frac{n(n+2M)}{p^2} + \frac{n^2M}{p^3} \right) \|\boldsymbol{w}_1^* - \boldsymbol{w}_2^*\|^2$$

$$+ \left( 1 - \frac{M}{p} \right) \left( 2 - \frac{n}{p} \right) \frac{n\sigma^2}{p-n-1} + \frac{M\sigma^2}{p-M-1}. \tag{40}$$

### C.1 PROOF OF FORGETTING IN THEOREM 2

By observing eq. (37) and eq. (39), we see that the forgetting can be expressed as:

$$F_2 = \hat{c}_1 \|\boldsymbol{w}_1^*\|^2 + \hat{c}_2 \|\boldsymbol{w}_1^* - \boldsymbol{w}_2^*\|^2 + \hat{\text{noise}}(\sigma).$$

Before we investigate forgetting, we compare the coefficients $\hat{c}_1, \hat{c}_2$ and term $\hat{\text{noise}}(\sigma)$ as follows, with concurrent replay on the left and sequential replay on the right.

$$\left( -\frac{n+M}{p} \right) \left( 1 - \frac{n}{p} \right) < \left( -\frac{n+M}{p} + \frac{nM}{p^2} \right) \left( 1 - \frac{n}{p} \right)$$

$$\frac{n}{p} \left( 1 + \frac{M}{p-n-M-1} \right) > \left( 1 - \frac{M}{p} \right) \frac{n}{p},$$

$$\frac{(n+M)\sigma^2}{p-(n+M)-1} - \frac{n+M}{p} \cdot \frac{n\sigma^2}{p-n-1} > \left( 1 - \frac{n+2M}{p} + \frac{nM}{p^2} \right) \frac{n\sigma^2}{p-n-1} + \frac{M\sigma^2}{p-M-1}.$$

The comparison implies that $\hat{c}_1^{\text{(concurrent)}} < \hat{c}_1^{\text{(sequential)}}$, $\hat{c}_2^{\text{(concurrent)}} > \hat{c}_2^{\text{(sequential)}}$ and $\hat{\text{noise}}^{\text{(concurrent)}}(\sigma) > \hat{\text{noise}}^{\text{(sequential)}}(\sigma)$. Based on the calculation, we obtain the following conclusion:

$$F_2^{\text{(concurrent)}} > F_2^{\text{(sequential)}} \quad \textbf{if and only if} \quad \xi_1 \|\boldsymbol{w}_1^* - \boldsymbol{w}_2^*\|^2 + \xi_2 \sigma^2 > \|\boldsymbol{w}_1^*\|^2,$$

where $\xi_1 = \frac{\frac{nM}{p} \left( \frac{1}{p-n-M-1} + \frac{1}{p} \right)}{\frac{nM}{p^2} \left( 1 - \frac{n}{p} \right)}$ and $\xi_2 = \frac{\left( \frac{n+M}{p-n-M-1} - \left( 1 - \frac{M}{p} + \frac{nM}{p^2} \right) \frac{n}{p-n-1} - \frac{M}{p-M-1} \right)}{\frac{nM}{p^2} \left( 1 - \frac{n}{p} \right)}$. To make a clearer illustration, we provide the following two special cases.

- If the noise $\sigma$ is 0, and the task similarity is low enough (i.e., $\|\boldsymbol{w}_1^* - \boldsymbol{w}_2^*\|^2$ is large enough), sequential replay achieves a lower forgetting. More specifically, $F_2^{\text{(concurrent)}} \geq F_2^{\text{(sequential)}}$ **if and only if** $\|\boldsymbol{w}_1^* - \boldsymbol{w}_2^*\|^2 \geq \frac{(p-n)(p-n-M-1)}{p^2+p(p-n-M-1)} \|\boldsymbol{w}_1^*\|^2$,

- If task difference $\|\boldsymbol{w}_1^* - \boldsymbol{w}_2^*\|^2 = 0$ and the noise $\sigma$ is large enough, sequential replay achieves a lower forgetting. More specifically, $F_2^{\text{(concurrent)}} \geq F_2^{\text{(sequential)}}$ **if and only if**

$$\sigma \geq \frac{\frac{nM}{p^2} \left( 1 - \frac{n}{p} \right)}{\frac{n+M}{p-n-M-1} - \left( 1 - \frac{M}{p} + \frac{nM}{p^2} \right) \frac{n}{p-n-1} - \frac{M}{p-M-1}} \|\boldsymbol{w}_1^*\|^2.$$

### C.2 Proof of generalization error in Theorem 2

By observing eq. (38) and eq. (40), we see that the generalization error can be expressed as:

$$G_2 = \hat{d}_1(\|\boldsymbol{w}_1^*\|^2 + \|\boldsymbol{w}_2^*\|^2) + \hat{d}_2 \|\boldsymbol{w}_1^* - \boldsymbol{w}_2^*\|^2 + \hat{\text{noise}}(\sigma).$$

Before we compare generalization error, we first observe the coefficients $\hat{d}_1, \hat{d}_2$ and term $\hat{\text{noise}}(\sigma)$ as follows, with concurrent replay on the left and sequential replay on the right.

$$\left(1 - \frac{n+M}{p}\right)\left(1 - \frac{n}{p}\right) < \left(1 - \frac{M}{p}\right)\left(1 - \frac{n}{p}\right)^2$$

$$\frac{2n+M}{p} + \frac{2nM}{p(p-n-M-1)} - \frac{n(n+M)}{p^2} > \frac{2n+M}{p} - \frac{n(n+2M)}{p^2} + \frac{n^2 M}{p^3},$$

$$\frac{(n+M)\sigma^2}{p-(n+M)-1} + \left(1 - \frac{n+M}{p}\right)\frac{n\sigma^2}{p-n-1} > \left(1 - \frac{M}{p}\right)\left(2 - \frac{n}{p}\right)\frac{n\sigma^2}{p-n-1} + \frac{M\sigma^2}{p-M-1},$$

which implies that $\hat{d}_1^{(\text{concurrent})} < \hat{d}_1^{(\text{sequential})}$ and $\hat{d}_2^{(\text{concurrent})} > \hat{d}_2^{(\text{sequential})}$, $\hat{\text{noise}}^{(\text{concurrent})}(\sigma) > \hat{\text{noise}}^{(\text{sequential})}(\sigma)$. Based on our calculation, we obtain the following conclusion. Furthermore, we can obtain the following conclusion:

$$G_2^{(\text{concurrent})} \geq G_2^{(\text{sequential})} \quad \text{if and only if} \quad \mu_1 \|\boldsymbol{w}_1^* - \boldsymbol{w}_2^*\|^2 + \mu_2 \sigma^2 > \|\boldsymbol{w}_1^*\|^2,$$

where $\mu_1 = \frac{\frac{nM}{p}\left(\frac{2}{p-n-M-1} + \frac{1}{p} - \frac{n}{p^2}\right)}{\frac{nM}{p^2}\left(1 - \frac{n}{p}\right)}$ and $\mu_2 = \frac{\frac{n+M}{p-n-M-1} - \left(1 - \frac{M}{p} + \frac{nM}{p^2}\right)\frac{n}{p-n-1} - \frac{M}{p-M-1}}{\frac{nM}{p^2}\left(1 - \frac{n}{p}\right)}$. To provide a clearer illustration, we provide the following two special cases.

- If the noise $\sigma$ is 0, and the task similarity is small enough (i.e., $\|\boldsymbol{w}_1^* - \boldsymbol{w}_2^*\|^2$ is big enough), sequential replay has a smaller generalization error. More specifically, $G_2^{(\text{concurrent})} \geq G_2^{(\text{sequential})}$ **if and only if** $\|\boldsymbol{w}_1^* - \boldsymbol{w}_2^*\|^2 \geq \frac{(p-n)(p-n-M-1)}{2p^2 + (p-n)(p-n-M-1)}\left(\|\boldsymbol{w}_1^*\|^2 + \|\boldsymbol{w}_2^*\|^2\right)$.

- If the task difference $\|\boldsymbol{w}_1^* - \boldsymbol{w}_2^*\|^2 = 0$ and the noise $\sigma$ is big, sequential replay has a smaller generalization error. More specifically, $G_2^{(\text{concurrent})} \geq G_2^{(\text{sequential})}$ **if and only if**

$$\sigma^2 \geq \frac{\frac{nM}{p^2}\left(1 - \frac{n}{p}\right)}{\frac{n+M}{p-n-M-1} - \left(1 - \frac{M}{p} + \frac{nM}{p^2}\right)\frac{n}{p-n-1} - \frac{M}{p-M-1}}\left(\|\boldsymbol{w}_1^*\|^2 + \|\boldsymbol{w}_2^*\|^2\right)$$

## D Comparison between Concurrent and Sequential Replay Methods when $T = 3$

We recall that $M_{2,1} = M$ and $M_{3,1} = M_{3,2} = \frac{M}{2}$ under our equal memory allocation assumption. We assume that $\sigma = 0$. According to eqs. (33) and (34), we write performance of the concurrent replay method when $T = 3$ as follows.

$$F_3^{(\text{concurrent})} = \frac{1}{2}(\mathbb{E}\|\boldsymbol{w}_3 - \boldsymbol{w}_1^*\|^2 - \mathbb{E}\|\boldsymbol{w}_1 - \boldsymbol{w}_1^*\|^2 + \mathbb{E}\|\boldsymbol{w}_3 - \boldsymbol{w}_2^*\|^2 - \mathbb{E}\|\boldsymbol{w}_2 - \boldsymbol{w}_2^*\|^2)$$

$$= \frac{1}{2}\left(-\frac{2(n+M)}{p} + \frac{(n+M)^2}{p^2}\right)\left(1 - \frac{n}{p}\right)\|\boldsymbol{w}_1^*\|^2$$

$$+ \frac{1}{2}\left(-\frac{n+M}{p}\right)\left(1 - \frac{n+M}{p}\right)\left(1 - \frac{n}{p}\right)\|\boldsymbol{w}_2^*\|^2$$

$$+ \frac{1}{2}\left[\left(1 - \frac{2(n+M)}{p}\right)\frac{nM}{p(p-n-M-1)} + \frac{M^2}{2p(p-n-M-1)}\right.$$

$$\left. + \frac{n+M}{p}\left(1 - \frac{n}{p}\right)\left(1 - \frac{n+M}{p}\right)\right]\|\boldsymbol{w}_1^* - \boldsymbol{w}_2^*\|^2$$

$$+ \frac{1}{2}\left[\frac{n}{p} + \frac{nM}{p(p-n-M-1)}\right]\|\boldsymbol{w}_1^* - \boldsymbol{w}_3^*\|^2 + \frac{1}{2}\left[\frac{n}{p} + \frac{nM}{p(p-n-M-1)}\right]\|\boldsymbol{w}_2^* - \boldsymbol{w}_3^*\|^2.$$

(41)

And also, we have

$$G_3^{(\text{concurrent})} = \frac{1}{3}(\mathbb{E}\|\boldsymbol{w}_3 - \boldsymbol{w}_1^*\|^2 + \mathbb{E}\|\boldsymbol{w}_3 - \boldsymbol{w}_2^*\|^2 + \mathbb{E}\|\boldsymbol{w}_3 - \boldsymbol{w}_3^*\|^2)$$

$$= \frac{1}{3}\left(1 - \frac{n+M}{p}\right)^2\left(1 - \frac{n}{p}\right)(\|\boldsymbol{w}_1^*\|^2 + \|\boldsymbol{w}_2^*\|^2 + \|\boldsymbol{w}_3^*\|^2)$$

$$+ \frac{1}{3}\left[\left(3 - \frac{3(n+M)}{p}\right)\frac{nM}{p(p-n-M-1)} + \frac{3M^2}{4p(p-n-M-1)}\right.$$

$$\left.+ \frac{n+M}{p}\left(2 - \frac{3n}{p} - \frac{M}{p} + \frac{n(n+M)}{p^2}\right)\right]\|\boldsymbol{w}_1^* - \boldsymbol{w}_2^*\|^2$$

$$+ \frac{1}{3}\left[\frac{n}{p}\left(2 - \frac{2(n+M)}{p} + \frac{(n+M)^2}{p^2}\right) + \frac{M}{p}\left(1 - \frac{n+M}{p}\right) + \frac{M}{2p}\right.$$

$$\left.+ \frac{3nM}{2p(p-n-M-1)}\right]\|\boldsymbol{w}_1^* - \boldsymbol{w}_3^*\|^2$$

$$+ \frac{1}{3}\left[\frac{n}{p}\left(2 - \frac{n+M}{p}\right) + \frac{M}{2p} + \frac{3nM}{2p(p-n-M-1)}\right]\|\boldsymbol{w}_2^* - \boldsymbol{w}_3^*\|^2.$$

(42)

According to eqs. (35) and (36), the performance of sequential replay when $T = 3$ is provided as follows.

$$F_3^{(\text{sequential})} = \frac{1}{2}(\mathbb{E}\|\boldsymbol{w}_3 - \boldsymbol{w}_1^*\|^2 - \mathbb{E}\|\boldsymbol{w}_1 - \boldsymbol{w}_1^*\|^2 + \mathbb{E}\|\boldsymbol{w}_3 - \boldsymbol{w}_2^*\|^2 - \mathbb{E}\|\boldsymbol{w}_2 - \boldsymbol{w}_2^*\|^2)$$

$$= \frac{1}{2}\left[\left(1 - \frac{n}{p}\right)^3\left(1 - \frac{M}{p}\right)\left(1 - \frac{M}{2p}\right)^2 - \left(1 - \frac{n}{p}\right)\right]\|\boldsymbol{w}_1^*\|^2$$

$$+ \frac{1}{2}\left[\left(1 - \frac{n}{p}\right)^3\left(1 - \frac{M}{p}\right)\left(1 - \frac{M}{2p}\right)^2 - \left(1 - \frac{n}{p}\right)^2\left(1 - \frac{M}{p}\right)\right]\|\boldsymbol{w}_2^*\|^2$$

$$+ \frac{1}{2}\left[\left(1 - \frac{n}{p}\right)\left(1 - \frac{M}{p}\right)\frac{n}{p}\left(\left(1 - \frac{M}{2p}\right)^2\left(2 - \frac{n}{p}\right) - 1\right)\right.$$

$$\left.+ \left(1 - \frac{M}{2p}\right)^2\left(1 - \frac{n}{p}\right)\frac{M}{p} - \frac{M^2}{4p^2}\right]\|\boldsymbol{w}_1^* - \boldsymbol{w}_2^*\|^2$$

$$+ \frac{1}{2}\left(1 - \frac{M}{2p}\right)^2\frac{n}{p}\|\boldsymbol{w}_1^* - \boldsymbol{w}_3^*\|^2 + \left(1 - \frac{M}{2p}\right)^2\frac{n}{p}\|\boldsymbol{w}_2^* - \boldsymbol{w}_3^*\|^2.$$

(43)

And also, we have

$$G_3^{(\text{concurrent})} = \frac{1}{3}(\mathbb{E}\|\boldsymbol{w}_3 - \boldsymbol{w}_1^*\|^2 + \mathbb{E}\|\boldsymbol{w}_3 - \boldsymbol{w}_2^*\|^2 + \mathbb{E}\|\boldsymbol{w}_3 - \boldsymbol{w}_3^*\|^2)$$

$$= \frac{1}{3}\left(1 - \frac{n}{p}\right)^3\left(1 - \frac{M}{p}\right)\left(1 - \frac{M}{2p}\right)^2(\|\boldsymbol{w}_1^*\|^2 + \|\boldsymbol{w}_2^*\|^2 + \|\boldsymbol{w}_3^*\|^2)$$

$$+ \frac{1}{3}\left\{\left(1 - \frac{n}{p}\right)\left(1 - \frac{M}{2p}\right)^2\left[\left(1 - \frac{M}{p}\right)\left(2 - \frac{n}{p}\right)\frac{n}{p} + \frac{M}{p}\right] + \frac{M}{p} - \frac{M^2}{4p^2}\right\}\|\boldsymbol{w}_1^* - \boldsymbol{w}_2^*\|^2$$

$$+ \frac{1}{3}\left[\left(1 - \frac{M}{2p}\right)^2\frac{n}{p} + \left(1 - \frac{M}{2p}\right)^2\left(1 - \frac{n}{p}\right)\frac{M}{p} + \left(1 - \frac{n}{p}\right)\left(1 - \frac{M}{p}\right)\left(1 - \frac{M}{2p}\right)^2\frac{n}{p}\right.$$

$$\left.+ \left(1 - \frac{M}{2p}\right)\frac{M}{2p}\right]\|\boldsymbol{w}_1^* - \boldsymbol{w}_3^*\|^2$$

$$+ \frac{1}{3}\left\{\left(1 - \frac{M}{2p}\right)^2\frac{n}{p}\left[\left(1 - \frac{M}{p}\right)\left(1 - \frac{n}{p}\right) + 1\right] + \frac{M}{2p}\right\}\|\boldsymbol{w}_2^* - \boldsymbol{w}_3^*\|^2.$$

(44)

## D.1 COMPARISON OF FORGETTING WHEN $T = 3$

By observing eq. (41) and eq. (43), we can write forgetting in the same structure for both training methods:

$$F_3 = \frac{1}{2}\hat{c}_1 \left\| \boldsymbol{w}_1^* \right\|^2 + \frac{1}{2}\hat{c}_2 \left\| \boldsymbol{w}_2^* \right\|^2 + \frac{1}{2}\hat{c}_3 \left\| \boldsymbol{w}_1^* - \boldsymbol{w}_2^* \right\|^2 + \frac{1}{2}\hat{c}_4 \left\| \boldsymbol{w}_1^* - \boldsymbol{w}_3^* \right\|^2 + \frac{1}{2}\hat{c}_5 \left\| \boldsymbol{w}_2^* - \boldsymbol{w}_3^* \right\|^2 .$$

By comparing eq. (41) and eq. (43), we have the following conclusions: $1.\hat{c}_1^{(\text{concurrent})} < \hat{c}_1^{(\text{sequential})}$; $2.\hat{c}_2^{(\text{concurrent})} < \hat{c}_2^{(\text{sequential})}$; $3.\hat{c}_3^{(\text{concurrent})} > \hat{c}_3^{(\text{sequential})}$, when $p > \frac{5n+4M}{2}$; $4.\hat{c}_4^{(\text{concurrent})} > \hat{c}_4^{(\text{sequential})}$; $5.\hat{c}_5^{(\text{concurrent})} > \hat{c}_5^{(\text{sequential})}$. The proof of these conclusions is provided as follows.

*Proof.* 1. To prove $\hat{c}_1^{(\text{concurrent})} < \hat{c}_1^{(\text{sequential})}$:

$$\hat{c}_1^{(\text{sequential})} = \left[ \left(1 - \frac{n}{p}\right)^3 \left(1 - \frac{M}{p}\right) \left(1 - \frac{M}{2p}\right)^2 - \left(1 - \frac{n}{p}\right) \right]$$

$$= \left[ \left(1 - \frac{n}{p}\right)^2 \left(1 - \frac{M}{p}\right) \left(1 - \frac{M}{2p}\right)^2 - 1 \right] \left(1 - \frac{n}{p}\right)$$

$$> \left[ \left(1 - \frac{n}{p}\right)^2 \left(1 - \frac{M}{p}\right)^2 - 1 \right] \left(1 - \frac{n}{p}\right)$$

$$> \left[ \left(1 - \frac{n+M}{p}\right)^2 - 1 \right] \left(1 - \frac{n}{p}\right)$$

$$= \hat{c}_1^{(\text{concurrent})} .$$

2. To prove $\hat{c}_2^{(\text{concurrent})} < \hat{c}_2^{(\text{sequential})}$:

$$\hat{c}_2^{(\text{sequential})} = \left[ \left(1 - \frac{n}{p}\right)^3 \left(1 - \frac{M}{p}\right) \left(1 - \frac{M}{2p}\right)^2 - \left(1 - \frac{n}{p}\right)^2 \left(1 - \frac{M}{p}\right) \right]$$

$$> \left[ \left(1 - \frac{n}{p}\right)^3 \left(1 - \frac{M}{p}\right)^2 - \left(1 - \frac{n}{p}\right)^2 \left(1 - \frac{M}{p}\right) \right]$$

$$= \left(1 - \frac{n}{p}\right) \left[ \left(1 - \frac{n}{p}\right) \left(1 - \frac{M}{p}\right) - 1 \right] \left(1 - \frac{n}{p}\right) \left(1 - \frac{M}{p}\right)$$

$$= \left(1 - \frac{n}{p}\right) \left[ \frac{nM}{p^2} - \frac{n+M}{p} \right] \left(1 - \frac{n+M}{p} + \frac{nM}{p^2}\right)$$

$$= \left(1 - \frac{n}{p}\right) \left[ \frac{nM}{p^2} - \frac{n+M}{p} \right] \left(1 - \frac{n+M}{p} + \frac{nM}{p^2}\right)$$

$$= \left(1 - \frac{n}{p}\right) \left[ -\frac{n+M}{p} \right] \left(1 - \frac{n+M}{p}\right)$$

$$\qquad + \left(1 - \frac{n}{p}\right) \frac{nM}{p^2} \left(1 - \frac{2(n+M)}{p} + \frac{nM}{p^2}\right)$$

$$> \left(1 - \frac{n}{p}\right) \left[ -\frac{n+M}{p} \right] \left(1 - \frac{n+M}{p}\right)$$

$$= \hat{c}_2^{(\text{concurrent})} .$$

3. To prove $\hat{c}_3^{(\text{concurrent})} > \hat{c}_3^{(\text{sequential})}$ when $p > \frac{5n+4M}{2}$, we first notice that

$$\hat{c}_3^{(\text{concurrent})} = \left(1 - \frac{2(n+M)}{p}\right) \frac{nM}{p(p-n-M-1)} + \frac{M^2}{2p(p-n-M-1)}$$

$$\qquad + \frac{n+M}{p} \left(1 - \frac{n}{p}\right) \left(1 - \frac{n+M}{p}\right)$$

$$> \left(1 - \frac{2(n+M)}{p}\right)\frac{nM}{p^2} + \frac{M^2}{2p^2} + \frac{n+M}{p}\left(1 - \frac{n}{p}\right)\left(1 - \frac{n+M}{p}\right)$$

$$= \frac{n+M}{p}\left(1 - \frac{n}{p}\right)\left(1 - \frac{M}{p}\right) - \frac{n^2}{p^2} + \frac{n^3 - n^2M - 2nM^2}{p^3}.$$

On the other hand, we have:

$$\hat{c}_3^{(\text{sequential})} = \left(1 - \frac{n}{p}\right)\left(1 - \frac{M}{p}\right)\frac{n}{p}\left(\left(1 - \frac{M}{2p}\right)^2\left(2 - \frac{n}{p}\right) - 1\right)$$

$$+ \left(1 - \frac{M}{2p}\right)^2\left(1 - \frac{n}{p}\right)\frac{M}{p} - \frac{M^2}{4p^2}$$

$$= \left(1 - \frac{n}{p}\right)\left(1 - \frac{M}{p}\right)\frac{n}{p}\left(\left(1 - \frac{M}{p}\right)\left(2 - \frac{n}{p}\right) - 1\right) + \left(1 - \frac{M}{p}\right)\left(1 - \frac{n}{p}\right)\frac{M}{p}$$

$$+ \frac{M^2}{4p^2}\left[\left(2 - \frac{n}{p}\right)\left(1 - \frac{M}{p}\right)\left(1 - \frac{n}{p}\right)\frac{n}{p} + \left(1 - \frac{n}{p}\right)\frac{M}{p} - 1\right]$$

$$< \left(1 - \frac{n}{p}\right)\left(1 - \frac{M}{p}\right)\frac{n}{p}\left(\left(1 - \frac{M}{p}\right)\left(2 - \frac{n}{p}\right) - 1\right)$$

$$+ \left(1 - \frac{M}{p}\right)\left(1 - \frac{n}{p}\right)\frac{M}{p} + \frac{M^2}{4p^2}\left[\frac{2n}{p} + \frac{M}{p} - 1\right]$$

$$\overset{(i)}{<} \left(1 - \frac{n}{p}\right)\left(1 - \frac{M}{p}\right)\frac{n}{p}\left(\left(1 - \frac{M}{p}\right)\left(2 - \frac{n}{p}\right) - 1\right) + \left(1 - \frac{M}{p}\right)\left(1 - \frac{n}{p}\right)\frac{M}{p}$$

$$= \left(1 - \frac{n}{p}\right)\left(1 - \frac{M}{p}\right)\frac{n}{p}\left(1 - \frac{2M}{p} - \frac{n}{p} + \frac{nM}{p^2}\right) + \left(1 - \frac{M}{p}\right)\left(1 - \frac{n}{p}\right)\frac{M}{p}$$

$$= \left(1 - \frac{n}{p}\right)\left(1 - \frac{M}{p}\right)\frac{n+M}{p} + \left(1 - \frac{n}{p}\right)\left(1 - \frac{M}{p}\right)\frac{n}{p}\left(-\frac{n+2M}{p} + \frac{nM}{p^2}\right)$$

$$\overset{(ii)}{<} \left(1 - \frac{n}{p}\right)\left(1 - \frac{M}{p}\right)\frac{n+M}{p} + \left(1 - \frac{n+M}{p}\right)\frac{n}{p}\left(-\frac{n+2M}{p} + \frac{nM}{p^2}\right)$$

$$< \left(1 - \frac{n}{p}\right)\left(1 - \frac{M}{p}\right)\frac{n+M}{p} - \frac{n^2 + 2nM}{p^2} + \frac{n^3 + 4n^2M + 2nM^2}{P^3},$$

where $(i)$ follows from the face that $p > \frac{5n+4M}{2}$ and $(ii)$ follows from the fact that $-\frac{n+2M}{p} + \frac{nM}{p^2} < 0$. Furthermore, under the condition $p > \frac{5n+4M}{2}$, we have:

$$-\frac{n^2 + 2nM}{p^2} + \frac{n^3 + 4n^2M + 2nM^2}{P^3} < -\frac{n^2}{p^2} + \frac{n^3 - n^2M - 2nM^2}{p^3},$$

which completes the proof. 4. To prove $\hat{c}_4^{(\text{concurrent})} > \hat{c}_4^{(\text{sequential})}$:

$$\hat{c}_4^{(\text{concurrent})} = \frac{n}{p} + \frac{nM}{p(p-n-M-1)} > \frac{n}{p} > \left(1 - \frac{M}{2p}\right)^2\frac{n}{p} = \hat{c}_4^{(\text{sequential})}.$$

5. The proof of $\hat{c}_5^{(\text{concurrent})} > \hat{c}_5^{(\text{sequential})}$ is the same as $\hat{c}_4^{(\text{concurrent})} > \hat{c}_4^{(\text{sequential})}$.

### D.2  COMPARISON OF GENERALIZATION ERROR WHEN $T = 3$

By observing eq. (42) and eq. (44), we can write generalization error in the same structure for both training methods:

$$G_3 = \frac{1}{3}\hat{d}_1(\|\boldsymbol{w}_1^*\|^2 + \|\boldsymbol{w}_2^*\|^2 + \|\boldsymbol{w}_3^*\|^2) + \frac{1}{3}\hat{d}_2\|\boldsymbol{w}_1^* - \boldsymbol{w}_2^*\|^2 + \frac{1}{3}\hat{d}_3\|\boldsymbol{w}_1^* - \boldsymbol{w}_3^*\|^2 + \frac{1}{3}\hat{d}_4\|\boldsymbol{w}_2^* - \boldsymbol{w}_3^*\|^2.$$

By comparing eq. (42) and eq. (44), we have the following conclusions: $1. \hat{d}_1^{(\text{concurrent})} < \hat{d}_1^{(\text{sequential})}$; $2. \hat{d}_2^{(\text{concurrent})} > \hat{d}_2^{(\text{sequential})}$ when $p > \frac{4n+3M}{2}$; $3. \hat{d}_3^{(\text{concurrent})} > \hat{d}_3^{(\text{sequential})}$; $4. \hat{d}_4^{(\text{concurrent})} > \hat{d}_4^{(\text{sequential})}$. The proof of these relationships is provided as follows.

1. To prove $\hat{d}_1^{(\text{concurrent})} < \hat{d}_1^{(\text{sequential})}$:

$$\hat{d}_1^{(\text{sequential})} = \left(1 - \frac{n}{p}\right)^3 \left(1 - \frac{M}{p}\right) \left(1 - \frac{M}{2p}\right)^2$$

$$> \left(1 - \frac{n}{p}\right)^3 \left(1 - \frac{M}{p}\right)^2$$

$$> \left(1 - \frac{n+M}{p}\right)^2 \left(1 - \frac{M}{p}\right)$$

$$= \hat{d}_1^{(\text{concurrent})}.$$

2. To prove $\hat{d}_2^{(\text{concurrent})} > \hat{d}_2^{(\text{sequential})}$ when $p > \frac{4n+3M}{2}$, we first consider:

$$\hat{d}_2^{(\text{concurrent})} = \left(3 - \frac{3(n+M)}{p}\right) \frac{nM}{p(p-n-M-1)} + \frac{3M^2}{4p(p-n-M-1)}$$

$$+ \frac{n+M}{p}\left(2 - \frac{3n}{p} - \frac{M}{p} + \frac{n(n+M)}{p^2}\right)$$

$$> \left(3 - \frac{3(n+M)}{p}\right) \frac{nM}{p^2} + \frac{3M^2}{4p^2} + \frac{n+M}{p}\left(2 - \frac{3n}{p} - \frac{M}{p} + \frac{n(n+M)}{p^2}\right)$$

$$> 3\left(1 - \frac{n+M}{p}\right) \frac{nM}{p^2} + \frac{2(n+M)}{p} + \frac{n+M}{p}\left(-\frac{3n}{p} - \frac{n}{p} + \frac{n(n+M)}{p^2}\right)$$

$$= \frac{2(n+M)}{p} - \frac{3n^2 + nM + M^2}{p^2} + \frac{n^3 - n^2M - 2nM^2}{p^3}.$$

On the other hand, we have:

$$\hat{d}_2^{(\text{sequential})} = \left(1 - \frac{n}{p}\right)\left(1 - \frac{M}{2p}\right)^2 \left[\left(1 - \frac{M}{p}\right)\left(2 - \frac{n}{p}\right)\frac{n}{p} + \frac{M}{p}\right] + \frac{M}{p} - \frac{M^2}{4p^2}$$

$$= \left(1 - \frac{n}{p}\right)\left(1 - \frac{M}{p} + \frac{M^2}{4p^2}\right)\left[\left(1 - \frac{M}{p}\right)\left(2 - \frac{n}{p}\right)\frac{n}{p} + \frac{M}{p}\right] + \frac{M}{p} - \frac{M^2}{4p^2}$$

$$= \left(1 - \frac{n}{p}\right)\left(1 - \frac{M}{p}\right)\left[\left(1 - \frac{M}{p}\right)\left(2 - \frac{n}{p}\right)\frac{n}{p} + \frac{M}{p}\right] + \frac{M}{p}$$

$$+ \frac{M^2}{4p^2}\left[\left(1 - \frac{M}{p}\right)\left(2 - \frac{n}{p}\right)\frac{n}{p} + \frac{M}{p}\right] - \frac{M^2}{4p^2}$$

$$< \left(1 - \frac{n}{p}\right)\left(1 - \frac{M}{p}\right)\left[\left(1 - \frac{M}{p}\right)\left(2 - \frac{n}{p}\right)\frac{n}{p} + \frac{M}{p}\right] + \frac{M}{p}$$

$$+ \frac{M^2}{4p^2}\left[\frac{2n}{p} + \frac{M}{p} - 1\right]$$

$$< \left(1 - \frac{n}{p}\right)\left(1 - \frac{M}{p}\right)\left[\left(2 - \frac{n}{p}\right)\frac{n}{p} + \frac{M}{p}\right] + \frac{M}{p}$$

$$= \left(1 - \frac{n}{p}\right)\left(1 - \frac{M}{p}\right)\frac{2n}{p} - \left(1 - \frac{n}{p}\right)\left(1 - \frac{M}{p}\right)\frac{n^2}{p^2} + \frac{2M}{p}$$

$$+ \left(-\frac{n+M}{p} + \frac{nM}{p^2}\right)\frac{M}{p}$$

$$= \frac{2(n+M)}{p} - \frac{3n^2 + 3nM + M^2}{p^2} + \frac{n^3 + 3n^2M + nM^2}{p^3} - \frac{n^3M}{p^4}$$

$$< \frac{2(n+M)}{p} - \frac{3n^2 + 3nM + M^2}{p^2} + \frac{n^3 + 3n^2M + nM^2}{p^3}.$$

Under the condition $p > \frac{4n+3M}{2}$, we have:

$$-\frac{3n^2 + 3nM + M^2}{p^2} + \frac{n^3 + 3n^2M + nM^2}{p^3} < -\frac{3n^2 + nM + M^2}{p^2} + \frac{n^3 - n^2M - 2nM^2}{p^3},$$

which completes the proof.

3. To prove $\hat{d}_3^{\text{(concurrent)}} > \hat{d}_3^{\text{(sequential)}}$, we first have:

$$\hat{d}_3^{\text{(concurrent)}} = \frac{n}{p}\left(2 - \frac{2(n+M)}{p} + \frac{(n+M)^2}{p^2}\right) + \frac{M}{p}\left(1 - \frac{n+M}{p}\right) + \frac{M}{2p}$$

$$+ \frac{3nM}{2p(p-n-M-1)}$$

$$> \frac{n}{p}\left(2 - \frac{2(n+M)}{p} + \frac{(n+M)^2}{p^2}\right) + \frac{M}{p}\left(1 - \frac{n+M}{p}\right) + \frac{M}{2p} + \frac{3nM}{2p^2}.$$

On the other hand, we have:

$$\hat{d}_3^{\text{(sequential)}} = \left(1 - \frac{M}{2p}\right)^2\frac{n}{p} + \left(1 - \frac{M}{2p}\right)^2\left(1 - \frac{n}{p}\right)\frac{M}{p}$$

$$+ \left(1 - \frac{n}{p}\right)^2\left(1 - \frac{M}{p}\right)\left(1 - \frac{M}{2p}\right)^2\frac{n}{p} + \left(1 - \frac{M}{2p}\right)\frac{M}{2p}$$

$$< \left(1 - \frac{M}{2p}\right)^2\frac{n}{p}\left[1 + \left(1 - \frac{n}{p}\right)^2\left(1 - \frac{M}{p}\right)\right]$$

$$+ \left(1 - \frac{n}{p}\right)\left(1 - \frac{M}{p}\right)\frac{M}{p} + \frac{M^3}{4p^3} + \frac{M}{2p}$$

$$< \frac{n}{p}\left(2 - \frac{2n+M}{p} + \frac{n^2+2nM}{p^2}\right) + \frac{n}{p}\left(-\frac{M}{p} + \frac{M^2}{4p^2}\right)$$

$$+ \frac{M}{p}\left(1 - \frac{n+M}{p}\right) + \frac{nM^2}{p} + \frac{M^3}{4p^3} + \frac{M}{2p}$$

$$= \frac{n}{p}\left(2 - \frac{2n+2M}{p} + \frac{n^2+2nM+M^2}{p^2}\right) + \frac{M}{p}\left(1 - \frac{n+M}{p}\right)$$

$$+ \frac{nM^2+M^3}{4p^3} + \frac{M}{2p}$$

$$< \frac{n}{p}\left(2 - \frac{2(n+M)}{p} + \frac{(n+M)^2}{p^2}\right) + \frac{M}{p}\left(1 - \frac{n+M}{p}\right) + \frac{M}{2p} + \frac{3nM}{2p^2}.$$

By combining the above equations, we complete the proof.

4. To prove $\hat{d}_4^{\text{(concurrent)}} > \hat{d}_4^{\text{(sequential)}}$, we first have:

$$\hat{d}_4^{\text{(sequential)}} = \left(1 - \frac{M}{2p}\right)^2\frac{n}{p}\left[\left(1 - \frac{M}{p}\right)\left(1 - \frac{n}{p}\right) + 1\right] + \frac{M}{2p}$$

$$< \frac{n}{p}\left[\left(1 - \frac{M}{p}\right)\left(1 - \frac{n}{p}\right) + 1\right] + \frac{M}{2p}$$

$$= \frac{n}{p}\left[2 - \frac{n+M}{p}\right] + \frac{M}{2p} + \frac{n^2M}{p^3}$$

$$< \frac{n}{p}\left[2 - \frac{n+M}{p}\right] + \frac{M}{2p} + \frac{3nM}{2p(p-n-M-1)}$$

$$< \hat{d}_4^{\text{(concurrent)}}.$$

# E    COMPARISON BETWEEN CONCURRENT AND SEQUENTIAL REPLAY FOR GENERAL $T$

In order to develop the comparison between concurrent and sequential replay methods for general $T$, we need to compare the coefficients in Theorem 1 between concurrent and sequential replay methods. In this section, we assume that $M \geq 2$.

### E.1 COMPARISON OF COEFFICIENTS OF FORGETTING IN THEOREM 1

We first observe the terms $\beta_1$ and $\beta_2$ in eq. (34) before we start to compare the forgetting under different training methods. We separate the term $\beta_1$ into two following parts.

$$
\begin{aligned}
\beta_1 = & \left.\sum_{l=0}^{t-i-1}\left(1-\frac{n+M}{p}\right)^l \sum_{j=1}^{t-l-2}\sum_{k=j+1}^{t-l-1}\frac{(\frac{M}{t-l-1})^2}{p(p-n-M-1)}\left\|\boldsymbol{w}_j^*-\boldsymbol{w}_k^*\right\|^2\right\}\beta_1^+ \\
& +\sum_{l=t-i}^{t-2}\left(1-\frac{n+M}{p}\right)^l \sum_{j=1}^{t-l-2}\sum_{k=j+1}^{t-l-1}\frac{(\frac{M}{t-l-1})^2}{p(p-n-M-1)}\left\|\boldsymbol{w}_j^*-\boldsymbol{w}_k^*\right\|^2 \\
& \left.-\sum_{l=0}^{i-2}\left(1-\frac{n+M}{p}\right)^l \sum_{j=1}^{i-l-2}\sum_{k=j+1}^{i-l-1}\frac{(\frac{M}{i-l-1})^2}{p(p-n-M-1)}\left\|\boldsymbol{w}_j^*-\boldsymbol{w}_k^*\right\|^2\right\}\beta_1^-, \quad (45)
\end{aligned}
$$

where $\beta_1^+$ consists of terms $\delta_{j,k}^+\left\|\boldsymbol{w}_j^*-\boldsymbol{w}_k^*\right\|^2$ with $\delta_{j,k}^+\geq 0$ for $j=[k-1]; k=i,i+1,..,t-1$. Then, we take a closer look at $\beta_1^-$.

$$
\begin{aligned}
\beta_1^- =& \sum_{l=0}^{i-2}\left(1-\frac{n+M}{p}\right)^{t-i+l}\sum_{j=1}^{i-l-2}\sum_{k=j+1}^{i-l-1}\frac{(\frac{M}{i-l-1})^2}{p(p-n-M-1)}\left\|\boldsymbol{w}_j^*-\boldsymbol{w}_k^*\right\|^2 \\
& -\sum_{l=0}^{i-2}\left(1-\frac{n+M}{p}\right)^l\sum_{j=1}^{i-l-2}\sum_{k=j+1}^{i-l-1}\frac{(\frac{M}{i-l-1})^2}{p(p-n-M-1)}\left\|\boldsymbol{w}_j^*-\boldsymbol{w}_k^*\right\|^2 \\
=& \sum_{l=0}^{i-2}\left[\left(1-\frac{n+M}{p}\right)^{t-i}-1\right]\left(1-\frac{n+M}{p}\right)^l\sum_{j=1}^{i-l-2}\sum_{k=j+1}^{i-l-1}\frac{(\frac{M}{i-l-1})^2}{p(p-n-M-1)}\left\|\boldsymbol{w}_j^*-\boldsymbol{w}_k^*\right\|^2 \\
\geq& -\frac{T(n+M)}{p}\sum_{l=0}^{i-2}\sum_{j=1}^{i-l-2}\sum_{k=j+1}^{i-l-1}\frac{(\frac{M}{i-l-1})^2}{p(p-n-M-1)}\left\|\boldsymbol{w}_j^*-\boldsymbol{w}_k^*\right\|^2. \quad (46)
\end{aligned}
$$

This shows that $\beta_1^-$ consists of terms $\delta_{j,k}^-\left\|\boldsymbol{w}_j^*-\boldsymbol{w}_k^*\right\|^2$ with $\delta_{j,k}^-\geq-\frac{T^2(n+M)M^2}{p^3}$ for $j\in[k-1], k\in[i-1]$. Therefore, $\beta_1$ consists of terms $\delta_{j,k}\left\|\boldsymbol{w}_j^*-\boldsymbol{w}_k^*\right\|^2$ where

$$
\delta_{j,k}=\delta_{j,k}^++\delta_{j,k}^-\geq-\frac{T^2(n+M)M^2}{p^3}, \quad (47)
$$

for $j,k\neq t$. By the same argument, we have:

$$
\begin{aligned}
\beta_2 = & \left.\sum_{l=0}^{t-i-1}\left(1-\frac{n+M}{p}\right)^l\sum_{j=1}^{t-l-1}\frac{\frac{nM}{t-l-1}}{p(p-n-M-1)}\left\|\boldsymbol{w}_j^*-\boldsymbol{w}_{t-l}^*\right\|^2\right\}\beta_2^+ \\
& +\sum_{l=t-i}^{t-2}\left(1-\frac{n+M}{p}\right)^l\sum_{j=1}^{t-l-1}\frac{\frac{nM}{t-l-1}}{p(p-n-M-1)}\left\|\boldsymbol{w}_j^*-\boldsymbol{w}_{t-l}^*\right\|^2 \\
& \left.-\sum_{l=0}^{i-2}\left(1-\frac{n+M}{p}\right)^l\sum_{j=1}^{i-l-1}\frac{\frac{nM}{i-l-1}}{p(p-n-M-1)}\left\|\boldsymbol{w}_j^*-\boldsymbol{w}_{i-l}^*\right\|^2\right\}\beta_2^-, \quad (48)
\end{aligned}
$$

where $\beta_2^+$ consists of terms $\eta_{j,k}^+\left\|\boldsymbol{w}_j^*-\boldsymbol{w}_k^*\right\|^2$ with $\eta_{j,k}^+\geq 0$ for $j\in[k-1], k=i+1, i+2,..,t$ and $\beta_2^-$ consists of terms $\eta_{j,k}^-\left\|\boldsymbol{w}_j^*-\boldsymbol{w}_k^*\right\|^2$ with $\eta_{j,k}^-\geq-\frac{T^2(n+M)nM}{p^3}$ for. Therefore, $\beta_2$ consists of terms $\eta_{j,k}\left\|\boldsymbol{w}_j^*-\boldsymbol{w}_k^*\right\|^2$ for $j\in[k-1], k=2,3,..,i$ where

$$
\eta_{j,k}=\eta_{j,k}^++\eta_{j,k}^-\geq-\frac{T^2(n+M)nM}{p^3}. \quad (49)
$$

Now, we compare the coefficients in forgetting in Theorem 1. We first fix the index $i$, meaning that we consider the generalization error on the task $i$.. The proof of $c_i^{(concurrent)}<c_i^{(sequential)}$ follows from Lemma 15 if $p>2T^3(n+M)^2$.

The proof of $c_{ijk}^{\text{(concurrent)}} > c_{ijk}^{\text{(sequential)}}$ are as follows.

1. we prove $c_{i1i}^{\text{(concurrent)}} > c_{i1i}^{\text{(sequential)}}$ if $p > 5T^4(n+M)nM$. We start from $c_{i1i}^{\text{(sequential)}}$. We first upper bound part of the coefficient $c_{i1i}^{\text{(sequential)}}$:

$$\frac{n}{p}\left\{\prod_{l=0}^{t-2}\left[\left(1-\frac{M}{(t-l-1)p}\right)^{t-l-1}\left(1-\frac{n}{p}\right)\right]-\prod_{l=0}^{i-2}\left[\left(1-\frac{M}{(i-l-1)p}\right)^{i-l-1}\left(1-\frac{n}{p}\right)\right]\right\}$$

$$\overset{(i)}{<}\frac{n}{p}\left[\left(1-\frac{n+M}{p}\right)^{t-1}-\left(1-\frac{n+M}{p}\right)^{i-1}\right]+\frac{T^2(n+M)nM}{p^3} \tag{50}$$

where $(i)$ follows from Lemma 16. We then rewrite the rest part of $c_1^{\text{(sequential)}}$ as follows.

$$\sum_{l=0}^{t-2}\prod_{k=0}^{l-1}\left[\left(1-\frac{M}{(t-k-1)p}\right)^{t-k-1}\left(1-\frac{n}{p}\right)\right]\left(1-\frac{M}{(t-l-1)p}\right)^{t-l-2}\frac{M}{(t-l-1)p}$$

$$-\sum_{l=0}^{i-2}\prod_{k=0}^{l-1}\left[\left(1-\frac{M}{(i-k-1)p}\right)^{i-k-1}\left(1-\frac{n}{p}\right)\right]\left(1-\frac{M}{(i-l-1)p}\right)^{i-l-2}\frac{M}{(i-l-1)p}$$

$$=\sum_{l=0}^{t-i-1}\prod_{k=0}^{l-1}\left[\left(1-\frac{M}{(t-k-1)p}\right)^{t-k-1}\left(1-\frac{n}{p}\right)\right]\left(1-\frac{M}{(t-l-1)p}\right)^{t-l-2}\frac{M}{(t-l-1)p}$$

$$+\sum_{l=t-i}^{t-2}\prod_{k=0}^{l-1}\left[\left(1-\frac{M}{(t-k-1)p}\right)^{t-k-1}\left(1-\frac{n}{p}\right)\right]\left(1-\frac{M}{(t-l-1)p}\right)^{t-l-2}\frac{M}{(t-l-1)p}$$

$$-\sum_{l=0}^{i-2}\prod_{k=0}^{l-1}\left[\left(1-\frac{M}{(i-k-1)p}\right)^{i-k-1}\left(1-\frac{n}{p}\right)\right]\left(1-\frac{M}{(i-l-1)p}\right)^{i-l-2}\frac{M}{(i-l-1)p}$$

$$=\sum_{l=0}^{t-i-1}\prod_{k=0}^{l-1}\left[\left(1-\frac{M}{(t-k-1)p}\right)^{t-k-1}\left(1-\frac{n}{p}\right)\right]\left(1-\frac{M}{(t-l-1)p}\right)^{t-l-2}\frac{M}{(t-l-1)p}$$

$$+\sum_{l=0}^{i-2}\prod_{k=0}^{l-i+t-1}\left[\left(1-\frac{M}{(t-k-1)p}\right)^{t-k-1}\left(1-\frac{n}{p}\right)\right]\left(1-\frac{M}{(i-l-1)p}\right)^{i-l-2}\frac{M}{(i-l-1)p}$$

$$-\sum_{l=0}^{i-2}\prod_{k=0}^{l-1}\left[\left(1-\frac{M}{(i-k-1)p}\right)^{i-k-1}\left(1-\frac{n}{p}\right)\right]\left(1-\frac{M}{(i-l-1)p}\right)^{i-l-2}\frac{M}{(i-l-1)p}$$

$$\overset{(i)}{<}\sum_{l=0}^{t-i-1}\left(1-\frac{n+M}{p}\right)^l\frac{M}{(t-l-1)p}-\frac{M}{T^2p^2}+\frac{T^2(n+M)M^2}{p^3}$$

$$+\sum_{l=0}^{i-2}\left[\left(1-\frac{n+M}{p}+\frac{(n+M)M}{p^2}\right)^{l-i+t}-\left(1-\frac{n+M}{p}\right)^l\right]$$

$$\cdot\left(1-\frac{M}{(i-l-1)p}\right)^{i-l-2}\frac{M}{(i-l-1)p}$$

$$\overset{(ii)}{<}\sum_{l=0}^{t-i-1}\left(1-\frac{n+M}{p}\right)^l\frac{M}{(t-l-1)p}-\frac{M}{T^2p^2}+\frac{T^2(n+M)M^2}{p^3}$$

$$+\sum_{l=0}^{i-2}\left[\left(1-\frac{n+M}{p}\right)^{l-i+t}+\frac{T^2(n+M)M}{p^2}-\left(1-\frac{n+M}{p}\right)^l\right]\frac{M}{(i-l-1)p}$$

$$<\sum_{l=0}^{t-1}\left(1-\frac{n+M}{p}\right)^l\frac{M}{(t-l-1)p}-\sum_{l=0}^{i-1}\left(1-\frac{n+M}{p}\right)^l\frac{M}{(i-l-1)p}$$

$$-\frac{M}{T^2p^2}+\frac{2T^2(n+M)M^2}{p^3}, \tag{51}$$

where $(i)$ follows from eq. (58) and lemmas 10 and 11, $(ii)$ follows from Lemma 12 By combining eqs. (50) and (51),

$$c_{i1i}^{(\text{sequential})} < \frac{n}{p}\left[\left(1 - \frac{n+M}{p}\right)^{t-1} - \left(1 - \frac{n+M}{p}\right)^{i-1}\right] + \sum_{l=0}^{t-1}\left(1 - \frac{n+M}{p}\right)^{l}\frac{M}{(t-l-1)p}$$

$$- \sum_{l=0}^{i-1}\left(1 - \frac{n+M}{p}\right)^{l}\frac{M}{(i-l-1)p} + \frac{T^2(n+M)nM}{p^3} - \frac{M}{T^2p^2} + \frac{2T^2(n+M)M^2}{p^3}$$

$$\overset{(i)}{<} \frac{n}{p}\left[\left(1 - \frac{n+M}{p}\right)^{t-1} - \left(1 - \frac{n+M}{p}\right)^{i-1}\right] + \sum_{l=0}^{t-1}\left(1 - \frac{n+M}{p}\right)^{l}\frac{M}{(t-l-1)p}$$

$$- \sum_{l=0}^{i-1}\left(1 - \frac{n+M}{p}\right)^{l}\frac{M}{(i-l-1)p} - \frac{T^2(n+M)M^2}{p^3} - \frac{T^2(n+M)nM}{p^3}$$

$$\overset{(ii)}{\leq} c_{i1i}^{(\text{concurrent})} \tag{52}$$

where $(i)$ follows from the fact that $p > 5T^4(n+M)nM$, $(ii)$ follows from our observation in eqs. (47) and (49).

2. Next, we prove $c_{iji}^{(\text{concurrent})} > c_{iji}^{(\text{sequential})}$ if $p > 5T^4(n+M)nM$, for $j = 2, 3, ..., i-1$. We first notice that $c_{iji}^{(\text{sequential})}$ consists of two parts. We bound the first part by

$$\sum_{l=0}^{t-j-1}\prod_{k=0}^{l-1}\left[\left(1 - \frac{M}{(t-k-1)p}\right)^{t-k-1}\left(1 - \frac{n}{p}\right)\right]\left(1 - \frac{M}{(t-l-1)p}\right)^{t-j-l-1}\frac{M}{(t-l-1)p}$$

$$- \sum_{l=0}^{i-j-1}\prod_{k=0}^{l-1}\left[\left(1 - \frac{M}{(i-k-1)p}\right)^{i-k-1}\left(1 - \frac{n}{p}\right)\right]\left(1 - \frac{M}{(i-l-1)p}\right)^{i-j-l-1}\frac{M}{(i-l-1)p}$$

$$\overset{(i)}{<} \sum_{l=0}^{t-j-1}\left(1 - \frac{n+M}{p}\right)^{l}\frac{M}{(t-l-1)p} - \sum_{l=0}^{i-j-1}\left(1 - \frac{n+M}{p}\right)^{l}\frac{M}{(i-l-1)p}$$

$$- \frac{M}{T^2p^2} + \frac{2T^2(n+M)M^2}{p^3}, \tag{53}$$

For the rest part of $c_{iji}^{(\text{sequential})}$, we have

$$\prod_{k=0}^{t-j-1}\left[\left(1 - \frac{M}{(t-k-1)p}\right)^{t-k-1}\left(1 - \frac{n}{p}\right)\right]\left(1 - \frac{M}{(j-1)p}\right)^{j-1}\frac{n}{p}$$

$$- \prod_{k=0}^{i-j-1}\left[\left(1 - \frac{M}{(i-k-1)p}\right)^{i-k-1}\left(1 - \frac{n}{p}\right)\right]\left(1 - \frac{M}{(j-1)p}\right)^{j-1}\frac{n}{p}$$

$$\overset{(i)}{<} \left\{\left(1 - \frac{n+M}{p}\right)^{i-j-1}\left[\left(1 - \frac{n+M}{p}\right)^{t-i} - 1\right] + \frac{T^2(n+M)M}{p^2}\right\}\left(1 - \frac{M}{(j-1)p}\right)^{j-1}\frac{n}{p}$$

$$< \left\{\left(1 - \frac{n+M}{p}\right)^{i-j-1}\left[\left(1 - \frac{n+M}{p}\right)^{t-i} - 1\right]\right\} + \frac{T^2(n+M)nM}{p^3}, \tag{54}$$

where $(i)$ follows from Lemma 16. By combining eqs. (53) and (54), we have

$$c_{j}^{(\text{sequential})} < \sum_{l=0}^{t-j-1}\left(1 - \frac{n+M}{p}\right)^{l}\frac{M}{(t-l-1)p} - \sum_{l=0}^{i-j-1}\left(1 - \frac{n+M}{p}\right)^{l}\frac{M}{(i-l-1)p}$$

$$+ \left\{\left(1 - \frac{n+M}{p}\right)^{i-1}\left[\left(1 - \frac{n+M}{p}\right)^{t-i} - 1\right]\right\}$$

$$+ \frac{T^2(n+M)nM}{p^3} - \frac{M}{T^2p^2} + \frac{2T^2(n+M)M^2}{p^3}$$

$$\overset{(i)}{<} \sum_{l=0}^{t-j-1} \left(1 - \frac{n+M}{p}\right)^l \frac{M}{(t-l-1)p} - \sum_{l=0}^{i-j-1} \left(1 - \frac{n+M}{p}\right)^l \frac{M}{(i-l-1)p}$$

$$+ \left\{ \left(1 - \frac{n+M}{p}\right)^{i-j-1} \left[\left(1 - \frac{n+M}{p}\right)^{t-i} - 1\right] \right\}$$

$$- \frac{T^2(n+M)M^2}{p^3} - \frac{T^2(n+M)nM}{p^3}$$

$$\overset{(ii)}{\leq} c_{iji}^{(\text{concurrent})}, \tag{55}$$

where $(i)$ follows from the fact that $p > 5T^4(n+M)nM$, $(ii)$ follows from our observation in eqs. (47) and (49).

3. We prove $c_{iji}^{(\text{concurrent})} > c_{iji}^{(\text{sequential})}$ for $j = i, i+1, ..., t-1$ if $p > T^4(n+M)M$. According to the same derivation as eqs. (60) and (62), we have

$$c_{iji}^{(\text{sequential})} < \sum_{l=0}^{t-j-1} \left(1 - \frac{n+M}{p}\right)^l \frac{M}{(t-l-1)p} \left(1 - \frac{n+M}{p}\right)^{t-j} \frac{n}{p}$$

$$- \frac{M}{T^2p^2} + \frac{T^2(n+M)M^2}{p^3}$$

$$< \sum_{l=0}^{t-j-1} \left(1 - \frac{n+M}{p}\right)^l \frac{M}{(t-l-1)p} \left(1 - \frac{n+M}{p}\right)^{t-j} \frac{n}{p}$$

$$- \frac{T^2(n+M)M^2}{p^3} - \frac{T^2(n+M)nM}{p^3}$$

$$\overset{(i)}{\leq} c_{iji}^{(\text{concurrent})},$$

where $(i)$ follows from our observation in eqs. (47) and (49).

4. Last, we prove $c_{iTi}^{(\text{concurrent})} > c_{iTi}^{(\text{sequential})}$ if $p > T^2(n+M)M$. We have:

$$c_{iTi}^{(\text{sequential})} = \left(1 - \frac{M}{(t-1)p}\right)^{t-1} \frac{n}{p} < \left(1 - \frac{M}{(t-1)p}\right) \frac{n}{p} < \frac{n}{p} - \frac{nM}{p^2}$$

$$\overset{(i)}{<} \frac{n}{p} - \frac{T^2(n+M)M^2}{p^3} - \frac{T^2(n+M)nM}{p^3}$$

$$\overset{(ii)}{\leq} c_{iTi}^{(\text{concurrent})}, \tag{56}$$

where $(i)$ follows from the fact that $p > T^2(n+M)M$, $(ii)$ follows from our observation in eqs. (47) and (49).

5. As illustrated in eqs. (45) and (48), we obtain the following conclusions. For $j = [k-1]; k = i, i+1, .., t-1$, we have $c_{ijk}^{(\text{concurrent})} > c_{ijk}^{(\text{sequential})}$, following the fact that $c_{ijk}^{(\text{concurrent})} > 0$ and $c_{ijk}^{(\text{sequential})} = 0$. However, for $j = [k-1]; k \in [i-1]$, we have $c_{ijk}^{(\text{concurrent})} < c_{ijk}^{(\text{sequential})}$, following the fact that $c_{ijk}^{(\text{concurrent})} < 0$ and $c_{ijk}^{(\text{sequential})} = 0$. We note that the impact of these components on forgetting is significantly small under a large $p$, following the fact that the disadvantage terms in sequential replay $\beta_1^-$ and $\beta_2^-$ in eqs. (45) and (48) are of order $\mathcal{O}(\frac{1}{p^3})$, while the advantage of other coefficients is of order $\mathcal{O}(\frac{1}{p^2})$.

### E.2 COMPARISON OF COEFFICIENTS OF GENERALIZATION ERROR IN THEOREM 1

We comparison of coefficients of Generalization error in Theorem 1 as follows. We first fix the index $i$, meaning that we consider the generalization error on the task $i$.

1. We first prove $d_0^{(\text{concurrent})} < d_0^{(\text{sequential})}$. According to Lemma 10, we have:

$$d_{0T}^{(\text{concurrent})} = \left(1 - \frac{n}{p}\right)\left(1 - \frac{n+M}{p}\right)^{t-1}$$

$$< \left(1 - \frac{n}{p}\right) \prod_{l=0}^{t-2}\left[\left(1 - \frac{M}{(t-l-1)p}\right)^{t-l-1}\left(1 - \frac{n}{p}\right)\right]$$

$$= d_{0T}^{(\text{sequential})}$$

2. Now, we prove $d_{i1iT}^{(\text{concurrent})} > d_{i1iT}^{(\text{sequential})}$ if $p > 2T^4(n+M)nM$. We first consider:

$$\frac{n}{p}\prod_{l=0}^{t-2}\left[\left(1 - \frac{M}{(t-l-1)p}\right)^{t-l-1}\left(1 - \frac{n}{p}\right)\right]$$

$$\overset{(i)}{<} \frac{n}{p}\left(1 - \frac{n+M}{p} + \frac{(n+M)M}{p^2}\right)^{t-1}$$

$$\overset{(ii)}{<} \frac{n}{p}\left(1 - \frac{n+M}{p}\right)^{t-1} + \frac{T^2(n+M)nM}{p^3}, \tag{57}$$

where $(i)$ follows from Lemma 11 and $(ii)$ follows from Lemma 12.

We also notice that:

$$\sum_{l=0}^{t-2}\prod_{k=0}^{l-1}\left[\left(1 - \frac{M}{(t-k-1)p}\right)^{t-k-1}\left(1 - \frac{n}{p}\right)\right]\left(1 - \frac{M}{(t-l-1)p}\right)^{t-l-2}\frac{M}{(t-l-1)p}$$

$$= \sum_{l=0}^{t-3}\prod_{k=0}^{l-1}\left[\left(1 - \frac{M}{(t-k-1)p}\right)^{t-k-1}\left(1 - \frac{n}{p}\right)\right]\left(1 - \frac{M}{(t-l-1)p}\right)^{t-l-2}\frac{M}{(t-l-1)p}$$

$$+ \prod_{k=0}^{t-3}\left[\left(1 - \frac{M}{(t-k-1)p}\right)^{t-k-1}\left(1 - \frac{n}{p}\right)\right]\left(1 - \frac{M}{p}\right)\frac{M}{p}$$

$$\overset{(i)}{<} \left(1 - \frac{1}{Tp}\right)\sum_{l=0}^{t-3}\left(1 - \frac{n+M}{p}\right)^{l}\frac{M}{(t-l-1)p}$$

$$+ \left(1 - \frac{n+M}{p} + \frac{(n+M)M}{p^2}\right)^{t-2}\left(1 - \frac{M}{p}\right)\frac{M}{p}$$

$$\overset{(ii)}{<} \left(1 - \frac{1}{Tp}\right)\sum_{l=0}^{t-3}\left(1 - \frac{n+M}{p}\right)^{l}\frac{M}{(t-l-1)p}$$

$$+ \left[\left(1 - \frac{n+M}{p}\right)^{t-2} + \frac{T^2(n+M)M}{p^2}\right]\left(1 - \frac{M}{p}\right)\frac{M}{p}$$

$$< \sum_{l=0}^{t-2}\left(1 - \frac{n+M}{p}\right)^{l}\frac{M}{(t-l-1)p} - \frac{M}{T^2p^2} + \frac{T^2(n+M)M^2}{p^3}, \tag{58}$$

where $(i)$ follows from Lemmas 11 and 14 and $(ii)$ follows from Lemma 12. By combining eqs. (57) and (58), we can conclude:

$$d_{i1iT}^{(\text{sequential})} < \frac{n}{p}\left(1 - \frac{n+M}{p}\right)^{t-1} + \sum_{l=0}^{t-2}\left(1 - \frac{n+M}{p}\right)^{l}\frac{M}{(t-l-1)p}$$

$$+ \frac{T^2(n+M)nM}{p^3} - \frac{M}{T^2p^2} + \frac{T^2(n+M)M^2}{p^3}$$

$$\overset{(i)}{<} \frac{n+M}{p}\left(1 - \frac{n+M}{p}\right)^{t-1} + \sum_{l=0}^{t-2}\left(1 - \frac{n+M}{p}\right)^{l}\frac{M}{(t-l-1)p}$$

$$= d_{i1iT}^{(\text{concurrent})} \tag{59}$$

where $(i)$ follows from the fact that $p > 2T^4(n+M)nM$.

3. Next, we prove $d_{ijiT}^{(\text{concurrent})} > d_{ijiT}^{(\text{sequential})}$ if $p > T^4(n+M)M$, for $j = 2, 3, ..., t-1$. We first have:

$$\sum_{l=0}^{t-j-1} \prod_{k=0}^{l-1} \left[ \left(1 - \frac{M}{(t-k-1)p}\right)^{t-k-1} \left(1 - \frac{n}{p}\right) \right] \left(1 - \frac{M}{(t-l-1)p}\right)^{t-j-l-1} \frac{M}{(t-l-1)p}$$

$$= \sum_{l=0}^{t-j-2} \prod_{k=0}^{l-1} \left[ \left(1 - \frac{M}{(t-k-1)p}\right)^{t-k-1} \left(1 - \frac{n}{p}\right) \right] \left(1 - \frac{M}{(t-l-1)p}\right)^{t-j-l-1} \frac{M}{(t-l-1)p}$$

$$+ \prod_{k=0}^{t-j-2} \left[ \left(1 - \frac{M}{(t-k-1)p}\right)^{t-k-1} \left(1 - \frac{n}{p}\right) \right] \frac{M}{jp}$$

$$\overset{(i)}{<} \left(1 - \frac{1}{Tp}\right) \sum_{l=0}^{t-j-2} \left(1 - \frac{n+M}{p}\right)^{l} \frac{M}{(t-l-1)p} + \left(1 - \frac{n+M}{p} + \frac{(n+M)M}{p^2}\right)^{t-j-1} \frac{M}{jp}$$

$$\overset{(ii)}{<} \left(1 - \frac{1}{Tp}\right) \sum_{l=0}^{t-j-2} \left(1 - \frac{n+M}{p}\right)^{l} \frac{M}{(t-l-1)p} + \left(1 - \frac{n+M}{p}\right)^{t-j-1} \frac{M}{jp} + \frac{T^2(n+M)M^2}{jp^3}$$

$$< \sum_{l=0}^{t-j-1} \left(1 - \frac{n+M}{p}\right)^{l} \frac{M}{(t-l-1)p} - \frac{M}{T^2 p^2} + \frac{T^2(n+M)M^2}{p^3} \tag{60}$$

where $(i)$ follows from Lemmas 11 and 14, $(ii)$ follows Lemma 12. Therefore, if $p > T^4(n+M)M$, we have:

$$\sum_{l=0}^{t-j-1} \prod_{k=0}^{l-1} \left[ \left(1 - \frac{M}{(t-k-1)p}\right)^{t-k-1} \left(1 - \frac{n}{p}\right) \right] \left(1 - \frac{M}{(t-l-1)p}\right)^{t-j-l-1} \frac{M}{(t-l-1)p}$$

$$< \sum_{l=0}^{t-j-1} \left(1 - \frac{n+M}{p}\right)^{l} \frac{M}{(t-l-1)p}. \tag{61}$$

Furthermore, we have:

$$\prod_{k=0}^{t-j-1} \left[ \left(1 - \frac{M}{(t-l-1)p}\right)^{t-k-1} \left(1 - \frac{n}{p}\right) \right] \left(1 - \frac{M}{(j-1)p}\right)^{j-1} \frac{n}{p}$$

$$\overset{(i)}{<} \left(1 - \frac{n+M}{p}\right)^{t-j} \frac{n}{p} \tag{62}$$

where $(i)$ follows from Lemmas 11 and 14. Therefore, by combining eqs. (61) and (62), we have:

$$d_{ijiT}^{(\text{sequential})} < \sum_{l=0}^{t-j-1} \left(1 - \frac{n+M}{p}\right)^{l} \frac{M}{(t-l-1)p} + \left(1 - \frac{n+M}{p}\right)^{t-j} \frac{n}{p} \leq d_{ijiT}^{(\text{concurrent})}. \tag{63}$$

4. Last, we prove $d_{iTiT}^{(\text{concurrent})} > d_{iTiT}^{(\text{sequential})}$. The proof is straightforward:

$$d_{iTiT}^{(\text{sequential})} = \left(1 - \frac{M}{(t-1)p}\right)^{t-1} \frac{n}{p} < \frac{n}{p} \leq d_{iTiT}^{(\text{concurrent})}.$$

5. Moreover, for the other choices of $j, k$ we have $d_{iTiT}^{(\text{concurrent})} \geq 0$ and $d_{iTiT}^{(\text{sequential})} = 0$.

# F  PROOF OF THEOREM 3

Now, we provide a particular example in which sequential replay has less forgetting than concurrent replay. Since $F_T = \frac{1}{T-1} \sum_{i=1}^{T-1} (\mathcal{L}_i(\boldsymbol{w}_T) - \mathcal{L}_i(\boldsymbol{w}_i))$, we focus on proving

$$[\mathcal{L}_i(\boldsymbol{w}_T) - \mathcal{L}_i(\boldsymbol{w}_i)]^{(\text{concurrent})} > [\mathcal{L}_i(\boldsymbol{w}_T) - \mathcal{L}_i(\boldsymbol{w}_i)]^{(\text{sequential})}$$

if $p > 2T^2(n+M)nM$ for each $i \in [T-1]$, which leads to the final conclusion. Since $\boldsymbol{w}_i^*$ are orthonormal, we have $\|\boldsymbol{w}_i^*\|^2 = 1$ and $\|\boldsymbol{w}_i^* - \boldsymbol{w}_j^*\|^2 = 2$ for $i \neq j$. Now we consider when $t = T$. Recall the discussion about $\beta_2$ in eq. (48). Then, we consider

$$
\begin{aligned}
2\beta_2^+ &= \sum_{l=0}^{T-i-1} \left(1 - \frac{n+M}{p}\right)^l \frac{2nM}{p(p-n-M-1)} \\
&= \frac{2nM}{p(p-n-M-1)} \cdot \frac{[1 - (1-\frac{n+M}{p})^{T-i}]}{1 - (1-\frac{n+M}{p})} \\
&> \frac{2nM}{p^2} \cdot \frac{-\sum_{k=1}^{T-i} \binom{T-i}{k}(-\frac{n+M}{p})^k}{\frac{n+M}{p}}
\end{aligned}
\tag{64}
$$

We note that for any $k \in [3, T-i-1]$ and $k$ is odd, we have

$$
\begin{aligned}
&\binom{T-i}{k}\left(-\frac{n+M}{p}\right)^k + \binom{T-i}{k+1}\left(-\frac{n+M}{p}\right)^{k+1} \\
&= \frac{(T-i)!}{k!(T-i-k-1)!}\left(-\frac{n+M}{p}\right)^k \left[\frac{1}{T-i-k} + \frac{1}{k+1}\left(-\frac{n+M}{p}\right)\right] \\
&< \frac{(T-i)!}{k!(T-i-k-1)!}\left(-\frac{n+M}{p}\right)^k \left[\frac{1}{T} - \frac{n+M}{p}\right] \\
&\overset{(i)}{<} 0,
\end{aligned}
$$

where $(i)$ follows from the fact that $p > T(n+M)$. By simply discussing when $T-i$ is odd or even, we can have

$$
\begin{aligned}
-\sum_{k=1}^{T-i}\binom{T-i}{k}\left(-\frac{n+M}{p}\right)^k &> -\binom{T-i}{1}\left(-\frac{n+M}{p}\right) - \binom{T-i}{2}\left(-\frac{n+M}{p}\right)^2 \\
&= \frac{(T-i)(n+M)}{p} - \frac{(T-i)(T-i-1)(n+M)^2}{2p^2}.
\end{aligned}
$$

By substituting the above equation into eq. (64), we can have

$$
\begin{aligned}
2\beta_2^+ &> \frac{2nM}{p(n+M)} \cdot \left[\frac{(T-i)(n+M)}{p} - \frac{(T-i)(T-i-1)(n+M)^2}{2p^2}\right] \\
&= \frac{2(T-i)nM}{p^2} - \frac{(T-i)(T-i-1)(n+M)nM}{p^3} \\
&\overset{(i)}{\geq} \frac{(T-i)(n+M)M}{p^2} + \frac{M}{p^2} - \frac{T^2(n+M)nM}{p^3}
\end{aligned}
\tag{65}
$$

where $(i)$ follows from the fact that $n \geq M+1$. Now, we can conclude:

$$
[\mathcal{L}_i(\boldsymbol{w}_T) - \mathcal{L}_i(\boldsymbol{w}_i)]^{\text{(concurrent)}}
$$

$$
= c_0^{\text{(concurrent)}} + 2\sum_{j=1}^{T} c_j^{\text{(concurrent)}}
$$

$$
\overset{(i)}{>} \left(1 - \frac{n}{p}\right)\left[\left(1 - \frac{n+M}{p}\right)^{T-1} - \left(1 - \frac{n+M}{p}\right)^{i-1}\right] + 2\sum_{j=1}^{T} c_j^{\text{(sequential)}} + 2\beta_1^+ + 2\beta_2^+
$$

$$
\geq \left(1 - \frac{n}{p}\right)\left[\left(1 - \frac{n+M}{p}\right)^{T-1} - \left(1 - \frac{n+M}{p}\right)^{i-1}\right] + 2\sum_{j=1}^{T} c_j^{\text{(sequential)}} + 2\beta_2^+
\tag{66}
$$

where $(i)$ follows from eqs. (52), (55) and (56). On the other hand, we have:

$$
[\mathcal{L}_i(\boldsymbol{w}_T) - \mathcal{L}_i(\boldsymbol{w}_i)]^{\text{(sequential)}}
$$

$$\overset{(i)}{<} \left(1 - \frac{n}{p}\right)\left[\left(1 - \frac{n+M}{p}\right)^{T-1} - \left(1 - \frac{n+M}{p}\right)^{i-1}\right] + 2\sum_{j=1}^{T} c_j^{(\text{sequential})}$$

$$+ \frac{(T-i)(n+M)M}{p^2} + \frac{T^3(n+M)^2M^2}{p^4}, \tag{67}$$

where $(i)$ follows from Lemma 17. By combining eqs. (65) to (67) and the fact that $p > 2T^2(n + M)nM$, we have

$$[\mathcal{L}_i(\boldsymbol{w}_T) - \mathcal{L}_i(\boldsymbol{w}_i)]^{(\text{concurrent})} > [\mathcal{L}_i(\boldsymbol{w}_T) - \mathcal{L}_i(\boldsymbol{w}_i)]^{(\text{sequential})},$$

which completes the proof.

Now, we provide a particular example in which sequential replay achieves a lower generalization error, as presented in Theorem 3. Since $G_T = \frac{1}{T}\sum_{i=1}^{T} \mathcal{L}_i(\boldsymbol{w}_T)$, we focus on proving $\mathcal{L}_i^{(\text{concurrent})}(\boldsymbol{w}_T) > \mathcal{L}_i^{(\text{sequential})}(\boldsymbol{w}_T)$ if $p > 2T^4(n + M + 1)^2M$ for each $i \in [T]$, which leads to the final conclusion. Since $\boldsymbol{w}_i^*$ are orthonormal, we have $\|\boldsymbol{w}_i^*\|^2 = 1$ and $\|\boldsymbol{w}_i^* - \boldsymbol{w}_j^*\|^2 = 2$ for $i \neq j$. We first consider

$$\sum_{l=0}^{t-2}\left(1 - \frac{n+M}{p}\right)^l \sum_{j=1}^{t-l-1} \frac{\frac{nM}{t-l-1}}{p(p-n-M-1)}\|\boldsymbol{w}_j^* - \boldsymbol{w}_{t-l}^*\|^2$$

$$= \sum_{l=0}^{T-2}\left(1 - \frac{n+M}{p}\right)^l \sum_{j=1}^{T-l-1} \frac{\frac{2nM}{T-l-1}}{p^2}$$

$$> (T-1)\left(1 - \frac{n+M}{p}\right)^T \frac{2nM}{p^2}$$

$$> \left(1 - \frac{T(n+M)}{p}\right)\frac{2(T-1)nM}{p^2}$$

$$\overset{(i)}{\geq} \left(1 - \frac{T(n+M)}{p}\right)\frac{(T-1)(n+M+1)M}{p^2}, \tag{68}$$

where $(i)$ follows from the fact that $n \geq M + 1$. Therefore, by combining eqs. (33) and (68), we have:

$$\mathcal{L}_i^{(\text{concurrent})}(\boldsymbol{w}_T) > \left(1 - \frac{n}{p}\right)\left(1 - \frac{n+M}{p}\right)^{T-1}$$

$$+ 2\left\{\left(1 - \frac{n+M}{p}\right)^{T-1}\frac{n}{p} + \sum_{l=0}^{T-2}\left(1 - \frac{n+M}{p}\right)^l \frac{M}{(T-l-1)p}\right\}$$

$$+ 2\sum_{j=2}^{T-1}\left\{\sum_{l=0}^{T-j-1}\left(1 - \frac{n+M}{p}\right)^l \frac{M}{(T-l-1)p} + \left(1 - \frac{n+M}{p}\right)^{T-j}\frac{n}{p}\right\} + \frac{2n}{p}$$

$$+ \left(1 - \frac{T(n+M)}{p}\right)\frac{(T-1)(n+M+1)M}{p^2}. \tag{69}$$

On the other hand, we have:

$$\mathcal{L}_i^{(\text{sequential})}(\boldsymbol{w}_T) \overset{(i)}{<} \left(1 - \frac{n}{p}\right)\left(1 - \frac{n+M}{p} + \frac{(n+M)M}{p^2}\right)^{T-1}$$

$$+ 2\left\{\left(1 - \frac{n+M}{p}\right)^{T-1}\frac{n}{p} + \sum_{l=0}^{T-2}\left(1 - \frac{n+M}{p}\right)^l \frac{M}{(T-l-1)p}\right\}$$

$$+ 2\sum_{j=2}^{T-1}\left\{\sum_{l=0}^{T-j-1}\left(1 - \frac{n+M}{p}\right)^l \frac{M}{(T-l-1)p} + \left(1 - \frac{n+M}{p}\right)^{T-j}\frac{n}{p}\right\} + \frac{2n}{p}$$

$$\overset{(ii)}{<} \left(1 - \frac{n}{p}\right)\left(1 - \frac{n+M}{p}\right)^{T-1}$$

$$+ 2\left\{\left(1 - \frac{n+M}{p}\right)^{T-1}\frac{n}{p} + \sum_{l=0}^{T-2}\left(1 - \frac{n+M}{p}\right)^{l}\frac{M}{(T-l-1)p}\right\}$$

$$+ 2\sum_{j=2}^{T-1}\left\{\sum_{l=0}^{T-j-1}\left(1 - \frac{n+M}{p}\right)^{l}\frac{M}{(T-l-1)p} + \left(1 - \frac{n+M}{p}\right)^{T-j}\frac{n}{p}\right\} + \frac{2n}{p}$$

$$+ \left(\frac{(T-1)(n+M)M}{p^2} + \frac{T^3(n+M)^2M^2}{2p^4}\right) \tag{70}$$

where $(i)$ follows from Lemma 11 and eqs. (59) and (63), $(ii)$ follows from Lemma 13 and the fact that $1 - \frac{n}{p} < 1$. To build the relationship between eqs. (69) and (70), we have:

$$\left(1 - \frac{T(n+M)}{p}\right)\frac{(T-1)(n+M+1)M}{p^2} - \left(\frac{(T-1)(n+M)M}{p^2} + \frac{T^3(n+M)^2M^2}{2p^4}\right)$$

$$= \frac{(T-1)M}{p^2} - \frac{T(T-1)(n+M)(n+M+1)M}{p^3} - \frac{T^3(n+M)^2M^2}{2p^4}$$

$$\overset{(i)}{>} 0 \tag{71}$$

where $(i)$ follows from the fact that $p > 2T^2(n+M+1)^2M$. By combining eqs. (69) to (71), we can conclude: $\mathcal{L}_i^{(\text{concurrent})}(\boldsymbol{w}_T) > \mathcal{L}_i^{(\text{sequential})}(\boldsymbol{w}_T)$.

# G  EXPERIMENT DETAILS

**Dataset.** We evaluate our Hybrid Replay on CIFAR-100 (Krizhevsky et al. (2009)), a real-world dataset for image classification. It's composed of a total of 100 different classes, each containing 500 non-overlapping training images and 100 testing images. In line with prior works Guo et al. (2022) and Sun et al. (2022), we randomly split the original dataset into 10 tasks under a task-incremental setup, each containing 10 non-overlapping classes.

**Implementation Details.** For training on CIFAR-100, we employ a non-pretrained ResNet-18 as our DNN backbone. Following Van de Ven et al. (2022), we adopt a multi-headed output layer such that each task is assigned its own output layer, consistent with the typical Task Incremental CL setup. During supervised training, we explicitly provide the task identifier (ranging from 0 to 9) alongside the image-label pairs as additional input to the model. For simplicity, we use a reservoir sampling strategy to construct the replay buffer. Our replay buffer size is 50 per class. Other than the image corruption, we didn't apply any data augmentation prior to training.

For all experiments on *Concurrent Replay*, we use the SGD (Stochastic Gradient Descent) optimizer for 30 epochs per task, with a minibatch size of 128, momentum of 0.9, weight decay of $1e^{-4}$, and an initial learning rate of 0.05 that is reduced by a factor of 0.1 after 20 epochs.

For all experiments on *Sequential Replay*, we use the SGD optimizer for 30 epochs per task, with a minibatch size of 64, momentum of 0.9, weight decay of $1e^{-3}$, and an initial learning rate of 0.001 that is reduced by a factor of 0.1 after each 12 epochs. We slightly adjust these training parameters for hybrid training due to the relatively smaller number of trained images which increases the risk of overfitting.

**Task Corruption.** For experiments described in Section 6.2, we control the similarity level of the dataset by applying data corruption to different number of tasks. We provide a list of sample images under different image corruption schemes in fig. 3. For the scenario "Original Dataset", we don't apply any image corruption. For the scenario "1 Corruption", we apply the Glass corruption on $\mathcal{T}_1$. For the scenario "2 Corruption", we apply Glass corruption on $\mathcal{T}_1$, and rotational color swaping on $\mathcal{T}_2$. For the scenario "3 Corruption", we apply Glass corruption on $\mathcal{T}_1$, rotational color swaping on $\mathcal{T}_3$, and elastic pixelation on $\mathcal{T}_5$.

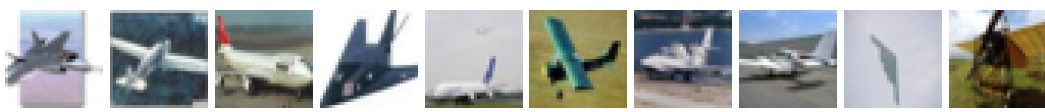

(a) Sample images without corruption.

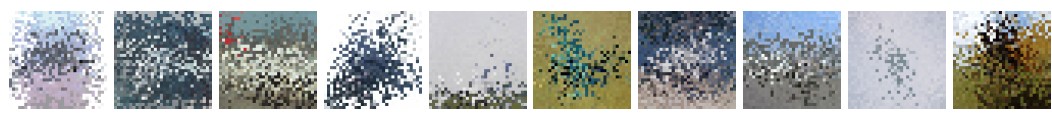

(b) Glass Corruption: the images are transformed to simulate the effect of viewing through frosted glass, inducing localized blurring and pixel displacement.

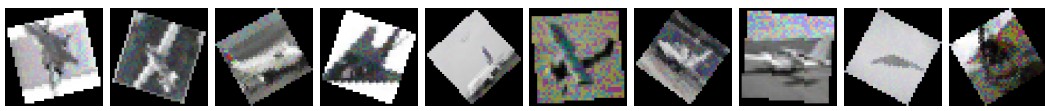

(c) Color-swapping and Rotation Corruption: the images are randomly rotated by arbitrary angles, and a subset of pixels undergoes random permutation of RGB channels.

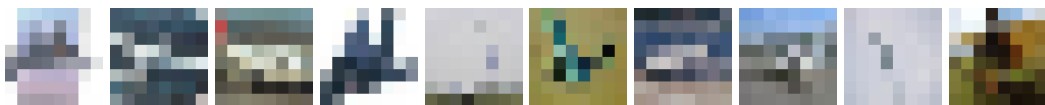

(d) Elastic and Pixelate Image Corruption: the images are subjected to smooth, non-linear spatial deformations followed by pixelation, resulting in a low-resolution appearance.

Figure 3: Sample images for demonstrating the corruption schemes listed.

