# OpenReview forum: "Replay concurrently or sequentially?  A theoretical perspective on replay in continual learning"
_ICLR.cc/2025/Conference — ICLR 2025 Conference Withdrawn Submission_

### Official Review · Reviewer_7Sfj · 2024-10-26

**Soundness:** 1
**Presentation:** 2
**Contribution:** 3
**Rating:** 3
**Confidence:** 4

**Summary:**

The paper first deals with replay protocols in continual learning, suggesting that sequential replay may in some cases be more appropriate than the common concurrent replay. To justify this, it performs a theoretical analysis using overparameterized linear models that proves, under certain conditions, that sequential replay can indeed outperform concurrent replay, at least for the two-task setting. Generalizations to T>2 tasks are given as well. An experimental study based on DNNs is then presented using the CIFAR benchmark, which suggests that a hybrid strategy of concurrent and sequential replay, based on task similarity, works moderately better than just concurrent replay.

**Strengths:**

The paper presents an original and interesting idea for performing continual learning that is certainly intuitive. The theoretical analysis on linear models is convincing, and the proposed hybrid strategy could make sense. An experiment using real-world data and DNNs is performed as well, showing moderate improvements.

**Weaknesses:**

-  It is unclear whether the theoretical analysis will carry over to DNNs and other non-linear models, and no insight is provided on this.
- The theoretical guarantees for more than two tasks seem pretty restrictive, even when working with overparameterized linear models only
- The theoretical analysis and the experiments are only very loosely related. E.g., theory is about regression, experiments are about classification.
- The experiment does not report significant improvement, and it seems numbers are based on one run only, and could therefore be coincidence.
- No ideas on how to actually measure task similarity are put forward, let alone implemented.

More comments (not part of the overall decision, but came up during reading):
l103: GEM + A-GEM should be mentioned in related work for constraint-based methods
l164: t ∈ [T ] is a strange notation, I take it to mean {1,2,...,T} but it quite seems unusual to me
eqn (5): something is wrong here: should it not be \frac 1 {T-1} \sum_i \mathcal L_i(w_i) - L_i(w_T) ?
eqn (5): another common definition of forgetting is using max_i instead of sum_i in this expression, please justify your choice
Theorem 1: This is interesting, but you should motivate very well how that generalizes to DNNs! In DNNs, we work with classification, and these linear models are regression models. And DNNs are strongly non-linear, so there is actually nothing that can be transferred here. At least that what is seems to me...
Theorem 3: The assumption that p = O(T^4 n^2 M^2) is rather restrictive, the number of features depending on the fourth power of T in particular. Please comment on how helpful this is in practice!
6.2 This setting is not well-described, especially the role of corruption
6.2 The reported performance differences are not significant, and are not averaged over runs, and do not contain standard deviations. So they could be just random fluctuations...
6.2 For sequential replay, I feel you should at least propose an idea on how to measure task similarity, keeping in mind that you can only measure this on data retained in the replay buffer.

**Questions:**

- Abstract: where does this insight about human learning come from? Is there evidence from psychology that you could cite?
- eqn (3): this is a regression task, why not consider a linear model that leads to classification tasks, for better alignment with experiments?
- general: what is the rationale between doing theory on linear models and generalizing results to DNNs? Are there any results that would allow this, maybe at least in principle or in approximation?
- 6.2 Why are you using a task-incremental and not a class-incremental setting? Task-incremental CL produces in general much better accuracies than class-incremental CL, that could be misleading when comparing results ...
- 6.2 Please give justifications  on why you are using CIFAR at all. This experiment seems very far removed from common CL benchmarks on CIFAR, so why not use a simpler one?

---

### Official Review · Reviewer_sayA · 2024-10-30

**Soundness:** 2
**Presentation:** 2
**Contribution:** 2
**Rating:** 5
**Confidence:** 3

**Summary:**

The paper provides a theoretical analysis of replay methods for continual learning using linear regression models. The novel theory highlights the (somewhat surprising) role of task similarity in the choice between concurrent replay (the standard method) and sequential replay.

**Strengths:**

- Although replay-based methods are the state-of-the-art in continual learning, there is not much theoretical analysis about them. As the paper points out, it is quite challenging to study replay theoretically, and the paper provides a first step in this direction.
- The theoretical result that sequential replay may be better is novel and interesting.

**Weaknesses:**

**Writing**: while I understand that the derivations must be relegated to the Appendix, the paper could do a better job explaining the results in the main text. For example, I believe many terms in Prop.1 are undefined.

**Memory size**: it would be useful if the paper spent more time describing how the relationship between sequential and concurrent replay changes with the memory size.

**Memory freshness assumption**:
- This is the biggest limitation of the paper. Assuming that the memory is fresh (new samples) at each step is a bit unreasonable. In most CL settings, this would not be possible since all the examples will be used in the first step and the method cannot access past data.
- Memory freshness also sidesteps the main issue with replay: buffer overfitting. Any measure about generalization and forgetting will be highly incorrect if it ignores this factor.
- More in general, it is unclear to me whether linear regressors are a good model to formally analyze replay-based methods.

**Algorithm (and limitations of sequential replay)**
- (Algo. 1) assumes new memory samples at each step.
- The replay method used is suboptimal. Usually, sampling minibatches from memory and new data is better than sampling from the union of the two datasets, since the two datasets are unbalanced (replay may be much smaller, as is the case in these experiments).
- for small buffer sizes, sequential replay will train on a small dataset, which may increase overfitting. This is not an issue in sequential replay, where minibatches can balance old and new data appropriately.

**Experiments**: experimental results are quite limited.
- theoretical results (Fig.2) do not show what happens with different memory sizes)
- (Tab. 1)
	- I fail to understand the motivation behind the training schedule (hybrid replay only at task 5).
	- I expected more extensive experiments, with multiple datasets and multiple buffer sizes.
	- The appendix mentions that the two methods have very different training schedules. It is not mentioned how the hyperparameters were found. It is unclear if hybrid is better because of the schedule or because of the better replay.

Overall, my general evaluation is that, if the paper can properly motivate the very limiting assumption (freshness), the theoretical results could be interesting, although they currently need further work on the presentation in the main text. Instead, the experiments and proposed methodology are currently insufficient and require extensive work. In its current version, I believe the paper would even improve by completely removing the experimental part and focusing on the theory.

**Questions:**

See weaknesses above.

---

### Official Review · Reviewer_WNAZ · 2024-10-31

**Soundness:** 2
**Presentation:** 2
**Contribution:** 2
**Rating:** 3
**Confidence:** 4

**Summary:**

This paper studies two types of replay in continual learning using a linear model. The first type, concurrent replay, is when samples stored in memory are combined with current task's samples during training. This type of replay is commonly practiced and is contrasted with a new type of replay, called sequential, where samples from previous tasks are replayed sequentially, and one task at time.
These two types of replay are compared by examining closed forms of forgetting and generalization under some distributional assumptions. The main finding is that if the underlying parameters for the sequence of tasks are different, then sequential replay would have a better performance than concurrent one. This insight is used to design and algorithm for replay. Simulations using linear models and experiments using DNN are presented to support the theoretical findings.

**Strengths:**

The main question that the paper is asking is original and interesting. Analysis of sample replay in this setting is also novel, and the explicit forms presented for forgetting and generalization are interesting and seem to have the potential to lead to useful insights.
The paper is well organized.

**Weaknesses:**

Setup: It is not clear to me that the setup is appropriate for the main question of the paper. The linear setup is such that if the tasks are different, the model might not have the capacity to learn more than one task. The paper does not discuss this limitation and how it is affected by overparameterization. One way to address this is to find out what the best linear function would be in this setting.
A related issue is scale of forgetting/generalization. It's easy to see that the generalization error of all zeros vector would be  proportional to $ \sum_i^{T-1}   \lVert  w_i^* \rVert $, while the generalization error they have given for the model parameters also has a similar term with multipliers $d_{0T}$ , plus additional terms. It is not clear to me when/if  generalization error of the trained model's parameters is smaller than that of all zeros vector.  This makes it difficult to judge the significance of the results.

The setup indicates that the true parameters $w_i^*$ are sparse but it seems that this is not used in the proofs, since there is no sparsity parameter in the statement of the results. Is this needed?

The setup, which is very similar to the one in Lin et al.(2023), is presented to be more "comprehensive" than the one in Evron et al. (2022) (see line 132). I find this misleading, since in this setup all the tasks share the same Normal distribution on covariates, while in Evron et al (2022) the tasks were arbitrary subspaces. I think the setups are simply not comparable.

Clarity and organization:
I find that there is not enough emphasis and exploration of details that would give more intuition or insight into how results were obtained and why they hold. For example, in lines 80-83, they discuss partitioning the training data to be able to analyze replay. It is not clear to me whether this is an assumption or just the technique used. This is not further discussed in the main body of the paper. Another example is the replay setup. In lines 177-178, it is stated that "memory data are all fresh", does that mean that replay is only done with the last task?
If this is the case, then it should be stated clearly. Additionally, it is not clear how the replay samples are picked.

Clarity of the theoretical results:
Based on equations 5 and 6, forgetting and generalization are defined for $w_T$ which is random, because of random samples. This means that the explicit forms of forgetting and generalization given in Theorem 1 (equation 9) must also be random. However, the coefficients given there are not random (proposition 1). Is there an expectation missing somewhere?

In line 325, it is stated that "the expected value of forgetting and generalization error approach 0 when $p \rightarrow \infty$. I think that is incorrect based on Proposition 1. When $p \rightarrow \infty$ and other variables are fixed, $r_a \rightarrow 1$. This means that at least the first term in equation 9 would not go to zero.


It is not clear to me that the algorithm presented is practical, since it needs to calculate the tasks' gaps, which depend on the ground truth parameters.  Specifically, the procedure DIVIDEBUFFER is not defined in the paper.

Experiments:
In the experiments using DNNs, it is not clear how replay is implemented. It is also not clear how the experimental setup is related to the theoretical results. In the theoretical results, all tasks share the same distribution for input samples ($x$ ), and the parameters $w^*_i$ are different among tasks. In the experiments, the input samples are images, which are corrupted.  Additionally, there is no measure of uncertainty reported for the numbers in Table 1.

**Questions:**

I think it would help the clarity of the results if some details that are presented in the main body of the paper are moved to the appendix and more intuition is presented in the main body of the paper instead. For example, in section 3 on page 3, notation is introduced that will not appear throughout the rest of the main body of the paper.  Proposition 1 is dense and is not completely unpacked throughout the rest of the paper. Is it essential to display the forms of these coefficients at this level of detail?
It would perhaps also be helpful to emphasize how the coefficients in Theorem 1 depend on the parameters of the problem ($n$, $p$, ...).
In general, I think it would be helpful to discuss the behavior of the coefficients.

why does the expression for $d_{ijkt}^(sequential)$ only include cases where $k=I$ while the concurrent one doesn't?

In terms of experiments, I would encourage the authors to think of experimental setups that are more in line with the setup of the paper. For example, instead of corrupting images, they could swap labels of different classes, I believe this would be more in line with a shift in ground truth parameters.
Additionally, running the experiments multiple times and reporting confidence intervals would be helpful.

It would be also helpful to discuss whether Algorithm 1 is feasible without having access to the ground truths. If that is not the case, then this is a serious limitation that should be mentioned.

---

### Official Review · Reviewer_AYZN · 2024-11-03

**Soundness:** 3
**Presentation:** 3
**Contribution:** 2
**Rating:** 5
**Confidence:** 4

**Summary:**

This paper studies concurrent and sequential replay strategies in continual learning (CL). It finds that sequential replay is more effective for dissimilar tasks and proposes a hybrid approach that combines both methods. Experiments show that this hybrid strategy reduces forgetting and improves performance, providing a new perspective for future CL designs.

**Strengths:**

1. This paper introduces a new perspective in CL, questioning the standard concurrent replay method and proposing a sequential replay strategy.
2. The authors provide a theoretical analysis based on overparameterized linear models. They give explicit closed-form solutions of forgetting and generalization errors for both concurrent and sequential replay.
3. For two-task case, they give a comparison between concurrent and sequential replay and highlight the conditions under which each approach performs better than the other.
3. Based on their theoretical analyses, the authors propose a hybrid replay method that leverages both sequential and concurrent replay strategies based on task similarity. They conduct experiments to validate their theoretical findings.

**Weaknesses:**

1. The theoretical comparison of two replay methods is derived for two-task case, which may limit their applicability to real-world scenarios.
2. The performance of the hybrid approach relies on the accuracy of the task similarity measure, which is difficulty to estimate in practice.
3. They don't provide theoretical analyses for the proposed hybrid replay method. Therefore, I think their theoretical contribution is limited.

**Questions:**

1. I wonder whether it's possible to figure out a better algorithm to choose replay samples from old tasks, rather than just assuming that they are newly sampled? I think this question is more interesting for both theoretical research and pratical application.

---

### Note · Authors · 2024-12-02

I have read and agree with the venue's withdrawal policy on behalf of myself and my co-authors.